# The Impact of Uncertainty on Regularized Learning in Games

Pierre-Louis Cauvin [1]  Davide Legacci [1]  Panayotis Mertikopoulos [1]

## Abstract

In this paper, we investigate how randomness and uncertainty influence learning in games. Specifically, we examine a perturbed variant of the dynamics of "follow-the-regularized-leader" (FTRL), where the players' payoff observations and strategy updates are continually impacted by random shocks. Our findings reveal that, in a fairly precise sense, "uncertainty favors extremes": in *any* game, regardless of the noise level, every player's trajectory of play reaches an arbitrarily small neighborhood of a pure strategy in finite time (which we estimate). Moreover, even if the player does not ultimately settle at this strategy, they return arbitrarily close to some (possibly different) pure strategy infinitely often. This prompts the question of which sets of pure strategies emerge as robust predictions of learning under uncertainty. We show that (*a*) the only possible limits of the FTRL dynamics under uncertainty are pure Nash equilibria; and (*b*) a span of pure strategies is stable and attracting if and only if it is *closed under better replies*. Finally, we turn to games where the deterministic dynamics are recurrent—such as zero-sum games with interior equilibria—and show that randomness disrupts this behavior, causing the stochastic dynamics to drift toward the boundary on average.

## 1  Introduction

The surge of breakthrough applications of game theory to machine learning and data science—from multi-agent reinforcement learning to online ad auctions and recommender systems—has brought to the forefront the fundamental question of "as if" rationality: whether selfishly-minded, myopic agents may eventually learn to behave "as if" they were fully rational. This question dates back to Nash [53, p. 21] who observed that, in the presence of uncertainty, "the players of the game may not have full knowledge of [the game's] structure, or the ability and inclination to go through any complex reasoning process to calculate an equilibrium. But the participants are still supposed to adapt by accumulating empirical information on the relative advantages of the various pure strategies at their disposal."

A natural context to study this question is that of no-regret learning algorithms and dynamics—and, in particular, the family of learning dynamics known as *"follow-the-regularized-leader"* (FTRL). The leading example of this class is the replicator dynamics of Taylor & Jonker [63], the continuous-time limit of the exponential / multiplicative weights (EW) algorithm [4, 5, 40, 64], and a central object of study in a rich literature at the intersection of learning theory, games, and optimization, cf. the textbook treatments of Hofbauer & Sigmund [25], Weibull [66] and Sandholm [59]. The FTRL dynamics enjoy a broad range of appealing properties—from (near-)optimal regret guarantees [33, 62] and convergence to equilibrium in potential games [23, 43], to a version of the folk theorem of evolutionary game theory [26, 42]—so, together with their many variants and extensions, they have become virtually synonymous with sequential learning in games.

At the same time, many of the most powerful and widely used results of FTRL rely—often implicitly—on having access to perfect, deterministic information and exact knowledge of the players' payoffs. This level of precision is often difficult to justify in real-world decision-making scenarios, whether due to imperfect observations, stochastic fluctuations in the players' environment, or intrinsic variabilities in the observed outcomes. In view of this, we ask the question:

*What is the impact of noise, randomness,*
*and uncertainty on the FTRL dynamics?*

**Our contributions in the context of related work.**   Our findings can be summarized by a disarmingly simple mantra:

*"Uncertainty favors extremes."*

By this, we mean that, irrespective of the structure of the game or the level of noise and uncertainty, the FTRL dynamics tend to favor pure over mixed strategies.

---

[1]Univ. Grenoble Alpes, CNRS, Inria, Grenoble INP, LIG, 38000 Grenoble, France. Correspondence to: Pierre-Louis Cauvin <pierre-louis.cauvin@univ-grenoble-alpes.fr>.

*Proceedings of the 42nd International Conference on Machine Learning*, Vancouver, Canada. PMLR 267, 2025. Copyright 2025 by the author(s).

In more detail, our model for noise and uncertainty in the dynamics hinges on the introduction of a catch-all, martingale noise term, which is flexible enough to capture all sources of observational uncertainty, stochastic disturbances, and/or any other elements of randomness in the players' learning dynamics. This leads to a stochastic differential equation (SDE) driven by possibly correlated noise, which we refer to as the *stochastic FTRL dynamics*. Earlier dynamics of this type have been considered by Foster & Young [20] (in the context of the replicator dynamics as a model of pairwise proportional imitation), Fudenberg & Harris [21] (in the context of population dynamics), and Mertikopoulos & Moustakas [41] (for learning in games with exponential / multiplicative weights), with follow-up works by Bravo & Mertikopoulos [13], Cabrales [14], Engel & Piliouras [17], Hofbauer & Imhof [24], Imhof [28], and many others. We review the relevant literature in Section 3 and Appendix F.

Our first main result establishes a universal property of the stochastic FTRL dynamics, marking a clear departure from the noiseless, deterministic regime. To state it informally:

> *In any game, for any noise level, every player reaches an almost pure strategy in finite time.*

Even though the dynamics may not settle at this strategy, they will return arbitrarily close to some (possibly different) pure strategy infinitely often, ultimately leading to the following important consequence:

> *The only possible limits of the stochastic FTRL dynamics are pure Nash equilibria.*

The contrapositive of this statement is also significant, because it shows that, unless the underlying game admits a pure Nash equilibrium, the stochastic FTRL dynamics cannot converge. In turn, this prompts the question of which pure strategies are present in sets that are stable and attracting under the stochastic FTRL dynamics. As it turns out, there is a third universal principle at play here:

> *A span of pure strategies is stable and attracting iff it is closed under better replies.*

This echoes earlier results for the replicator dynamics in continuous [56] or discrete time [12], but it otherwise sets the stochastic FTRL dynamics apart from other models of randomness and uncertainty—namely the stochastic dynamics of Foster & Young [20] and Fudenberg & Harris [21] which *do not* satisfy this principle. In particular, it encompasses as a special case the following equivalence: *a state is stochastically asymptotically stable if and only if it is a strict Nash equilibrium.* This principle is sometimes referred to

as the "folk theorem" of evolutionary game theory, and it mirrors the deterministic version of [18], and the discrete-time analogue of [22]. Still, it is important to underline that this equivalence fails outright in other stochastic models of learning, indicating that the way that uncertainty enters the process has a critical impact on the dynamics' rationality properties.

Finally, we analyze the impact of uncertainty in cases where the deterministic FTRL dynamics are recurrent—that is, they return infinitely often arbitrarily close to where they started. This class of games includes two-player zero-sum games with a fully mixed equilibrium [48, 55] and, more broadly, the class of harmonic games [1, 15, 34]. In stark contrast to the deterministic case, uncertainty causes the dynamics to drift on average toward the boundary, escaping from any compact set of fully mixed strategies in finite time, and taking infinite time to return to it once outside.

All these results underline the central mantra that "uncertainty favors extremes". Specifically, in the presence of randomness and uncertainty, players are much more likely to alternate indefinitely—and randomly—between their pure strategies instead of staying close to a mixed equilibrium. This conclusion has important consequences for predicting the outcome of a learning process as it underscores the fragility of mixed equilibria in an unequivocal manner.

## 2 Preliminaries

**2.1. Basic definitions from game theory.**   Throughout our paper, we will work with *finite games in normal form*. A game of this type consists of the following primitives:

1. A finite set of *players* indexed by $i \in \mathcal{N} = \{1, \dots, N\}$.

2. For each player $i \in \mathcal{N}$, a finite set of *actions*—or *pure strategies*—indexed by $\alpha_i \in \mathcal{A}_i = \{1, \dots, n_i\}$.

3. For each player $i \in \mathcal{N}$, an associated *payoff function* $u_i \colon \mathcal{A} \to \mathbb{R}$, where $\mathcal{A} = \prod_j \mathcal{A}_j$ denotes the game's space of action profiles $\alpha = (\alpha_1, \dots, \alpha_N)$.

A finite game as above will be denoted by $\Gamma \equiv \Gamma(\mathcal{N}, \mathcal{A}, u)$.

When playing the game, each player $i \in \mathcal{N}$ may randomize their choice of action by means of a *mixed strategy*, that is, a probability distribution $x_i$ over $\mathcal{A}_i$. In terms of notation, we will write $x_{i\alpha_i}$ for the probability with which player $i \in \mathcal{N}$ selects $\alpha_i \in \mathcal{A}_i$, and $\mathcal{X}_i := \Delta(\mathcal{A}_i)$ for the strategy space of player $i \in \mathcal{N}$ (that is, the probability simplex over $\mathcal{A}_i$). Also, in a slight—but useful—abuse of notation, we will identify $\alpha_i \in \mathcal{A}_i$ with the mixed strategy that assigns all probability weight to $\alpha_i$, thus justifying the terminology "pure strategies" for the elements of $\mathcal{A}_i$.

Moving forward, we will write $x = (x_1, \dots, x_N)$ for the players' *mixed strategy profile* and $\mathcal{X} \equiv \prod_i \mathcal{X}_i$ for the

game's *strategy space*. We will also employ the standard game-theoretic shorthand $(x_i; x_{-i}) = (x_1, \ldots, x_i, \ldots, x_N)$ for the strategy profile where player $i$ plays $x_i \in \mathcal{X}_i$ against the mixed strategy $x_{-i} \in \prod_{j \neq i} \mathcal{X}_j$ of all other players. Accordingly, the associated *mixed payoff* of player $i \in \mathcal{N}$ in the mixed strategy profile $x \in \mathcal{X}$ is given by

$$u_i(x) := \mathbb{E}_{\alpha \sim x}[u_i(\alpha)] = \sum_{\alpha \in \mathcal{A}} u_i(\alpha) x_\alpha \qquad (1)$$

where $x_\alpha$ denotes the joint probability of playing $\alpha \in \mathcal{A}$ under $x \in \mathcal{X}$. To distinguish between "strategy-like" and "payoff-like" variables, we will write $\mathcal{Y}_i := \mathbb{R}^{\mathcal{A}_i}$ and $\mathcal{Y} = \prod_i \mathcal{Y}_i$ for the game's "*payoff space*", in direct analogy with the game's strategy space $\mathcal{X} = \prod_i \mathcal{X}_i$.

More concisely, a game in normal form can be fully described by its *payoff field* $v(x) = (v_i(x))_{i \in \mathcal{N}}$ where

$$v_i(x) := (u_i(\alpha_i; x_{-i}))_{\alpha_i \in \mathcal{A}_i} = \nabla_{x_i} u_i(x) \qquad (2)$$

denotes the *mixed payoff vector* of player $i \in \mathcal{N}$, that is, the vector of payoffs to pure strategies of player $i$ against the mixed strategy profile $x_{-i}$ of $i$'s opponents. The game's mixed payoffs may then be recovered from (2) as

$$u_i(x) = \sum_{\alpha_i \in \mathcal{A}_i} u_i(\alpha_i; x_{-i}) x_{i\alpha_i} = \langle v_i(x), x_i \rangle \qquad (3)$$

so $v(x)$ captures all relevant payoff information in the game.

**2.2. Nash equilibria.** The leading solution concept in game theory is that of a *Nash equilibrium* (NE). Formally, $x^* \in \mathcal{X}$ is said to be a Nash equilibrium if it is unilaterally stable in the sense that

$$u_i(x^*) \geq u_i(x_i; x^*_{-i}) \quad \text{for all } x_i \in \mathcal{X}_i, i \in \mathcal{N}. \qquad \text{(NE)}$$

Writing $\text{supp}(x_i^*) = \{\alpha_i \in \mathcal{A}_i : x^*_{i\alpha_i} > 0\}$ for the support of $x_i^*$, it follows that $x^*$ is a Nash equilibrium if and only if

$$u_i(\alpha_i; x^*_{-i}) \geq u_i(\beta_i; x^*_{-i}) \qquad (4)$$

for all $\alpha_i \in \text{supp}(x_i^*)$ and all $\beta_i \in \mathcal{A}_i$. In turn, this characterization leads to the following taxonomy for Nash equilibria:

1. $x^*$ is called *strict* if inequality (4) is strict for all $\beta_i \neq \alpha_i$.

2. $x^*$ is called *pure* if $\text{supp}(x^*)$ is a singleton.

3. If $x^*$ is not pure, we say that it is *mixed*; and if $\text{supp}(x^*) = \mathcal{A}$, we say that $x^*$ is *fully mixed*.

We will revisit this classification several times in the sequel.

**2.3. Regret and regularized learning.** A fundamental requirement in online learning is the minimization of the player's *regret*, i.e., the aggregate payoff difference between a player's chosen strategy over time and the best fixed strategy in hindsight. Formally, assuming that play evolves

in continuous time, the (external) *regret* of player $i \in \mathcal{N}$ relative to a trajectory of play $x(t)$, $t \in [0, \infty)$ is defined as

$$\text{Reg}_i(T) = \max_{p_i \in \mathcal{X}_i} \int_0^T [u_i(p_i; x_{-i}(t)) - u_i(x(t))] \, dt \qquad (5)$$

and we say that the player has *no regret* under $x(t)$ if $\text{Reg}_i(T) = o(T)$ as $T \to \infty$.

The most widely used method to attain no regret is the family of policies known as *"follow-the-regularized-leader"* (FTRL) [42, 60, 61]. The main idea behind FTRL is that, at all times $t \geq 0$, each player $i \in \mathcal{N}$ plays a "regularized" best response to their cumulative payoff vector up to time $t$, thus leading to the dynamics

$$\underbrace{\dot{y}_i(t) = v_i(x(t))}_{\text{aggregate payoffs}} \qquad \underbrace{x_i(t) = Q_i(y_i(t))}_{\text{strategy update}} \qquad \text{(FTRL)}$$

where

$$Q_i(y_i) = \arg\max_{x_i \in \mathcal{X}_i} \{\langle y_i, x_i \rangle - h_i(x_i)\} \qquad (6)$$

denotes the *regularized best response map* of player $i \in \mathcal{N}$.

In the above, $y_i \in \mathcal{Y}_i = \mathbb{R}^{\mathcal{A}_i}$ plays the role of an auxiliary "score vector" that encodes the empirical performance of each strategy of player $i \in \mathcal{N}$ over time. As for the function $h_i$ that appears in the definition of $Q_i$, it is known as the method's *regularizer*, and it aims to "temper" the best-response correspondence $y_i \mapsto \arg\max_{p_i \in \mathcal{X}_i} \langle y_i, x_i \rangle$ in a way that incentivizes exploration.

The precise assumptions for $h_i$ in the literature can be somewhat varied. For our purposes—and to streamline our presentation—we will assume that $h_i$ decomposes as

$$h_i(x_i) = \sum_{\alpha_i \in \mathcal{A}_i} \theta_i(x_{i\alpha_i}) \qquad (7)$$

for some smooth kernel function $\theta_i \colon (0, 1) \to \mathbb{R}$ such that $\lim_{z \to 0^+} \theta_i'(z) = -\infty$ and $\inf_z \theta_i''(z) > 0$ (so $\theta_i$ is strongly convex and steep at 0). Finally, to simplify some expressions in the sequel, we will assume that $\theta_i'''(z) < 0$ for all $z \in (0, 1)$; however, we stress that this assumption is only made to render some expressions more transparent, and it does not affect the gist of our results.

For concreteness, we present below two prime examples of (FTRL).

**Example 1** (Entropic regularization). The go-to setup for (FTRL) is the entropic kernel $\theta_i(z) = z \log z$. By standard results, (6) yields the so-called *logit choice map*

$$\Lambda_i(y_i) := \frac{(\exp(y_{i\alpha_i}))_{\alpha_i \in \mathcal{A}_i}}{\sum_{\alpha_i \in \mathcal{A}_i} \exp(y_{i\alpha_i})}, \qquad (8)$$

and the resulting instance of (FTRL) is known as the *exponential*—or *multiplicative*—*weights algorithm*, cf. [3–5, 40, 64] and references therein. By another standard calculation, the evolution of $x(t)$ under (FTRL) follows the

replicator dynamics of Taylor & Jonker [63], viz.

$$\dot{x}_{i\alpha_i} = x_{i\alpha_i}[v_{i\alpha_i}(x) - u_i(x)]. \qquad \text{(RD)}$$

This is one of the most widely studied models for learning in games, and it will play a major role in our analysis. §

**Example 2** (Log-barrier regularization). Another standard example is the log-barrier kernel $\theta_i(z) = -\log z$. In this case, a direct derivation (which we omit) yields the *affine scaling dynamics*

$$\dot{x}_{i\alpha_i} = x_{i\alpha_i}^2 \left[ v_{i\alpha_i}(x) - \frac{1}{\|x_i\|_2^2} \sum_{\beta_i \in \mathcal{A}_i} x_{i\beta_i}^2 v_{i\beta_i}(x) \right]. \qquad \text{(AS)}$$

These dynamics have a long and celebrated history in optimization going back to the interior-point algorithms of Dikin [16] and Karmarkar [30, 31]; for a series of recent applications to online learning and multi-armed bandit problems, see [65] and references therein. §

Going back to the general case, Kwon & Mertikopoulos [33] showed that (FTRL) enjoys the constant regret bound

$$\text{Reg}_i(T) \le \max h_i - \min h_i. \qquad \text{(9)}$$

This guarantee justifies the popularity of (FTRL) as a no-regret policy and, in view of this, it will be our dynamics of choice for the sequel.

## 3 The stochastic FTRL dynamics

The dynamics (FTRL) have been studied extensively in the literature, but their applicability to real-world decision-making is constrained by the fact that they explicitly rely on perfect, deterministic information and exact knowledge of the players' payoffs. This level of precision is often difficult to attain in practice, whether due to imperfect observations, stochastic fluctuations in the players' environment, or intrinsic variabilities in the observed outcomes. To address this limitation, we provide below a stochastic version of (FTRL) which explicitly incorporates the effects of randomness and uncertainty, leading in this way to a more robust and realistic model for learning under noisy, uncertain conditions.

**3.1. Learning under uncertainty.** To put all this on a solid footing, we will consider the *stochastic FTRL dynamics*

$$\begin{aligned} dY_i(t) &= v_i(X(t))\, dt + dM_i(t) \\ X_i(t) &= Q_i(Y_i(t)), \end{aligned} \qquad \text{(S-FTRL)}$$

which should be seen as a rigorous formulation of the informal model

$$\dot{y}_i(t) = v_i(x(t)) + \text{"noise"}. \qquad \text{(10)}$$

In more detail, $M_i(t)$, $t \ge 0$, denotes a continuous square-integrable martingale with values in the payoff space $\mathcal{Y}_i =$

$\mathbb{R}^{\mathcal{A}_i}$ of player $i \in \mathcal{N}$, so (S-FTRL) itself represents an ordinary (Itô) stochastic differential equation.[1] In this regard, $M_i(t)$ plays the role of a catch-all, "colored noise" term intended to capture all sources of observational uncertainty, stochastic disturbances, and/or any other elements of randomness in the players' learning model.

More concretely, by the martingale representation theorem [54, Theorem 4.3.4], $M_i(t)$ can be written as $dM_i(t) = A_i(t) \cdot dW(t)$ where $W(t) = (W_k(t))_{1 \le k \le m}$ is an $m$-dimensional Brownian motion and $A_i(t)$ is a matrix-valued process in $\mathbb{R}^{n_i \times m}$.[2] Our only assumption here will be that the diffusion matrices $A_i$ are *state-dependent*, that is, they only depend on time through $X(t)$ as $A_i(t) \equiv \sigma_i(X(t))$ for some Lipschitz function $\sigma_i \colon \mathcal{X} \to \mathbb{R}^{n_i \times m}$. On that account, (S-FTRL) can be expressed in components as

$$dY_{i\alpha_i}(t) = v_{i\alpha_i}(X(t))\, dt + \sum_{k=1}^m \sigma_{i\alpha_i k}(X(t))\, dW_k(t) \quad \text{(11)}$$

or, more compactly, as

$$dY(t) = v(X(t))\, dt + \sigma(X(t)) \cdot dW(t) \qquad \text{(12)}$$

where we set $\sigma \equiv (\sigma_1, \dots, \sigma_m)^\top \in \mathbb{R}^{n \times m}$ for the overarching diffusion matrix of the process. This system will serve as our main model for learning in the presence of uncertainty, so some remarks are in order.

The first concerns the structure of the diffusion matrices $\sigma_i$, which, in their simplest, diagonal form, yield the system

$$dY_{i\alpha_i}(t) = v_{i\alpha_i}(X(t))\, dt + \sigma_{i\alpha_i}(X(t))\, dW_{i\alpha_i}(t) \qquad \text{(13)}$$

where each $W_{i\alpha_i}(t)$, $\alpha_i \in \mathcal{A}_i$, $i \in \mathcal{N}$, is a Brownian motion in $\mathbb{R}$, assumed independent across all $\alpha_i \in \mathcal{A}_i$ and all $i \in \mathcal{N}$. This uncorrelated model of uncertainty was first considered by Bravo & Mertikopoulos [13] and it can be derived as a special case of our general framework by taking $M_i = (\sigma_{i\alpha i} W_{i\alpha_i})_{\alpha_i \in \mathcal{A}_i}$ in (S-FTRL); see also [44, 45].

From a modeling standpoint, while relevant in a wide range of applications, the uncorrelated model (13) overlooks important cases where the players' payoffs are influenced by a common source of randomness – for example, the outcome of the coin toss in Matching Pennies, or the choice of routing path in a congestion game (where the delays along overlapping routes are inherently correlated over their common edges), etc. Capturing such scenarios requires the full extent of our framework, so we will work throughout with general diffusion matrices, and we will only zoom in on the uncorrelated case when it helps make some quantitative estimates more transparent and easier to state.

---

[1]For the requisite background to stochastic differential equations and stochastic analysis, we refer the reader to Øksendal [54].

[2]Formally, $W(t)$, $A_i(t)$ and $M_i(t)$ are all assumed to be adapted to some common, underlying filtration $\mathcal{F}_\bullet = (\mathcal{F}_t)_{t \ge 0}$.

Finally, from a technical perspective, we should note that the stated assumptions guarantee that the system (S-FTRL) is *well-posed*, that is, for any initial condition $Y(0) \in \mathcal{Y}$, it admits a unique strong solution that exists for all time. We state and prove this result formally in Appendix C using the property that the players' regularized best response maps $Q_i \colon \mathcal{Y}_i \to \mathcal{X}_i$ are Lipschitz continuous.

**3.2. Strategy dynamics and other stochastic models.**
Before moving forward with our analysis, we proceed to describe the exact way in which the players' strategies evolve over time under (S-FTRL). The relevant result is as follows:

**Proposition 1.** *The solutions of* (S-FTRL) *satisfy the stochastic differential equation*

$$dX_{i\alpha_i} = g_{i\alpha_i} \left[ v_{i\alpha_i} - \sum_{\beta_i \in \mathcal{A}_i} \chi_{i\beta_i} v_{i\beta_i} \right] dt \tag{14a}$$

$$+ g_{i\alpha_i} \left[ dM_{i\alpha_i} - \sum_{\beta_i \in \mathcal{A}_i} \chi_{i\beta_i} dM_{i\beta_i} \right] \tag{14b}$$

$$+ g_{i\alpha_i} \sum_{k=1}^{m} \left[ \psi_{i\alpha_i k}^2 - \sum_{\beta_i \in \mathcal{A}_i} \chi_{i\beta_i} \psi_{i\beta_i k}^2 \right] dt \tag{14c}$$

*where we set* $g_{i\alpha_i} = 1/\theta_i''(x_{i\alpha_i})$, $\chi_{i\alpha_i} = g_{i\alpha_i} / \sum_{\beta_i \in \mathcal{A}_i} g_{i\beta_i}$, *and* $\psi_{i\alpha_i k}^2 = -\frac{1}{2} \theta_i'''(x_{i\alpha_i}) g_{i\alpha_i}^2 \left[ \sigma_{i\alpha_i k} - \sum_{\beta_i \in \mathcal{A}_i} \chi_{i\beta_i} \sigma_{i\beta_i k} \right]^2$.

The proof of Proposition 1 is an arduous combination of Itô's lemma with elements of convex analysis in the spirit of [13], so we defer it to Appendix C. From a conceptual viewpoint, what is worth noting is that, despite the complicated form of (14), each term admits a relatively simple interpretation:

1. The term (14a) represents the evolution of the players' strategies under (FTRL), i.e., the noiseless regime $\sigma = 0$.

2. The martingale term (14b) captures the direct impact of the noise on the evolution of $X(t)$ under (S-FTRL).

3. Finally, the term (14c) represents the second-order Itô correction that is propagated to $X(t)$ through the players' regularized best response maps $Q_i \colon \mathcal{Y}_i \to \mathcal{X}_i$. This term also vanishes when $\sigma = 0$ but, in contrast to the martingale term (14b), it directly affects the drift of (14) and induces a measurable bias component in the dynamics.

To facilitate the comparison of these dynamics with other models in the literature, we will specialize to the replicator dynamics induced by the entropic setup of Example 1, with uncorrelated noise of the form (13). Here, a straightforward computation gives

$$g_{i\alpha_i} = \chi_{i\alpha_i} = x_{i\alpha_i} \tag{15}$$

and, in a slight abuse of notation

$$\psi_{i\alpha_i}^2 = \left[ \sigma_{i\alpha_i} - \sum_{\beta_i \in \mathcal{A}_i} x_{i\beta_i} \sigma_{i\beta_i} \right]^2. \tag{16}$$

We thus obtain the *stochastic replicator dynamics of exponential weights*

$$dX_{i\alpha_i} = X_{i\alpha_i} \left[ v_{i\alpha_i} - \sum_{\beta_i \in \mathcal{A}_i} X_{i\beta_i} v_{i\beta_i} \right] dt$$

$$+ X_{i\alpha_i} \left[ \sigma_{i\alpha_i} dW_{i\alpha_i} - \sum_{\beta_i \in \mathcal{A}_i} \sigma_{i\beta_i} X_{i\beta_i} dW_{i\beta_i} \right]$$

$$+ \frac{X_{i\alpha_i}}{2} \left[ \sigma_{i\alpha_i}^2 (1 - 2X_{i\alpha_i}) - \sum_{\beta_i \in \mathcal{A}_i} \sigma_{i\beta_i}^2 X_{i\beta_i} (1 - 2X_{i\beta_i}) \right] dt.$$
$$\text{(SRD-EW)}$$

These dynamics were first studied by Mertikopoulos & Moustakas [41] and they should be contrasted to the biological model of the *stochastic replicator dynamics with aggregate shocks* of Fudenberg & Harris [21], viz.

$$dX_{i\alpha_i} = X_{i\alpha_i} \left[ v_{i\alpha_i} - \sum_{\beta_i \in \mathcal{A}_i} X_{i\beta_i} v_{i\beta_i} \right] dt$$

$$+ X_{i\alpha_i} \left[ \sigma_{i\alpha_i} dW_{i\alpha_i} - \sum_{\beta_i \in \mathcal{A}_i} \sigma_{i\beta_i} X_{i\beta_i} dW_{i\beta_i} \right]$$

$$- X_{i\alpha_i} \left[ \sigma_{i\alpha_i}^2 X_{i\alpha_i} - \sum_{\beta_i \in \mathcal{A}_i} \sigma_{i\beta_i}^2 X_{i\beta_i}^2 \right] dt.$$
$$\text{(SRD-AS)}$$

Finally, a third related model is the *stochastic replicator dynamics of pairwise imitation* of Foster & Young [20]:

$$dX_{i\alpha_i} = X_{i\alpha_i} \left[ v_{i\alpha_i} - \sum_{\beta_i \in \mathcal{A}_i} X_{i\beta_i} v_{i\beta_i} \right] dt$$

$$+ X_{i\alpha_i} \left[ \sigma_{i\alpha_i} dW_{i\alpha_i} - \sum_{\beta_i \in \mathcal{A}_i} \sigma_{i\beta_i} X_{i\beta_i} dW_{i\beta_i} \right].$$
$$\text{(SRD-PI)}$$

The origins of the latter two models are quite distinct from (S-FTRL), and they do not stem from learning considerations: (SRD-AS) was originally derived as a biological model of population evolution under "aggregate shocks" to the population's reproductive fitness, while (SRD-PI) is based on economic microfoundations involving revision protocols in population games [46]. These models only coincide in the noiseless, deterministic regime; otherwise, in the presence of noise and uncertainty, the drift is distinct (because of the Itô correction), and as we show in the next section, this leads to drastically different behaviors in the long-run.

## 4 Analysis and results

We now proceed to state our main results for the stochastic dynamics (S-FTRL). To fix notation, we will assume throughout that (S-FTRL) is initialized at some fixed $x \in \mathcal{X}$, and we will write $\mathbb{P}_x$ and $\mathbb{E}_x$ for the law of the process starting at $X(0) \leftarrow x$ and the induced expectation thereof. Also,

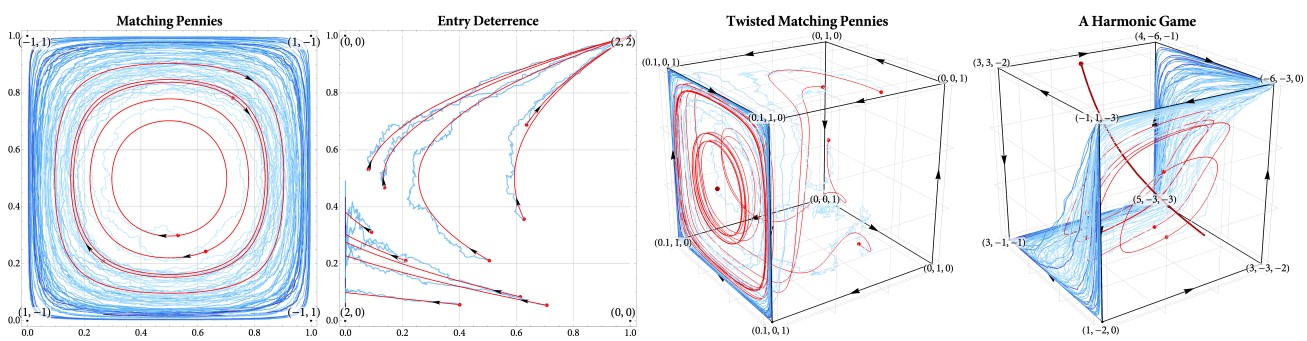

**Figure 1:** Trajectories of play under (FTRL) and (S-FTRL) in four different games (each game's payoffs are reported on the corresponding plot). Deterministic orbits are plotted in red, stochastic trajectories in shades of blue, with darker hues indicating later points in time. In all cases, the trajectories of (S-FTRL) drift toward the boundary, even when the corresponding deterministic orbits of (FTRL) do not.

to quantify the level of noise and uncertainty in (S-FTRL), we will consider the metrics

$$\sigma_{i,\min}^2 := \min_{x \in \mathcal{X}} \lambda_{\min}(\Sigma_i(x)) \quad \sigma_{\min}^2 := \min_{x \in \mathcal{X}} \lambda_{\min}(\Sigma(x))$$

$$\sigma_{i,\max}^2 := \max_{x \in \mathcal{X}} \lambda_{\max}(\Sigma_i(x)) \quad \sigma_{\max}^2 := \max_{x \in \mathcal{X}} \lambda_{\max}(\Sigma(x))$$
(17)

where $\Sigma_i \equiv \sigma_i \sigma_i^\top$ and $\Sigma \equiv \sigma \sigma^\top$ are the quadratic covariation matrices of the noise in (S-FTRL), and $\lambda_{\min}$ (resp. $\lambda_{\max}$) denote the minimum (resp. maximum) eigenvalues thereof.

To state our results, we will require the assumptions below:

**Assumption 1.** The noise in (S-FTRL) has $\sigma_{\min}^2 > 0$.

**Assumption 2.** The kernel functions $\theta_i$ enjoy the bound

$$\sup_{0 < z < 1} |\theta_i'''(z)| / [\theta_i''(z)]^2 < \infty \quad \text{for all } i \in \mathcal{N}. \quad (18)$$

Finally, for some—but not all—of our results, we will require the following boundedness assumption:

**Assumption 3.** The kernel functions $\theta_i$ are *bounded*, i.e., $\sup_{z \in (0,1)} |\theta_i(z)| < \infty$ for all $i \in \mathcal{N}$.

Of the above, Assumption 1 simply serves as a "certificate of uncertainty" that differentiates the stochastic dynamics (S-FTRL) from their deterministic counterpart (the case $\sigma = 0$). As for Assumption 2, this is a technical requirement which is satisfied by all standard regularizers used in practice (negentropy, Tsallis, log-barrier, etc.), so it is also very mild. Finally, Assumption 3 is also technical in nature, and it is used to facilitate certain bounds in the sequel; of all the examples mentioned so far, only the log-barrier kernel is excluded by this requirement.

**4.1. Playing almost pure strategies infinitely often.** Our first result for (S-FTRL) indicates a radical departure from the deterministic setting: it shows that, in *any* game, regardless of initialization, *every player reaches an arbitrarily small neighborhood of one of their pure strategies in finite time.*

---

**Theorem 1.** *Suppose Assumptions 1 and 2 hold, fix a sufficiently small accuracy threshold $\varepsilon > 0$, and let*

$$\tau_{i,\varepsilon} = \inf\{t \geq 0 : \max_{\alpha_i \in \mathcal{A}_i} X_{i\alpha_i}(t) \geq 1 - \varepsilon\} \quad (19)$$

*denote the time player $i \in \mathcal{N}$ takes to reach an $\varepsilon$-neighborhood of one of their pure strategies. Then $\tau_{i,\varepsilon}$ is finite with probability $1$, and we have*

$$\mathbb{E}_x[\tau_{i,\varepsilon}] \lesssim e^\lambda / \lambda \quad (20)$$

*where $\lambda > 0$ is a positive constant that scales as*

$$\lambda = \Theta\left(\frac{1 + \sigma_{i,\max}^2}{\sigma_{i,\min}^2} \theta_i''\left(\frac{\varepsilon}{n_i - 1}\right)^2\right). \quad (21)$$

---

We stress here that (*i*) Theorem 1 applies to *any* game; and (*ii*) the behavior described in the theorem is intimately tied to the stochastic nature of (S-FTRL). Indeed, in the noiseless, deterministic regime, the assertion of Theorem 1 is patently false: for example, in the $2 \times 2$ game of Matching Pennies, it is well known that the deterministic replicator dynamics—and, more generally, (FTRL)—cycle periodically at a constant distance from the game's unique Nash equilibrium [48, 55], so the conclusion of Theorem 1 does not hold in this case (cf. Fig. 1).

From a technical standpoint, the proof of Theorem 1 hinges on crafting a suitable Lyapunov function in the spirit of Imhof [28], and then bounding the action of the infinitesimal generator of the strategy dynamics (14) on said function in order to apply Dynkin's formula on $\tau_{i,\varepsilon}$. These estimates and the resulting calculations are fairly delicate and involved, so we defer all relevant details to Appendix D.

From a more high-level, conceptual viewpoint, Theorem 1 shows that, in a fairly precise sense, uncertainty favors pure strategies. This is further reinforced by the following consequences of Theorem 1:

**Corollary 1.** *Suppose that Assumptions 1 and 2 hold. Then, for every player $i \in \mathcal{N}$, the limit set of $X_i(t)$ contains a pure strategy with probability 1: specifically, there exists a (possibly random) sequence of times $t_n \nearrow \infty$ such that $X_i(t_n)$ converges to some pure strategy $\alpha_i \in \mathcal{A}_i$.*

Intuitively, under (S-FTRL), $X_i(t)$ reaches a neighborhood of some pure strategy in finite time. Even if $X_i(t)$ does not settle there, the strong Markov property of $X(t)$ and a repeated application of Theorem 1 shows that $X_i(t)$ approaches some other pure strategy, and so on ad infinitum, which ultimately yields the stated result. In fact, arguing in a similar way, we have the following guarantee:

**Corollary 2.** *Suppose Assumptions 1 and 2 hold. Then, with probability 1, there exists a (random) sequence $t_n \nearrow \infty$ such that $X(t_n)$ converges to the boundary $\mathrm{bd}(\mathcal{X})$ of $\mathcal{X}$.*

As in the case of Theorem 1, this assertion is demonstrably false in a deterministic context: focusing again on Matching Pennies, periodicity implies that the orbits of (FTRL) are bounded away from the boundary of $\mathcal{X}$, so the conclusion of Corollary 2 is false in this case. In fact, putting everything together, we obtain a much stronger characterization of the possible limits of the players' learning process:

**Corollary 3.** *Suppose Assumptions 1 and 2 hold. If*
$$\mathbb{P}_x(\lim_{t \to \infty} X(t) = \hat{x}) > 0 \qquad (22)$$
*then $\hat{x}$ must be pure; in words, the only possible limits of (S-FTRL) are pure strategies.*

A revealing illustration of Corollary 3 is the Entry Deterrence game in Fig. 1: in the deterministic case, around half of the initializations of (FTRL) converge to non-pure states; however, under the slightest degree of uncertainty, convergence to non-pure profiles is no longer possible.

All this prompts the following important questions:

1. Are *all* pure strategies valid candidates for convergence?
2. More generally, which pure strategies are present in sets that are stable and attracting under (S-FTRL)?

We devote the rest of this section to these two questions; all relevant proofs are deferred to Appendix D.

**4.2. The support of stable attractors.** We begin with the second question, which is more general. To that end, we first need to define the notion of "stable and attracting":

**Definition 1.** A subset $\mathcal{S} \subseteq \mathcal{X}$ is said to be *stochastically asymptotically stable* under (S-FTRL) if, for every tolerance level $\delta > 0$ and every neighborhood $\mathcal{U}$ of $\mathcal{S}$, there exists a neighborhood $\mathcal{U}_\delta$ of $\mathcal{S}$ such that $X(0) = x \in \mathcal{U}_\delta$ implies
$$\mathbb{P}_x\left( X(t) \in \mathcal{U} \text{ for all } t \geq 0 \text{ and } X(t) \xrightarrow[t \to \infty]{} \mathcal{S} \right) \geq 1 - \delta \quad (23)$$

With this in mind, let $\mathcal{B} = \prod_i \mathcal{B}_i$, $\mathcal{B}_i \subseteq \mathcal{A}_i$, be the set of pure strategies contained in some stable attractor of (S-FTRL), and let $\mathcal{S} = \prod_i \Delta(\mathcal{B}_i)$ denote the *span* of $\mathcal{B}$, i.e., the smallest face of $\mathcal{X}$ that contains all strategies in $\mathcal{B}$. We then ask:

> *When is $\mathcal{S}$ stochastically asymptotically stable under (S-FTRL)?*

As it turns out, this question admits a remarkably simple and compelling answer that hinges on a criterion of strategic stability known as *closedness under better replies*. To make this precise, following Ritzberger & Weibull [56], the set of *better replies* to $x \in \mathcal{X}$ is defined as

$$\mathtt{btr}(x) = \{q \in \mathcal{X} : u_i(q_i; x_{-i}) \geq u_i(x) \text{ for all } i \in \mathcal{N}\} \quad (24)$$

and $\mathcal{S}$ is *closed under better replies* (club) if $\mathtt{btr}(\mathcal{S}) \subseteq \mathcal{S}$. We then have the following striking equivalence:

**Theorem 2.** *Suppose Assumption 3 holds. Then, with notation as above, $\mathcal{S}$ is stochastically asymptotically stable under (S-FTRL) if and only if it is closed under better replies. If this is so, there exists an open initialization domain $\mathcal{U}_0 \subseteq \mathcal{X}$ such that, whenever $X(0) \in \mathcal{U}_0$, we have with arbitrarily high probability*

$$\mathrm{dist}_1(X(t), \mathcal{S}) \lesssim \Phi\left[ c_1 - c_2 t + \mathcal{O}\left( \sigma_{\max} \sqrt{t \log \log t} \right) \right] \tag{25}$$

*where $c_1$ and $c_2$ are constants ($c_2 > 0$), and the rate function $\Phi$ is given by $\Phi(z) = \max_{i \in \mathcal{N}} (\theta_i')^{-1}(z)$.*

Theorem 2 provides a sharp characterization of which spans of pure strategies are stable attractors of regularized learning, and how fast the dynamics converge to such sets. In this regard, Theorem 2 echoes (*i*) a classical finding by Ritzberger & Weibull [56] for (RD) in a *deterministic*, noiseless context;[3] and (*ii*) a more recent result by Boone & Mertikopoulos [12] for a range of regularized learning algorithms in *discrete* time.[4]

In this regard, Theorem 2 might appear unsurprising, but this is not so. The stochastic dynamics (SRD-AS) and (SRD-PI) are also stochastic variants of (RD), but Theorem 2 is false in both cases: in the former because *any* pure strategy may be stochastically asymptotically stable depending on the profile of the noise [24, 28]; in the latter because, in any game, *all* pure strategies are stochastically asymptotically stable for high enough noise [46]. This disconnect has to do with the Itô correction term that appears in the dynamics:

---

[3]Importantly, Theorem 2 also covers the deterministic case $\sigma = 0$; we are not aware of another source for this result in the literature.

[4]We note here in passing that there is a technical gap in the proof of [12] regarding stochastic asymptotic stability (and, more precisely, on establishing convergence for a *neighborhood* of initial conditions). We detail the issue in Appendix D.

in the case of (14), this term is "just right" so, even though uncertainty draws the dynamics toward the boundary, there is sufficient information remaining to identify the rationally admissible parts thereof.

Our proof strategy involves working directly with the score variables $Y(t)$, and hinges on establishing a sort of local "domination" result for those strategies that are not supported in $\mathcal{S}$ versus those that are. This has to be established in a neighborhood of $\mathcal{S}$, and doing this requires Assumption 3 in order to establish a lower and upper bounds for the distance from $\mathcal{S}$ relative to the ensuing score differences. This also requires using a different set of energy functions associated to $\mathcal{S}$, a step which in turn involves several fairly technical estimates. To streamline our presentation, we defer all relevant details to Appendix D.2.

**4.3. Nash equilibria and convergence.**   We now turn to our first question, namely the characterization of the possible limits of (S-FTRL). Our main result here is as follows:

> **Theorem 3.** *Suppose Assumptions 1–3 hold. Then:*
> 1. *$x^* \in \mathcal{X}$ is stochastically asymptotically stable under (S-FTRL) if and only if it is a strict equilibrium.*
> 2. *If $\mathbb{P}_x(\lim_{t\to\infty} X(t) = x^*) > 0$, then $x^*$ is a pure Nash equilibrium; in words, the only possible limits of (S-FTRL) are pure Nash equilibria.*

Since a pure profile is closed under better replies if and only if it is a strict Nash equilibrium, the first part of Theorem 3 is an immediate corollary of Theorem 2. Importantly, as was shown in [18], a similar conclusion holds under the noiseless dynamics (FTRL)—see also [22] for a discrete-time analogue. On that account, the more interesting part of Theorem 3 is the second one, which represents a drastic departure from the deterministic regime: it complements Corollary 3 in an essential way, showing that the possible limits of (S-FTRL) are not just pure strategies, but *pure Nash equilibria*. In other words:

> *If a game does not admit a pure Nash equilibrium, the dynamics (S-FTRL) do not converge.*

This assertion is unique to the stochastic setting and it offers a vivid demonstration of how "uncertainty favors extremes".

## 5  Consequences for recurrent dynamics

The results of the previous section show that, in the presence of noise, the dynamics of regularized learning tend to drift toward the boundary. This behavior comes into stark contrast with the deterministic regime where, in several classes of games, the dynamics of (FTRL) are known to be *Poincaré recurrent* – that is, almost all trajectories return

infinitely often arbitrarily close to their starting point. In this last section, our aim is to understand this disparity in greater depth, by zooming in on the behavior of (S-FTRL) in cases where the deterministic dynamics are Poincaré recurrent.

To provide some context, Poincaré recurrence was first established for (RD) in two-player zero-sum games with a fully mixed Nash equilibrium [55]. This result was subsequently extended to all FTRL dynamics, always for zero-sum games with a fully mixed Nash equilibrium [48], and finally, in a very recent paper [34], to a much more general class of games known as *harmonic games* [15]. This is the widest class of games known to exhibit recurrent behavior under (FTRL), and they can be characterized as follows (cf. Proposition E.1 in Appendix E): a game is harmonic whenever it admits a collection of positive weights $m_i > 0$, $i \in \mathcal{N}$, and a fully mixed strategy profile $q \in \operatorname{ri} \mathcal{X}$ – the game's *harmonic center* – such that

$$\sum_{i \in \mathcal{N}} m_i \langle v_i(x), x_i - q_i \rangle = 0 \quad \text{for all } x \in \mathcal{X}. \qquad (26)$$

A key property of harmonic games is that (FTRL) admits a *constant of motion* given by the expression

$$H(x) = \sum_{i \in \mathcal{N}} m_i D_i(q_i, x_i) \qquad (27)$$

where $D_i(q_i, x_i) = h_i(q_i) - h_i(x_i) - \langle \nabla h_i(x_i), q_i - x_i \rangle$ is the so-called *Bregman divergence* of the regularizer of player $i \in \mathcal{N}$. To streamline our presentation, we defer a detailed presentation of Bregman divergences to Appendix A; for our purposes, it suffices to keep in mind that $D_i$ can be seen as an asymmetric measure of distance between $q_i$ and $x_i$, with $D_i(q_i, x_i) = 0$ if and only if $x_i = q_i$, and $D_i(q_i, x_i) \to \infty$ whenever $x_i \to \operatorname{bd}(\mathcal{X}_i)$. As was shown in [34], the energy $H(x(t))$ of an orbit $x(t)$ of (FTRL) remains constant, so the trajectories of (FTRL) in harmonic games are contained away from the boundary $\operatorname{bd}(\mathcal{X})$ of $\mathcal{X}$, cf. Fig. 1.

By contrast, the behavior of (S-FTRL) in harmonic games is drastically different, even with the least amount of noise:

> **Theorem 4.** *Suppose Assumption 1 holds. Then, in any harmonic game, we have $\lim_{t\to\infty} \mathbb{E}[H(X(t))] = \infty$. Moreover, the time $\tau_M := \inf\{t \geq 0 : H(X(t)) > M\}$ that $X(t)$ takes to reach an energy level $M > 0$ is finite with probability 1 and bounded in expectation as*
>
> $$\mathbb{E}_x[\tau_M] \leq 2 \frac{M - H(x)}{\sigma_{\min}^2 \varepsilon(M)} \qquad (28)$$
>
> *where $\varepsilon(M) = \min\{\sum_i m_i \operatorname{tr}(\operatorname{Jac} Q(y)) : x_i = Q_i(y_i), H(x) \leq M\}$ is a positive constant.*

In particular, if applied to the entropic regularization setup of Example 1, Theorem 4 yields the following estimate.

**Corollary 4.** *Under* (SRD-EW)*, we have:*

$$\mathbb{E}_x[\tau_M] \lesssim M e^{\lambda M}/(\lambda \sigma_{\min}) \quad \text{for some } \lambda > 0. \qquad (29)$$

Theorem 4 shows that (S-FTRL) exhibits a markedly different behavior than its deterministic counterpart in harmonic games (and hence, in two-player zero-sum games with a fully mixed equilibrium): In the presence of noise, the energy $H(x)$ diverges to infinity on average, indicating in this way a mean drift towards the boundary bd($\mathcal{X}$) of $\mathcal{X}$.

From a technical standpoint, this involves using Dynkin's lemma on the energy function $H(x)$. More precisely, even though $H(x)$ is a constant of motion in the deterministic case, its Itô correction under (S-FTRL) leads to a constant positive drift, which only vanishes at the boundary of the game's strategy space. This implies that the limit $\lim_{t\to\infty} \mathbb{E}[H(X(t))]$ exists, and the rest of the proof follows by a contradiction argument in case this limit is assumed finite (this is where Dynkin's lemma and the infinitesimal generator of the process are evoked).

Moving forward, we should stress that Theorem 4 *does not* necessarily mean that the orbits of (S-FTRL) converge to the boundary with probability 1. Nonetheless, as we show below, (S-FTRL) reaches any neighborhood of the boundary in finite time on average, and takes infinite time to escape.

> **Theorem 5.** *Suppose Assumption 1 holds, let $\mathcal{K}$ be a compact subset of $\mathcal{X}$, and let $\tau_{\mathcal{K}} = \inf\{t \geq 0 : X(t) \notin \mathcal{K}\}$ be the time it takes $X(t)$ to escape $\mathcal{K}$ starting at $x \in \mathcal{K}$. Then, for any harmonic game, we have:*
> *1. $\mathbb{E}_x[\tau_{\mathcal{K}}] < \infty$ whenever $\mathcal{K}$ is disjoint from* bd($\mathcal{X}$)*.*
> *2. $\mathbb{E}_x[\tau_{\mathcal{K}}] = \infty$ whenever $\mathcal{K}$ contains* bd($\mathcal{X}$)*.*

While Theorem 5 states that the *average* return time to $\mathcal{K}$ is infinite, individual trajectories of (S-FTRL) may still reach it in finite time on a positive probability event. This distinction, based on return times to compact subsets, is known as the *transience/recurrence dichotomy* and, in this context, Theorem 5 implies that (S-FTRL) is *not* positive recurrent for a detailed discussion, cf. Appendix B.

The proof of Theorem 5, like that of Theorem 4, hinges on an application of Dynkin's lemma to the energy function $H(x)$—but this time, we require its bona fide, stopping time version. This involves lower- and upper-bounding the action of the generator of $X(t)$ on $H(x)$, and then integrating the result; we defer the relevant details to Appendix E.

In view of all this, it is natural to ask if at least *some* pure strategies are attracting. This is not the case:

**Theorem 6.** *Suppose Assumption 3 holds. If the game is harmonic, there is no proper face $\mathcal{S}$ of $\mathcal{X}$ that is stochastically asymptotically stable under* (S-FTRL)*.*

In words, this shows that (S-FTRL) is irreducible in harmonic games: even though orbits tend to drift toward the boundary in average, they do not collapse to any proper face thereof, so any stable attractor must span the entire strategy space of the game (cf. Fig. 1).

We should also stress here that the trajectories of (S-FTRL) do not necessarily converge to the boundary with probability 1; instead, they resemble a standard Brownian motion in this regard. This behavior stands in sharp contrast to (SRD-AS) and (SRD-PI) in zero-sum games, where orbits typically converge to the boundary with a nonzero drift, even under small perturbations [17, 24]. A detailed comparison of the dynamics' long-run behavior is provided in Appendix F.

## 6    Concluding remarks

Our aim in this paper was to quantify the impact of noise and uncertainty on the dynamics of regularized learning. Our findings can be summarized by the informal mantra that "uncertainty favors extremes", in the sense that the dynamics exhibit a tendency to drift toward the boundary of the players' strategy space: (*a*) every player reaches an almost pure strategy in finite time; (*b*) every player's limit set contains a pure strategy; (*c*) the only possible limits of the dynamics are pure Nash equilibria; and (*d*) game dynamics that are recurrent in the noiseless regime escape in expectation toward the boundary under uncertainty. We conjecture that there is an even stronger principle at play, namely that the only stochastically asymptotically stable limit sets of (S-FTRL) are entirely contained on the boundary of the game's strategy space; we pose this as an open problem for the community.

Several important directions remain open in the general context of stochastic FTRL dynamics. The first is what happens if the dynamics are run with a vanishing learning rate, as in [13]. In this case, we conjecture that an analogue of Theorem 2 continues to hold, but Theorem 4 fails: in games where the deterministic dynamics are recurrent (e.g., harmonic games), (S-FTRL) will most likely result in the sequence of play converging to some constant (Bregman) distance from a "central" equilibrium of the game (the game's strategic center in the case of harmonic games).

Another major extension of our work involves examining the impact of uncertainty in games with continuous action spaces—which, arguably, are more relevant for many applications of game theory to data science and machine learning—and, also, undertaking a discrete-time analysis that goes beyond the vanishing step-size considerations that are common in the stochastic approximation literature applied to learning in games [6, 8–10, 27, 37, 38, 47, 50]. Both of these directions would require significantly different tools and techniques (especially in the discrete-time setting), so we defer all this to the future.

## Acknowledgments

This research was supported in part by the French National Research Agency (ANR) in the framework of the PEPR IA FOUNDRY project (ANR-23-PEIA-0003), the "Investissements d'avenir" program (ANR-15-IDEX-02), and the LabEx PERSYVAL (ANR-11-LABX-0025-01), under grant IRGA-SPICE. PM is also a member of the Archimedes Research Unit/Athena RC, and was partially supported by project MIS 5154714 of the National Recovery and Resilience Plan Greece 2.0 funded by the European Union under the NextGenerationEU Program.

## Impact Statement

This paper presents work whose goal is to advance the field of Machine Learning. There are many potential societal consequences of our work, none of which we feel must be specifically highlighted here.

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

# A   Basic properties of regularizers, mirror maps, and Bregman/Fenchel divergences

In this appendix, we collect a number of standard properties concerning regularizers, their associated mirror maps, and Bregman/Fenchel distances; we closely follow [42, 47] for notation and conventions.

## A.1. Preliminary definitions.

*Remark.* All the constructions described in the following apply to each player of a finite game $\Gamma = \Gamma(\mathcal{N}, \mathcal{A}, u)$; to simplify notation, we suppress throughout the player index $i \in \mathcal{N}$.

Given a finite game, each player has effectively $n - 1$ degrees of freedom in the choice of a mixed strategy $x \in \mathcal{X}$, due to the constraint $\sum_{\alpha=1}^{n} x_\alpha = 1$. In light of this, we denote in the following by $\mathcal{V}$ the finite-dimensional vector space $\mathbb{R}^{n-1}$, equipped with the Euclidean norm $\|\cdot\|$, and denote with slight abuse of notation each player's *effective strategy set* as $\mathcal{X} = \{x \in \mathbb{R}^{n-1} : \sum_{\alpha=1}^{n-1} x_\alpha \leq 1\}$; with this convention, $\mathcal{X}$ is a closed convex subset of $\mathcal{V}$ with non-empty interior. We will furthermore write $\mathcal{Y} := \mathcal{V}^*$ for the dual space of $\mathcal{V}$, $\langle y, x \rangle$ for the canonical linear pairing between $x \in \mathcal{V}$ and $y \in \mathcal{V}^*$, and $\|y\|_* = \max\{\langle y, x \rangle : \|x\| \leq 1\}$ for the dual norm induced on $\mathcal{Y}$.

Following standard conventions in convex analysis, functions will take values in the extended real line $\mathbb{R} \cup \{\infty\}$, and a convex function $f : \mathcal{X} \to \mathbb{R}$ will be identified with the extended-real-valued function $\bar{f} : \mathcal{V} \to \mathbb{R} \cup \{\infty\}$ that agrees with $f$ on $\mathcal{X}$ and is identically equal to $\infty$ on $\mathcal{V} \setminus \mathcal{X}$; we will denote the *effective domain* of a convex function $f : \mathcal{V} \to \mathbb{R} \cup \{\infty\}$ on $\mathcal{V}$ as $\operatorname{dom} f := \{x \in \mathcal{V} : f(x) < \infty\}$.

For the scope of this work, a *regularizer on $\mathcal{X}$* is a convex function of Legendre type on $\operatorname{int}(\mathcal{X})$, namely a closed proper convex function $h : \mathcal{X} \to \mathbb{R} \cup \{\infty\}$ with the following properties:

(i) $\operatorname{int}(\mathcal{X}) = \operatorname{int}(\operatorname{dom} h)$;

(ii) *Steepness* – $h$ is smooth on $\operatorname{int}(\mathcal{X})$, and $\lim_{n \to \infty} \|\nabla h(x_n)\| \to \infty$ for any sequence $(x_n)_{n=1}^{\infty}$ in $\operatorname{int}(\mathcal{X})$ converging to a boundary point of $\operatorname{int}(\mathcal{X})$;

(iii) $h$ is $K$-strongly convex on $\operatorname{int}(\mathcal{X})$:

$$h(p) \geq h(x) + \langle \nabla h(x), p - x \rangle + \frac{K}{2} \|p - x\|^2 \quad \text{for all } p \in \mathcal{X}, x \in \operatorname{int}(\mathcal{X}). \tag{A.1}$$

Following Alvarez et al. [2, Proposition 4.10], a regularizer $h$ of Legendre type can be obtained as

$$h(x) = \sum_{\alpha=1}^{n-1} \theta(x_\alpha) + \theta\left(1 - \sum_{\alpha=1}^{n-1} x_\alpha\right) \quad \text{for all } x \in \mathcal{X}, \tag{A.2}$$

where $\theta : [0, \infty) \to \mathbb{R} \cup \{\infty\}$ is a convex function of Legendre type, called *kernel* of the regularizer, fulfilling

(i) $\theta(z) < \infty$ for all $z > 0$;

(ii) *Steepness* – $\theta$ is smooth on $(0, \infty)$, and $\lim_{z \to 0^+} \theta'(z) = -\infty$;

(iii) $\theta$ is strongly convex on $(0, \infty)$, namely $\inf_{z>0} \theta''(z) > 0$.

## A.2. Regularized best response map and Fenchel coupling.
In what follows, we will need the following fundamental objects:

1. The *convex conjugate $h^* : \mathcal{Y} \to \mathbb{R}$ of $h$*:

$$h^*(y) = \max_{x \in \mathcal{X}} \{\langle y, x \rangle - h(x)\} \qquad \text{for all } y \in \mathcal{Y}. \tag{A.3}$$

2. The *regularized best response map* – or *mirror map* – $Q : \mathcal{Y} \to \mathcal{X}$ induced by $h$:

$$Q(y) = \arg\max_{x \in \mathcal{X}} \{\langle y, x \rangle - h(x)\} \qquad \text{for all } y \in \mathcal{Y}. \tag{A.4}$$

3. The associated *Bregman distance $D : \mathcal{X} \times \operatorname{int}(\mathcal{X}) \to \mathbb{R}$ of $h$*:

$$D(p, x) = h(p) - h(x) - \langle \nabla h(x), p - x \rangle \qquad \text{for all } p \in \mathcal{X}, x \in \operatorname{int}(\mathcal{X}). \tag{A.5}$$

4. The *Fenchel coupling* $F: \mathcal{X} \times \mathcal{Y} \to \mathbb{R}$ of $h$:

$$F(p, y) = h(p) + h^*(y) - \langle y, p \rangle \qquad \text{for all } p \in \mathcal{X}, y \in \mathcal{Y}. \tag{A.6}$$

*Remark.* The terminology "Fenchel coupling" is due to [42].

The proposition below provides some basic properties concerning the first two objects above:

**Proposition A.1.** *Let $h$ be a regularizer on $\mathcal{X}$. Then:*

($a$) *The convex conjugate $h^*: \mathcal{Y} \to \mathbb{R}$ of $h$ is differentiable, and*

$$Q(y) = \nabla h^*(y) \quad \text{for all } y \in \mathcal{Y}. \tag{A.7}$$

($b$) *$\nabla h$ is one-to-one from $\mathrm{int}(\mathcal{X})$ to $\mathcal{Y}$, and $Q = (\nabla h)^{-1}$ namely*

$$x = Q(y) \iff y = \nabla h(x). \tag{A.8}$$

*for all $x \in \mathrm{int}(\mathcal{X})$ and $y \in \mathcal{Y}$.*

($c$) *$Q$ is $(1/K)$-Lipschitz continuous, that is,*

$$\|Q(y') - Q(y)\| \leq (1/K)\|y' - y\|_* \quad \text{for all } y, y' \in \mathcal{Y}. \tag{A.9}$$

*Proof.* Since these fact are relatively well-known, we only provide a specific pointer to the literature.

($a$) The equality $Q = \nabla h^*$ follows from Danskin's theorem, see e.g., Shalev-Shwartz [60, Section 2.7.2].

($b$) See Rockafellar [57, Theorem 26.5].

($c$) See Rockafellar & Wets [58, Theorem 12.60(b)]. ∎

**Lemma A.1.** *The trace of the Jacobian matrix of the mirror map* (A.4) *fulfills*

$$\mathrm{tr}\, \mathrm{Jac}\, Q(y) > 0 \quad \text{for all } y \in \mathcal{Y}. \tag{A.10}$$

*Proof.*

$$\mathrm{tr}\, \mathrm{Jac}\, Q_y = \mathrm{tr}\, \mathrm{Jac}\, \left[\nabla h^{-1}\right]_y \qquad \text{\% by (A.8)}$$

$$= \mathrm{tr}\, [\mathrm{Jac}\, \nabla h]^{-1}_{Q(y)} \tag{A.11}$$

$$= \mathrm{tr}\, [\mathrm{Hess}\, h]^{-1}_{Q(y)} \tag{A.12}$$

$$= \sum_{\alpha=1}^{n-1} \frac{1}{\mathrm{eig}_\alpha [\mathrm{Hess}\, h(Q(y))]}, \tag{A.13}$$

where $\mathrm{eig}_\alpha[\mathrm{Hess}\, h(Q(y))]$ is the $\alpha$-th eigenvalue of the Hessian matrix of $h$ at $Q(y)$. By strong (in particular, strict) convexity of $h$, $\mathrm{Hess}\, h(x)$ is symmetric an positive-definite for all $x \in \mathrm{int}(\mathcal{X}) = \mathrm{im}\, Q$, which concludes our proof. ∎

Next, we collect some basic properties of the Fenchel coupling.

**Proposition A.2.** *Let $h$ be a regularizer on $\mathcal{X}$. Then, for all $p \in \mathcal{X}$ and all $y, y' \in \mathcal{Y}$, we have:*

($a$) $F(p, y) \geq 0$ *with equality if and only if $p = Q(y)$.* $\qquad$ (A.14a)

($b$) $F(p, y) = D(p, Q(y))$. $\qquad$ (A.14b)

($c$) $F(p, y) \geq \frac{1}{2}K \|Q(y) - p\|^2$. $\qquad$ (A.14c)

*Proof.* ($a$) By the Fenchel–Young inequality, we have $h(p) + h^*(y) \geq \langle y, p \rangle$ for all $p \in \mathcal{X}, y \in \mathcal{Y}$, with equality if and only if $y = \nabla h(p)$, so our claim is immediate by Item ($b$) in Proposition A.1.

(b) Fix $p \in \mathcal{X}$ and $y \in \mathcal{Y}$, and let $x = Q(y)$; then, by the definition of $F$, we have

$$F(p, y) = h(p) + h^*(y) - \langle y, p \rangle \tag{A.15}$$
$$= h(p) + \langle y, x \rangle - h(x) - \langle y, p \rangle \qquad \text{\% since } x = Q(y)$$
$$= h(p) - h(x) - \langle \nabla h(x), p - x \rangle. \qquad \text{\% since } y = \nabla h(x)$$

(c) By the previous point and Eq. (A.1), Eq. (A.14c) follows immediately. ∎

*Remark.* In view of Proposition A.2, $D(p, x)$ can be seen as an anti-symmetric measure of divergence between $p \in \mathcal{X}$ and $x \in \text{int }\mathcal{X}$, and $F(p, y)$ as a "primal-dual" measure of divergence between $p \in \mathcal{X}$ and $y \in \mathcal{Y}$, with $F(p, y) = 0$ if and only if $Q(y) = p$, and $F(p, y) \to \infty$ whenever $Q(y) \to \text{bd}(\mathcal{X})$.

## B Elements of stochastic analysis

In this appendix, we review standard results from stochastic analysis and apply them to (S-FTRL) to classify the long-term behavior of its trajectories into two mutually exclusive categories: transience and recurrence.

**B.1. Infinitesimal generator and the transience/recurrence dichotomy.** Throughout this section, let us consider the time-homogeneous diffusion on $\mathbb{R}^n$ given by

$$dZ(t) = b(Z(t))\, dt + \Sigma(Z(t))dW(t) \tag{SDE}$$

where $W(t)$ is an $m$-dimensional standard Brownian motion, and $b \colon \mathbb{R}^n \to \mathbb{R}^n$, $\Sigma \colon \mathbb{R}^n \to \mathbb{R}^{n \times m}$ are Lipschitz continuous bounded functions. The *infinitesimal generator* associated to (SDE) is defined as the differential operator $\mathcal{L}$ acting on smooth functions $f \colon \mathbb{R}^n \to \mathbb{R}$ through

$$\mathcal{L}f(z) = \sum_{i=1}^{n} b_i(z) \frac{\partial f}{\partial z_i}(z) + \frac{1}{2} \sum_{i,j=1}^{n} a_{ij}(z) \frac{\partial^2 f}{\partial z_i \partial z_j}(z) \quad \text{for every } z \in \mathbb{R}^n, \tag{B.1}$$

where $a(z) := \Sigma(z)\Sigma(z)^T$ denotes the *principal symbol* of (SDE). Equivalently, $\mathcal{L}$ can also be characterized as the only operator such that the stochastic process $M_f(t)$ given by

$$M_f(t) := f(Z(t)) - f(Z(0)) - \int_0^t \mathcal{L}f(Z(s))\, ds \tag{B.2}$$

is a local martingale for every smooth function $f \colon \mathbb{R}^n \to \mathbb{R}$ (this follows from an application of Itô's formula). In practise, the representation from Eq. (B.2) is commonly used to estimate hitting times through what is often referred to as *Dynkin's lemma*:

**Lemma B.1** (Dynkin's lemma [54, Theorem 7.4.1]). *For every bounded stopping time $\tau$ and every smooth function $f \colon \mathbb{R}^n \to \mathbb{R}$,*

$$\mathbb{E}_z[f(Z(\tau))] = f(z) + \mathbb{E}_z\left[ \int_0^\tau \mathcal{L}f(Z(s))\, ds \right]. \tag{B.3}$$

Following classical notations, the infinitesimal generator $\mathcal{L}$ is said to be *uniformly elliptic* if there exists a constant $c > 0$ such that $u^T a(z)u \geq c\|u\|^2$ for every $u, y \in \mathbb{R}^n$. Intuitively, it means that the noise coefficient matrix is uniformly bounded away from zero or, in other words, that the noise does not vanish across the state space. More precisely, it also implies that every point of the space is explored with positive probability by trajectories of (SDE) (cf. Subsection 3.3.6.1 of [51], albeit in a much more general setting), in the sense that

$$\mathbb{P}_z(Z(t) = z' \text{ for some } t \geq 0) > 0 \quad \text{for every } z, z' \in \mathbb{R}^d. \tag{B.4}$$

Uniform ellipticity enables a full classification of the long-term behavior of (SDE) into two fundamental classes of processes: *recurrent* and *transient* processes. These behaviors are generally defined as follows:

**Definition 2.** Let $Z(t)$ be a solution orbit of (SDE).

- $Z(t)$ is said to be *recurrent* at a compact subset $\mathcal{K} \subseteq \mathbb{R}^n$ if the hitting time $\tau_\mathcal{K} = \inf\{t \geq 0 : Z(t) \in \mathcal{K}\}$ is almost-surely finite for some initial condition $z \in \mathbb{R}^n \setminus \mathcal{K}$. Moreover, if $\mathbb{E}_z[\tau_\mathcal{K}] < \infty$, then the process is further called *positive recurrent*, while it is called *null recurrent* if the expectation is infinite.

- $Z(t)$ is said to be *transient* from $z \in \mathbb{R}^n$ if it definitely exits every compact subsets in finite time when starting from $z$, i.e., if

$$\mathbb{P}_z(\text{there exists } t_0 < \infty \text{ such that } Z(t) \notin \mathcal{K} \text{ for every } t \geq t_0) = 1 \tag{B.5}$$

for every compact subset $\mathcal{K} \subseteq \mathbb{R}^n$.

Under uniform ellipticity, it can then be shown that recurrence and transience are the *only* behaviors that may happen in the long-run, and that they are perfect complementaries to one another:

**Lemma B.2** (Transience/recurrence dichotomy in $\mathbb{R}^n$ [11, 32]). *Let* (SDE) *be a diffusion on $\mathbb{R}^n$ with uniformly elliptic generator. Then,*

1. *If trajectories of* (SDE) *are positive recurrent (resp. null recurrent) for some compact subset $K \subseteq \mathbb{R}^n$ and initial condition $z \in \mathbb{R}^n$, then they are positive recurrent (resp. null recurrent) for every compact subsets and every initial conditions.*

2. *If trajectories of* (SDE) *are transient for some initial condition $z \in \mathbb{R}^n$, then they are transient for every initial conditions.*

3. *Trajectories of* (SDE) *are either all recurrent or all transient.*

*Remark* 1. The uniform ellipticity condition can be relaxed while preserving the transience/recurrence dichotomy and its significant implications. This is typically achieved through the use of *Hörmander conditions*, which require that a specific family of vector fields spans the entire space. However, these considerations go beyond the scope of this paper, and we instead refer interested readers to [7, 19, 51] for more concrete results on this topic.

**B.2. The transience/recurrence dichotomy in $\mathcal{X}$.** Instead of the Euclidean diffusion considered in Appendix B.1, we now turn our attention to the more intricate constrained diffusion (S-FTRL), which we recall is given by

$$\begin{aligned} dY_i(t) &= v_i(X(t))\, dt + \sigma_i(X) \cdot dW(t), \\ X_i &= Q_i(Y_i). \end{aligned} \tag{FTRL-S}$$

A natural question that arises is whether a similar dichotomy result to Lemma B.2 also holds for the player's choices within the (constrained) polyhedron $\mathcal{X}$. The goal of this subsection is to motivate and establish such a result.

One might be tempted to first consider the stochastic differential equation

$$dY_i = v_i(Q_i(Y)(t))\, dt + \sigma_i(Q_i(Y)) \cdot dW(t), \tag{B.6}$$

which is of the form given in Appendix B.1 with uniformly elliptic generator, and so should verify the transience / recurrence dichotomy. However, even if $Y(t)$ were itself recurrent (or transient), this would *not* necessarily imply that $X(t) = Q(Y(t))$ is also recurrent (or transient). Indeed, the inverse of the choice map $Q^{-1}$ is generally neither single-valued nor continuous, meaning that a compact subset of $\mathcal{X}$ does not necessarily remain compact in $\mathcal{Y}$ when mapped through $Q^{-1}$.

To bypass this issue, we will instead study the dynamics of (S-FTRL) in a third space $\mathcal{Z}$, sometimes referred to as the *space of payoff differences*.

Following the notations of Legacci et al. [34], let $\Gamma = \Gamma(\mathcal{N}, \mathcal{A}, u)$ be a given finite game, and let $\hat{\alpha}_i \in \mathcal{A}_i$ be a fixed benchmark strategy for each player $i \in \mathcal{N}$. We then consider the hyperplane $\mathcal{Z}_i = \{z_i \in \mathbb{R}^{n_i} : z_{i\hat{\alpha}_i} = 0\}$, which is evidently isomorphic to $\mathbb{R}^{n_i-1}$. Elements of each player's strategy space $\mathcal{Y}_i$ can then be mapped onto $\mathcal{Z}_i$ through the linear operator

$$\Pi_i \colon \mathcal{Y}_i \to \mathcal{Z}_i, \quad \Pi_i(y_i) = \{y_{i\alpha_i} - y_{i\hat{\alpha}_i} : \alpha_i \neq \hat{\alpha}_i.\} \tag{B.7}$$

The product space $\mathcal{Z} = \prod_{i \in \mathcal{N}} \mathcal{Z}_i$ is called the game's *payoff differences space*, and we denote by $\Pi$ the product projection map $\Pi = \prod_{i \in \mathcal{N}} \Pi_i$.

Importantly, if we define the payoff-adjusted choice map $\hat{Q} := \prod_{i \in \mathcal{N}} \hat{Q}_i \colon \mathcal{Z} \to \mathcal{X}$ by

$$\hat{Q}_i(z_i) = Q_i(y_i) \quad \text{for any } y_i \in \Pi_i^{-1}(z_i), \tag{B.8}$$

then we obtain the following regularity result:

**Lemma B.3.** *For every player $i \in \mathcal{N}$, strategy $x_i \in \mathcal{X}_i$ and action $\alpha_i \neq \hat{\alpha}_i$,*

$$z_{i\alpha_i} := \left[\hat{Q}_i^{-1}(x_i)\right]_{\alpha_i} = \theta_i'(x_{i\alpha_i}) - \theta_i'(x_{i\hat{\alpha}_i}). \tag{B.9}$$

*In particular, $\hat{Q}^{-1}\colon \operatorname{ri}\mathcal{X} \to \mathcal{Z}$ is a single-valued continuous map.*

*Proof.* The proof is immediate by the Karush-Kuhn-Tucker conditions applied to the convex problem posed in $Q$, stating that $y_{i\alpha_i} = \theta_i'(x_{i\alpha_i}) + \mu$ for some multiplier $\mu \in \mathbb{R}$, and the definition of $\hat{Q}$ through payoff differences. ∎

Consequently, *any* compact subset living in $\operatorname{ri}\mathcal{X}$ is mapped through $\hat{Q}^{-1}$ to a compact subset of $\mathcal{Z}$. Therefore, to establish a dichotomy result for stochastic dynamics within $\operatorname{ri}\mathcal{X}$, it suffices to demonstrate that the diffusion on $\mathcal{Z}$, obtained by mapping (S-FTRL) through $\Pi$, is uniformly elliptic.

Interpreting $\Pi$ as a full-rank matrix (which is possible because it is a linear function), it is easy to show that $Z(t) := \Pi Y(t)$ verifies the stochastic differential equation

$$dZ(t) = \Pi v(\hat{Q}(Z(t)))dt + \Pi \sigma(\hat{Q}(Z(t))) \cdot dW(t) \tag{FTRL-Z}$$

where $\sigma(x) = \left[\sigma_1(x) \cdots \sigma_N(x)\right]^T$ denotes the overarching diffusion matrix over all players.

Accordingly, the principal symbol of (FTRL-Z) is given by

$$a(z) := \Pi \sigma(\hat{Q}(z))\left[\Pi\sigma(\hat{Q}(z))\right]^T = \Pi\sigma(\hat{Q}(z))\sigma(\hat{Q}(z))^T \Pi^T \succcurlyeq \sigma_{\min}^2 \Pi\Pi^T \succcurlyeq \sigma_{\min}^2 \pi_{min}^2 I, \tag{B.10}$$

where $\sigma_{\min}$ denotes the smallest singular of $\sigma$, which is assumed to be positive (cf. Assumption 1), and $\pi_{min}$ denotes the smallest singular value of $\Pi$ (positive because $\Pi$ is full-rank). Consequently, the infinitesimal generator of (FTRL-Z) is uniformly elliptic, and thus Lemma B.2 holds true for its trajectories.

The arguments developed above naturally lead to the following definition of recurrence and transience for trajectories of (S-FTRL) evolving within the space $\operatorname{ri}\mathcal{X}$:

**Definition 3.** Let $X(t)$ be a solution orbit of (S-FTRL).

- $X(t)$ is said to be *recurrent in* $\operatorname{ri}\mathcal{X}$ if, for every relatively compact subsets $\mathcal{K} \subseteq \operatorname{ri}\mathcal{X}$ and every initial condition $x \in \operatorname{ri}\mathcal{X}$, the hitting time $\tau_{\mathcal{K}} = \inf\{t \geq 0 : X(t) \in \mathcal{K}\}$ is almost-surely finite. Moreover, if $\mathbb{E}_x[\tau_{\mathcal{K}}] < \infty$, then the process is further called *positive recurrent in* $\operatorname{ri}\mathcal{X}$, while it is called *null recurrent* is the expectation is infinite.

- $X(t)$ is said to be *transient in* $\operatorname{ri}\mathcal{X}$ if, for every initial condition $x \in \operatorname{ri}\mathcal{X}$, $X(t)$ converges to $\operatorname{bd}\mathcal{X}$ almost-surely.

Furthermore, Lemma B.2 can be applied to derive a transience/recurrence dichotomy for such trajectories:

**Theorem 7** (Transience/recurrence dichotomy in $\operatorname{ri}\mathcal{X}$). *Trajectories of* (S-FTRL) *are either all positive recurrent or all null recurrent or all transient in* $\operatorname{ri}\mathcal{X}$.

*Proof.* In virtue of Lemma B.3, any compact included in $\operatorname{ri}\mathcal{X}$ is mapped through $\hat{Q}^{-1}$ to a compact in $\mathcal{Z}$. Accordingly, $X(t)$ is recurrent (resp. transient) in $\operatorname{ri}\mathcal{X}$ whenever $Z(t)$ is recurrent (resp. transient) in $\mathcal{Z}$. But the stochastic differential equation (FTRL-Z) defining $Z(t)$ is uniformly elliptic, so either all trajectories of (FTRL-Z) are recurrent or all trajectories are transient (cf. Lemma B.2). Consequently, trajectories of (S-FTRL) are also either all recurrent or all transient in $\operatorname{ri}\mathcal{X}$. The same logic can also be adapted to show that either all trajectories are positive recurrent or all are null recurrent. ∎

**B.3. Additional standard results.** We conclude this appendix by reviewing several results from stochastic analysis, which are typically stated in classical textbooks for the specific case of Brownian motion, but that are more difficult to find in their full generality:

**Lemma B.4** (Law of iterated logarithm for martingales [36]). *Let $M(t)$ be a continuous square-integrable local martingale. Then, the trajectories of $M(t)$ verify*

$$\limsup_{t \to \infty} \frac{M(t)}{\sqrt{[M](t) \log\log[M](t)}} = 1 \tag{B.11}$$

*on the event $\{\lim_{t \to \infty}[M](t) = \infty\}$.*

**Lemma B.5** (Strong law of large numbers [39])**.** *Let $M(t)$ be a continuous square-integrable local martingale and let us define*

$$\rho_M(t) = \int_0^t \frac{d[M](s)}{(1+s)^2}. \tag{B.12}$$

*If $\lim_{t\to\infty} \rho_M(t) < \infty$ (a.s.), then $M(t)/t \to 0$ (a.s.).*

# C Omitted proofs from Section 3

In this appendix, we provide the proofs from Section 3 that were omitted in the main text. Specifically, we demonstrate that the stochastic differential equation defining (S-FTRL) admits a unique strong solution, and we present an explicit computation of the stochastic differential equation governing the evolution of the player's strategies in $\mathcal{X}$.

We recall that the stochastic dynamics under study is given by the stochastic differential equation

$$
\begin{aligned}
dY_i(t) &= v_i(X(t))\,dt + dM_i(t) \\
X_i(t) &= Q_i(Y_i(t)),
\end{aligned}
\tag{FTRL-S}
$$

where $M_i = \sigma_i(X(t))dW(t)$ for some underlying $m$-dimensional Brownian motion.

**Proposition C.1.** *The stochastic differential equation* (S-FTRL) *admits a unique strong solution $Y(t)$ for every initial condition $Y(0) \in \mathcal{Y}$.*

*Proof.* The existence and uniqueness of a strong solution can be proved using a classical Picard's iteration argument (see e.g., Øksendal [54, Theorem 5.2.1]). To do so, we only need to show that the drift and diffusion coefficients verify a mild growth condition and that they are Lipschitz continuous on $\mathcal{Y}$. First, notice that $Q$ is $(1/K)$-Lipschitz on $\mathcal{Y}$ where $K > 0$ denotes the smallest strong convexity constant among the regularizer functions $h_i$, $i \in \mathcal{N}$ (cf. Proposition A.1). As such, the composition of $v$ (resp. $\sigma$) with $Q$ remains Lipschitz-continuous. Furthermore, as $Q$ takes values into the compact set $\mathcal{X}$, it is immediate that the growth condition is verified (the coefficients are in fact all uniformly bounded on the state space). The stochastic dynamics (S-FTRL) therefore admits a unique strong solution as required. ∎

*Remark* 2. The uniqueness mentioned in Proposition C.1 should be understood in the sense of *equivalence up to evanescence*, meaning that any two solutions of (S-FTRL) must be equal at any time with probability 1; i.e., they should be indistinguishable from one another as stochastic processes.

**Proposition 1.** *The solutions of* (S-FTRL) *satisfy the stochastic differential equation*

$$dX_{i\alpha_i} = g_{i\alpha_i}\left[v_{i\alpha_i} - \sum_{\beta_i \in \mathcal{A}_i} \chi_{i\beta_i} v_{i\beta_i}\right] dt \tag{14a}$$

$$+ g_{i\alpha_i}\left[dM_{i\alpha_i} - \sum_{\beta_i \in \mathcal{A}_i} \chi_{i\beta_i}\, dM_{i\beta_i}\right] \tag{14b}$$

$$+ g_{i\alpha_i} \sum_{k=1}^m \left[\psi_{i\alpha_i k}^2 - \sum_{\beta_i \in \mathcal{A}_i} \chi_{i\beta_i} \psi_{i\beta_i k}^2\right] dt \tag{14c}$$

*where we set $g_{i\alpha_i} = 1/\theta_i''(x_{i\alpha_i})$, $\chi_{i\alpha_i} = g_{i\alpha_i} / \sum_{\beta_i \in \mathcal{A}_i} g_{i\beta_i}$, and $\psi_{i\alpha_i k}^2 = -\frac{1}{2}\theta_i'''(x_{i\alpha_i})g_{i\alpha_i}^2\left[\sigma_{i\alpha_i k} - \sum_{\beta_i \in \mathcal{A}_i} \chi_{i\beta_i}\sigma_{i\beta_i k}\right]^2$.*

*Proof of Proposition 1.* For simplicity (and without loss of generality), let us suppress the player's index $i \in \mathcal{N}$ in the remaining of the proof. Due to $h$ being assumed steep and decomposable as $h(x) = \sum_\alpha \theta(x_\alpha)$ (see Appendix A for the collection of assumptions verified by $\theta$), the Karush-Kuhn-Tucker conditions applied to the convex optimization problem of Eq. (6) posed to define $Q$ leads to $y_\alpha = \theta'(x_\alpha) + \lambda$ for some Lagrange multiplier $\lambda \in \mathbb{R}$ whenever $x = Q(y)$. In particular, applying this identity to the solution orbits $Y(t)$ and $X(t)$ of (S-FTRL) yields the existence of a continuous stochastic process $\lambda(t)$ such that

$$Y_\alpha(t) = \theta'(X_\alpha(t)) + \lambda(t) \quad \text{for every } t \geq 0. \tag{C.1}$$

Applying the classical Itô's formula (see e.g., Section 3.3 of Karatzas & Shreve [29]) on both sides of the equality therefore gives

$$d\lambda = dY_\alpha - \theta''(X_\alpha)dX_\alpha - \frac{1}{2}\theta'''(X_\alpha)d[X_\alpha], \tag{C.2}$$

where $[X_\alpha](t)$ denotes the quadratic variation process of $X(t)$. Before proceeding with the proof, let us introduce some notation that will be useful for simplifying the subsequent computations:

$$\theta''_\alpha := \theta''(X_\alpha), \quad \theta'''_\alpha := \theta''(X_\alpha); \tag{C.3}$$

$$g_\alpha := 1/\theta''_\alpha, \quad G := \sum_\beta g_\beta, \quad \chi_\alpha := g_\alpha/G. \tag{C.4}$$

If we rearrange the terms of Eq. (C.2) and take the sum over all $\alpha \in \mathcal{A}$, we get

$$0 = \sum_\alpha dX_\alpha = \sum_\alpha g_\alpha dY_\alpha - \frac{1}{2}\sum_\alpha g_\alpha \theta'''_\alpha d[X_\alpha] - G d\lambda, \tag{C.5}$$

where the first equality comes from the simplex constraint $\sum_\alpha X_\alpha = 1$. Consequently, $\lambda(t)$ is given explicitly by

$$d\lambda = \sum_\alpha \chi_\alpha dY_\alpha - \frac{1}{2}\sum_\alpha \chi_\alpha \theta'''_\alpha d[X_\alpha]. \tag{C.6}$$

By inserting this expression into Eq. (C.2), we therefore get

$$dX_\alpha = g_\alpha\left(dY_\alpha - \sum_\beta \chi_\beta dY_\beta\right) - \frac{1}{2}g_\alpha\left(\theta'''_\alpha d[X_\alpha] - \sum_\beta \chi_\beta \theta'''_\beta d[X_\beta]\right) \tag{C.7}$$

$$= g_\alpha\left(v_\alpha - \sum_\beta \chi_\beta v_\beta\right)dt + g_\alpha \sum_k\left(\sigma_{\alpha k} - \sum_\beta \chi_\beta \sigma_{\beta k}\right)dW_k - \frac{1}{2}g_\alpha\left(\theta'''_\alpha d[X_\alpha] - \sum_\beta \chi_\beta \theta'''_\beta d[X_\beta]\right). \tag{C.8}$$

The quadratic variation of $X_\alpha(t)$ can thus be computed as

$$d[X_\alpha] = g_\alpha^2 \sum_{kl}\left(\sigma_{\alpha k} - \sum_\beta \chi_\beta \sigma_{\beta k}\right)\left(\sigma_{\alpha l} - \sum_\beta \chi_\beta \sigma_{\beta l}\right)d[W_l, W_l] = g_\alpha^2 \sum_k\left(\sigma_{\alpha k} - \sum_\beta \chi_\beta \sigma_{\beta k}\right)^2 dt, \tag{C.9}$$

which ultimately proves the stochastic differential equation stated in Proposition 1 upon substituting $d[X_\alpha]$ with its expression in Eq. (C.8). ∎

## D   Omitted proofs from Section 4

In this appendix, we collect the proofs of the results presented in Section 4.

**D.1.  Proof of Theorem 1.**   Our aim in this appendix is to prove Theorem 1, which we restate below for convenience.

**Theorem 1.** *Suppose Assumptions 1 and 2 hold, fix a sufficiently small accuracy threshold $\varepsilon > 0$, and let*

$$\tau_{i,\varepsilon} = \inf\{t \geq 0 : \max_{\alpha_i \in \mathcal{A}_i} X_{i\alpha_i}(t) \geq 1 - \varepsilon\} \tag{19}$$

*denote the time player $i \in \mathcal{N}$ takes to reach an $\varepsilon$-neighborhood of one of their pure strategies. Then $\tau_{i,\varepsilon}$ is finite with probability $1$, and we have*

$$\mathbb{E}_x[\tau_{i,\varepsilon}] \lesssim e^\lambda/\lambda \tag{20}$$

*where $\lambda > 0$ is a positive constant that scales as*

$$\lambda = \Theta\left(\frac{1 + \sigma_{i,\max}^2}{\sigma_{i,\min}^2}\theta_i''\left(\frac{\varepsilon}{n_i - 1}\right)^2\right). \tag{21}$$

To facilitate a more modular proof of Theorem 1, we reformulate its results as two distinct propositions detailed below:

**Proposition D.1.** *The hitting time $\tau_{i,\varepsilon}$ is finite with probability $1$.*

**Proposition D.2.** *The expectation of $\tau_{i,\varepsilon}$ is upper-bounded by*

$$\mathbb{E}_x[\tau_{i,\delta}] \lesssim e^\lambda/\lambda \tag{D.1}$$

*where $\lambda > 0$ is a positive constant that scales as*

$$\lambda = \Theta\left(\frac{1 + \sigma_{i,\max}^2}{\sigma_{i,\min}^2} \theta_i''\left(\frac{\delta}{n_i - 1}\right)^2\right). \tag{D.2}$$

Before proving any of these results, let us restate the stochastic differential equation defining the evolution of $X(t)$ (cf. Proposition 1) as

$$dX_{i\alpha_i} = \sum_{\beta_i} g_{i\alpha_i}\left[\delta_{\alpha_i\beta_i} - G_i^{-1}g_{i\beta_i}\right](v_{i\beta_i} + S_{i\beta_i})dt + \sum_{\beta_i} g_{i\alpha_i}\left[\delta_{\alpha_i\beta_i} - G_i^{-1}g_{i\beta_i}\right]dM_{i\beta_i}, \tag{D.3}$$

where:

a) $g_{i\alpha_i} = 1/\theta_i''(x_{i\alpha_i})$;

b) $G_i = \sum_\beta g_{i\beta_i}$;

c) $S_{i\alpha_i} = -\frac{1}{2}\frac{\theta_i'''(x_{i\alpha_i})}{[\theta_i''(x_{i\alpha_i})]^2}\sum_{k=1}^m\left[\sigma_{i\alpha_i k} - \sum_{\beta_i} G_i^{-1}g_{i\beta_i}\sigma_{i\beta_i k}\right]^2$.

Borrowing the idea from Imhof [28] – who has studied the same type of property for the replicator dynamics with aggregate shocks (SRD-AS) in symmetric games – we aim to construct a well-behaved Lyapunov function $\phi_i$ for the collection $\mathcal{A}_i$ of pure strategies of player $i \in \mathcal{N}$. More precisely, such a function should verify the following properties:

1. $\phi_i(x) \geq 0$ for every $x \in \mathcal{X}$, with equality reached only when $x_i$ is a pure strategy;

2. Away from pure strategies of $\mathcal{X}_i$, $\phi_i$ decreases in average along trajectories of (S-FTRL).

Intuitively, the existence of such a function would suggest that the solution orbits of (S-FTRL) converge on average toward a neighborhood of pure strategies in $\mathcal{X}_i$, which in turn would imply that the time required to reach such a neighborhood is finite.

For the sake of readability, let us drop completely the reference to the player's index $i$ in the remainder of this appendix. This can be done without any loss of generality, as we are focusing solely on the hitting time of a single player.

Accordingly, let us propose $\phi$ of the parametric form

$$\phi(x) = n - 1 + e^\lambda - \sum_\alpha e^{\lambda x_\alpha} \tag{D.4}$$

with $\lambda > 0$ a positive parameter that will be chosen afterwards (depending on the parameters of interest of the problem).

**Claim 1.** $\phi(x) \geq 0$ *with equality reached only when $x$ is a pure strategy.*

*Proof.* This claim follows from the inequality $e^a + e^b \leq e^{a+b} + 1$ holding for every $a, b \geq 0$ (this is easily shown by fixing $b$ and taking derivative with respect to $a$). Indeed, iterating this inequality on every elements $\lambda x_\alpha \geq 0$, we get

$$\sum_\alpha e^{\lambda x_\alpha} \leq e^{\lambda \sum_\alpha x_\alpha} + n - 1 = e^\lambda + n - 1, \tag{D.5}$$

where the right-hand side is reached exclusively for pure strategies. Consequently, we readily obtain

$$\phi(x) \geq n - 1 + e^\lambda - e^\lambda - (n - 1) = 0 \tag{D.6}$$

as required. ∎

**Claim 2.** *Away from pure strategies, $\phi$ decreases in average along trajectories of* (S-FTRL).

*Proof.* More precisely, we will show that outside the neighborhood $\mathcal{U}_\varepsilon = \{x \in \mathcal{X} : x_\alpha > 1 - \varepsilon \text{ for some } \alpha \in \mathcal{A}\}$, there exists a parameter $\lambda$ big enough such that

$$\frac{d}{dt}\mathbb{E}_x[\phi(X(t))] =: \mathcal{L}\phi(x) < 0 \quad \text{for every } x \in \mathcal{X} \setminus \mathcal{U}_\varepsilon. \tag{D.7}$$

To do so, let us first notice that $\phi \in C^2(\mathcal{X})$ and that its partial derivatives are given by

$$\frac{\partial \phi}{\partial x_\alpha} = -\lambda e^{\lambda x_\alpha} \quad \text{and} \quad \frac{\partial^2 \phi}{\partial x_\alpha \partial x_\beta} = -\lambda^2 e^{\lambda x_\alpha}\delta_{\alpha\beta}. \tag{D.8}$$

As such, Itô's lemma and the stochastic differential equation from Eq. (D.3) can be used to obtain

$$\mathcal{L}\phi(x) = -\lambda \underbrace{\sum_{\alpha\beta} g_\alpha[\delta_{\alpha\beta} - G^{-1}g_\beta](v_\beta + S_\beta)e^{\lambda x_\alpha}}_{(I)} - \frac{\lambda^2}{2}\underbrace{\sum_\alpha g_\alpha^2 e^{\lambda x_\alpha}\sum_{\beta\gamma}[\delta_{\alpha\beta} - G^{-1}g_\beta][\delta_{\alpha\gamma} - G^{-1}g_\gamma]\Sigma_{\beta\gamma}}_{(II)}, \tag{D.9}$$

where we recall that $\Sigma = \sigma\sigma^T$ denotes the quadratic covariation matrix of noise (for player $i$).

- Lower bound for $(I)$: From the definition of $g_\alpha$ and $G$, and the fact $\theta$ is strongly convex and steep, it is evident that $0 \leq G^{-1}g_\alpha \leq 1$ for every $\alpha \in \mathcal{A}$. Furthermore, by Assumption 2, we have $|\theta'''(x_\alpha)/[\theta''(x_\alpha)]^2| \leq M$ for some constant $M < \infty$. As such, we can already bound the absolute value of $S_\alpha$ by

$$|S_\alpha| \leq \frac{M}{2}\sum_{k=1}^m\left[\sigma_{\alpha k}^2 + \left(\sum_\beta G^{-1}g_\beta\sigma_{\beta k}\right)^2\right] \leq \frac{M}{2}\sum_{k=1}^m(\sigma_{\max}^2 + \sigma_{\max}^2) = mM\sigma_{\max}^2 \tag{D.10}$$

where we recall that $\sigma_{\max}^2$ denotes the maximal eigenvalue of $\Sigma$ uniformly on $x \in \mathcal{X}$. Using this bound in tandem with $|v_\alpha(x)| \leq M_v < \infty$ for every $\alpha \in \mathcal{A}$, $x \in \mathcal{X}$ (holding from continuity of $v_\alpha$ on the compact $\mathcal{X}$), then leads to

$$(I) \geq -\sum_\alpha n\left(M_v + mM\sigma_{\max}^2\right)g_\alpha e^{\lambda x_\alpha} = -\frac{B}{2}\sum_\alpha g_\alpha e^{\lambda x_\alpha} \tag{D.11}$$

where $B = 2n(M_v + mM\sigma_{\max}^2)$.

- Lower bound for $(II)$: Notice that $\Sigma \succeq \sigma_{\min}^2 I$ with $\sigma_{\min} > 0$ by Assumption 1, which allows us to bound $(II)$ as

$$(II) \geq \sum_\alpha g_\alpha^2 e^{\lambda x_\alpha}\sigma_{\min}^2\sum_\beta[\delta_{\alpha\beta} - G^{-1}g_\beta]^2 = \sigma_{\min}^2\left\{\sum_\alpha g_\alpha^2(1 - G^{-1}g_\alpha)^2 e^{\lambda x_\alpha} + \sum_{\alpha\neq\beta} g_\alpha^2[G^{-1}g_\beta]^2 e^{\lambda x_\alpha}\right\} \tag{D.12}$$

$$\geq \sigma_{\min}^2\sum_\alpha g_\alpha^2(1 - G^{-1}g_\alpha)^2 e^{\lambda x_\alpha}. \tag{D.13}$$

Combining the bounds for both $(I)$ and $(II)$ therefore leads to

$$\mathcal{L}\phi(x) \leq \frac{\lambda}{2}\sum_\alpha g_\alpha e^{\lambda x_\alpha}\left[B - \lambda\sigma_{\min}^2 g_\alpha(1 - G^{-1}g_\alpha)^2\right]. \tag{D.14}$$

Now, let us consider the region of interest $R_\varepsilon = \mathcal{X} \setminus \mathcal{U}_\varepsilon$. Any $x \in R_\varepsilon$ thus verifies $x_\alpha \leq 1 - \varepsilon$ for every $\alpha \in \mathcal{A}$. Furthermore, notice that there always exists at least one action $\alpha \in \mathcal{A}$ such that $x_\alpha \geq 1/n$.

Accordingly, let us the decompose the set of actions as $\mathcal{A}_+(x) = \{\alpha \in \mathcal{A} : x_\alpha \geq 1/n\}$ and $\mathcal{A}_-(x) = \mathcal{A} \setminus \mathcal{A}_+(x)$. We also define the following quantities, which will be useful in establishing the bound:

a) $H_{max} = \max_{x \leq 1/n} 1/\theta''(x) < \infty$ and $H_{min} = \min_{x \geq 1/n} 1/\theta''(x) > 0$;

b) $c_\varepsilon = \min\{g_\alpha(x)(1 - G^{-1}(x)g_\alpha(x))^2 : \alpha \in \mathcal{A}_+(x), x \in R_\varepsilon\} > 0$.

where the positiveness of both $H_{min}$ and $c_\varepsilon$ comes from the fact that $g_\alpha(x) = 0$ only when $x_\alpha = 0$ (consequence of the regularizer being steep).

Using these notations we can then write, for any $x \in R_\varepsilon$,

$$\mathcal{L}\phi(x) \leq \frac{\lambda}{2}\left\{B\sum_{\alpha \in \mathcal{A}_-} g_\alpha e^{\lambda x_\alpha} + (B - \lambda\sigma_{\min}^2 c_\varepsilon)\sum_{\alpha \in \mathcal{A}_+} g_\alpha e^{\lambda x_\alpha}\right\} \tag{D.15}$$

$$\leq \frac{\lambda}{2}\left\{H_{max}B(n-1)e^{\lambda/n} - \sigma_{\min}^2 c_{\varepsilon}\left(\lambda - \frac{B}{\sigma_{\min}^2 c_{\varepsilon}}\right)\sum_{\alpha \in \mathcal{A}_+} g_{\alpha}e^{\lambda x_{\alpha}}\right\}. \tag{D.16}$$

Consequently, if we choose $\lambda$ big enough so that

$$\lambda \geq \frac{1}{\sigma_{\min}^2 c_{\varepsilon}}\left[B + \frac{H_{max}}{H_{min}}(n-1)B + 1\right], \tag{D.17}$$

then we readily get

$$\mathcal{L}\phi(x) \leq \frac{\lambda}{2}\left\{H_{max}B(n-1)e^{\lambda/n} - \left(\frac{H_{max}}{H_{min}}(n-1)B + 1\right)\sum_{\alpha \in \mathcal{A}_+} g_{\alpha}e^{\lambda x_{\alpha}}\right\} \tag{D.18}$$

$$\leq \frac{\lambda}{2}\left\{H_{max}B(n-1)e^{\lambda/n} - \frac{H_{max}}{H_{min}}(n-1)BH_{min}e^{\lambda/n} - H_{min}e^{\lambda/n}\right\} \tag{D.19}$$

$$= -\frac{\lambda}{2}H_{min}e^{\lambda/n} < 0 \tag{D.20}$$

for every $x \in R_{\varepsilon}$, which verifies the claim. ∎

Building on the two previous claims, we are now ready to prove Proposition D.1 and Proposition D.2.

*Proof of Proposition D.1.* Let $\tau_{\varepsilon} = \inf\{t \geq 0 : X_{\alpha}(t) \geq 1 - \varepsilon \text{ for some } \alpha \in \mathcal{A}\}$. For every fixed $t \geq 0$, Dynkin's formula yields

$$\mathbb{E}_x[\phi(X(\tau_{\varepsilon} \wedge t))] = \phi(x) + \mathbb{E}_x\left[\int_0^{\tau_{\varepsilon} \wedge t} \mathcal{L}\phi(X(s))\, ds\right]. \tag{D.21}$$

Claims 1 and 2 can then be used to argue that there exists a (deterministic) parameter $\lambda$ big enough so that

$$0 \leq n + e^{\lambda} - \frac{\lambda}{2}H_{min}e^{\lambda/n}\,\mathbb{E}_x[\tau_{\varepsilon} \wedge t] \tag{D.22}$$

for every $t \geq 0$. Rearranging the terms and taking the monotone limit as $t \nearrow \infty$ therefore give

$$\mathbb{E}_x[\tau_{\varepsilon}] \leq \frac{2}{\lambda}\frac{e^{\lambda} + n}{H_{min}e^{\lambda/n}} < \infty \tag{D.23}$$

for every initial condition $x \in \mathcal{X}$, which finishes the proof. ∎

*Proof of Proposition D.2.* Based on Eq. (D.23), it is evident that the hitting time $\tau_{\varepsilon}$ has an average bounded as

$$\mathbb{E}_x[\tau_{\varepsilon}] \lesssim \frac{e^{\lambda}}{\lambda} \tag{D.24}$$

for $\lambda$ big enough. Furthermore, due to the condition from Eq. (D.17), $\lambda$ can also be chosen proportional to $\frac{\sigma_{max}^2 + 1}{\sigma_{\min}^2 c_{\varepsilon}}$ (where the hidden multiplicative constants depend neither on noise nor on $\varepsilon$).

To prove Proposition D.2, we therefore only need to show that $c_{\varepsilon}$ is of order $[\theta''(\varepsilon/(n-1))]^{-2}$ for $\varepsilon$ small enough.

Recall that $c_{\varepsilon}$ is defined through the minimization problem

$$c_{\varepsilon} = \min\{g_{\alpha}(x)(1 - G^{-1}(x)g_{\alpha}(x))^2 : \alpha \in \mathcal{A}_+(x), x \in R_{\varepsilon}\}, \tag{D.25}$$

where $R_{\varepsilon} = \{x \in \mathcal{X} : x_{\alpha} \leq 1 - \varepsilon \text{ for every } \alpha \in \mathcal{A}\}$ and $\mathcal{A}_+(x) = \{\alpha \in \mathcal{A} : x_{\alpha} \geq 1/n\}$. For the sake of clarity, let us also define

$$c(x, \alpha) = g_{\alpha}(x)[1 - G^{-1}(x)g_{\alpha}(x)]^2 \quad \text{and} \quad c(x) = \min_{\alpha \in \mathcal{A}_+(x)} c(x, \alpha), \tag{D.26}$$

so that $c_{\varepsilon} = \min_{x \in R_{\varepsilon}} c(x)$ for any $\varepsilon > 0$.

The main idea of the proof is the following observation: $c(x) \geq 0$ for every $x \in \mathcal{X}$, with equality occurring if and only if $x$ is a pure strategy (this is a consequence of $g_\alpha(x) = 0$ if and only if $x_\alpha = 0$). Accordingly, for $\varepsilon$ small enough, we should expect $c_\varepsilon$ to be reached for an $x \in R_\varepsilon$ that is the closest possible to a pure strategy.

To make this argument rigorous, note that by definition $\theta''$ is locally decreasing around 0, and so there exists an $\eta < 1/n$ small enough verifying $\theta''(x) \geq \theta''(y)$ for every $x \leq y \leq \eta$. The proof is then separated into two parts: for $\varepsilon < \eta$ small enough, we decompose the minimization space $R_\varepsilon$ as $R_\eta$ and $R_\varepsilon \setminus R_\eta$, and find a lower-bound for both spaces independently.

Before jumping into those steps, notice that $c(x, \alpha)$ can be rewritten as

$$\frac{g_\alpha(x)}{G(x)^2} \Big( \sum_{\alpha \neq \beta} g_\beta(x) \Big)^2, \tag{D.27}$$

where the term $g_\alpha(x) G^{-2}(x)$ is lower-bounded uniformly on $\alpha \in \mathcal{A}_+(x)$ and $x \in R_\varepsilon$.

*Step 1: Minimization on $R_\eta$.* As $c(x) \to 0$ when $x$ converges to a pure strategy, there exists an $\varepsilon' < \eta$ small enough such that, for any $\varepsilon < \varepsilon'$, $c(x) > c(x^*)$ whenever $x \in R_\eta$ and $x^* \in \mathcal{U}_\varepsilon$. In particular, for $x^* = (1 - \varepsilon)e_\alpha + \frac{\varepsilon}{n-1} \sum_{\beta \neq \alpha} e_\beta$, it implies that

$$\min_{x \in R_\eta} c(x) > c(x^*) \gtrsim \Big( \sum_{\alpha \neq \beta} [\theta''(x_\beta^*)]^{-1} \Big)^2 \propto \Big[ \theta'' \Big( \frac{\varepsilon}{A-1} \Big) \Big]^{-2} \tag{D.28}$$

by definition of $x^*$.

*Step 2: Minimization on $R_\varepsilon \setminus R_\eta$.* Let $x \in R_\varepsilon \setminus R_\eta$, i.e., $x_\alpha \leq 1 - \varepsilon$ for every $\alpha \in \mathcal{A}$ and there exists a $\beta \in \mathcal{A}$ such that $x_\beta > 1 - \eta$. In particular, we know that $x_\beta \leq 1 - \varepsilon$, so there exists $\gamma' \neq \beta$ verifying

$$\frac{\varepsilon}{n-1} \leq x_{\gamma'} \leq \eta. \tag{D.29}$$

Furthermore, $\eta$ was initially taken so that $\eta < 1/n$, thus $\mathcal{A}_+(x) = \{\beta\}$, which leads to

$$c(x) = c(x, \beta) \gtrsim \Big( \sum_{\gamma \neq \beta} [\theta''(x_\gamma)]^{-1} \Big)^2 \geq [\theta''(x_{\gamma'})]^{-2} \geq \Big[ \theta'' \Big( \frac{\varepsilon}{A-1} \Big) \Big]^{-2}, \tag{D.30}$$

where the last inequality comes from the local monotony of $\theta''$ for $\frac{\varepsilon}{n-1} \leq x_{\gamma'} \leq \eta$.

Combining those two steps, we conclude that

$$c_\varepsilon = \min_{x \in R_\varepsilon} c(x) = \min \Big\{ \min_{x \in R_\eta} c(x), \min_{x \in R_\varepsilon \setminus R_\eta} c(x) \Big\} \gtrsim \Big[ \theta'' \Big( \frac{\varepsilon}{n-1} \Big) \Big]^{-2}. \tag{D.31}$$

On the other hand, taking $x = (1 - \varepsilon)e_\alpha + \frac{\varepsilon}{n-1} \sum_{\beta \neq \alpha} e_\beta \in R_\varepsilon$ leads to

$$c_\varepsilon \leq c(x, \alpha) = \frac{1}{\theta''(1-\varepsilon)} \Big[ \frac{1}{\theta''(1-\varepsilon)} + \frac{n-1}{\theta''(\varepsilon/(n-1))} \Big]^{-2} (n-1)^2 \Big[ \theta'' \Big( \frac{\varepsilon}{n-1} \Big) \Big]^{-2} \tag{D.32}$$

$$\leq (A-1)^2 \theta''(1-\varepsilon) \Big[ \theta'' \Big( \frac{\varepsilon}{n-1} \Big) \Big]^{-2} \tag{D.33}$$

$$\lesssim \Big[ \theta'' \Big( \frac{\varepsilon}{n-1} \Big) \Big]^{-2}, \tag{D.34}$$

which finally shows that $c_\varepsilon = \Theta\Big( \Big[ \theta'' \Big( \frac{\varepsilon}{n-1} \Big) \Big]^{-2} \Big)$ as needed. ∎

**D.2. Proof of Theorem 2.** Our goal in this appendix is to prove the following theorem:

**Theorem 2.** *Suppose Assumption 3 holds. Then, with notation as above, $\mathcal{S}$ is stochastically asymptotically stable under (S-FTRL) if and only if it is closed under better replies. If this is so, there exists an open initialization domain $\mathcal{U}_0 \subseteq \mathcal{X}$ such that, whenever $X(0) \in \mathcal{U}_0$, we have with arbitrarily high probability*

$$\mathrm{dist}_1(X(t), \mathcal{S}) \lesssim \Phi\Big[ c_1 - c_2 t + \mathcal{O}\Big( \sigma_{\max} \sqrt{t \log \log t} \Big) \Big] \tag{25}$$

*where $c_1$ and $c_2$ are constants ($c_2 > 0$), and the rate function $\Phi$ is given by $\Phi(z) = \max_{i \in \mathcal{N}} (\theta_i')^{-1}(z)$.*

Let $\mathcal{B} = \prod_{i \in \mathcal{N}} \mathcal{B}_i$ be a product of pure strategies, and let $\mathcal{S}$ be the face spanned by $\mathcal{B}$. Recall from Appendix A that, for every player $i \in \mathcal{N}$, the Bregman divergence is defined as

$$D_i(p_i, x_i) = h_i(p_i) - h_i(x_i) - \langle \nabla h_i(x_i), p_i - x_i \rangle \quad \text{for all } p_i \in \mathcal{X}_i, x_i \in \text{ri } \mathcal{X}_i. \tag{D.35}$$

For each player $i \in \mathcal{N}$, let $\mathcal{C}_i = \mathcal{A}_i \setminus \mathcal{B}_i$ denote the collection of pure actions not included in $\mathcal{B}_i$. The main idea behind proving the stability of closed under better replies faces is to show that the family of energy functions

$$E_{i\alpha_i}(x) = D_i(e_{i\alpha_i}, x_i) \quad \text{for } x_i \in \text{ri } \mathcal{X}_i, \alpha_i \in \mathcal{C}_i, i \in \mathcal{N}, \tag{D.36}$$

can be combined to yield a well-behaved "Lyapunov-like" function for $\mathcal{S}$.

The definition of these energy functions is motivated by the following property, which will also play a crucial role in deriving convergence rates:

**Lemma D.1.** *If the kernels verify $\theta_i(0) < \infty$ for every player $i \in \mathcal{N}$, then there exist constants $c_1 < \infty$ and $c_2 > -\infty$ such that*

$$\sum_{i \in \mathcal{N}} \sum_{\alpha_i \in \mathcal{C}_i} \Phi_i[c_2 - E_{i\alpha_i}(x)] \leq \text{dist}_1(x, \mathcal{S}) \leq 2 \sum_{i \in \mathcal{N}} \sum_{\alpha_i \in \mathcal{C}_i} \Phi_i[c_1 - E_{i\alpha_i}(x)] \tag{D.37}$$

*with $\Phi_i$ defined as $\Phi_i(z) = (\theta_i')^{-1}(z)$.*

*Proof.* From the definitions of the Bregman divergence and of the regularizers, we can write

$$E_{i\alpha_i}(x) = h_i(e_{i\alpha_i}) - h_i(x_i) - \langle \nabla h_i(x_i), e_{i\alpha_i} - x_i \rangle = c_{i\alpha_i} - \sum_{\beta_i} \theta_i(x_{i\beta_i}) - \theta_i'(x_{i\alpha_i}) + \sum_{\beta_i} x_{i\beta_i} \theta_i'(x_{i\beta_i}), \tag{D.38}$$

where $c_{i\alpha_i}$ is a finite constant. As a result, the convexity of $\theta_i$ yields

$$E_{i\alpha_i}(x) \leq c_{i\alpha_i} - n_i \min_{z \in [0,1]} \theta_i(z) - \theta_i'(x_{i\alpha_i}) + \theta_i'(1) \leq c_1 - \theta_i'(x_{i\alpha_i}), \tag{D.39}$$

for some constant $c_1 < \infty$ obtained by taking the maximum over all actions $\alpha_i \in \mathcal{C}_i$ and all players $i \in \mathcal{N}$. On the other hand, note that for all players $i \in \mathcal{N}$ and every $z \in (0, 1)$, we have

$$z\theta_i'(z) = \int_0^z \theta_i'(z) dt \geq \int_0^z \theta_i'(t) dt = \theta_i(z) - \theta_i(0), \tag{D.40}$$

where the inequality comes from the (strong) convexity of $\theta_i$. As such, we also get

$$E_{i\alpha_i}(x) \geq c_{i\alpha_i} - \sum_{\beta_i} \theta_i(x_{i\beta_i}) - \theta_i'(x_{i\alpha_i}) + \sum_{\beta_i} [\theta_i(x_{i\beta_i}) - \theta_i(0)] = c_{i\alpha_i} - n_i \theta_i(0) - \theta_i'(x_{i\alpha_i}) \geq c_2 - \theta_i'(x_{i\alpha_i}), \tag{D.41}$$

where $c_2 > -\infty$ thanks to the assumption that $\theta_i(0) < \infty$. Putting both sides together, we therefore obtain

$$\Phi_i[c_2 - E_{i\alpha_i}(x)] \leq x_{i\alpha_i} \leq \Phi_i[c_1 - E_{i\alpha_i}(x)] \tag{D.42}$$

for every $\alpha_i \in \mathcal{C}_i, i \in \mathcal{N}, x \in \text{ri } \mathcal{X}$. Now, note that by definition of the face $\mathcal{S}$,

$$\text{dist}_1(x, \mathcal{S}) := \min_{x' \in \mathcal{S}} \|x - x'\|_1 = \min_{x' \in \mathcal{S}} \sum_{i \in \mathcal{N}} \left( \sum_{\alpha_i \in \mathcal{C}_i} x_{i\alpha_i} + \sum_{\alpha_i \in \mathcal{B}_i} |x_{i\alpha_i} - x_{i\alpha_i}'| \right) \geq \sum_{i \in \mathcal{N}} \sum_{\alpha_i \in \mathcal{C}_i} x_{i\alpha_i}. \tag{D.43}$$

Furthermore, if we pick some arbitrary pure action $\alpha^* \in \mathcal{B}$ and define

$$x_{i\alpha_i}' = \begin{cases} 0 & \text{if } \alpha_i \in \mathcal{C}_i \\ x_{i\alpha_i} & \text{if } \alpha_i \in \mathcal{B}_i \setminus \{\alpha_i *\} \\ x_{i\alpha_i^*} + \sum_{\beta_i \in \mathcal{C}_i} x_{i\beta_i} & \text{if } \alpha_i = \alpha_i^* \end{cases} \tag{D.44}$$

for each $x \in \mathcal{X}$, then $x' \in \mathcal{S}$ by construction and we get

$$\|x - x'\|_1 = \sum_{i \in \mathcal{N}} \left( \sum_{\alpha_i \in \mathcal{C}_i} x_{i\alpha_i} + \sum_{\beta \in \mathcal{C}_i} x_{i\beta_i} \right) = 2 \sum_{i \in \mathcal{N}} \sum_{\alpha_i \in \mathcal{C}_i} x_{i\alpha_i}. \tag{D.45}$$

Consequently, Eqs. (D.43) and (D.45) yield

$$\sum_{i \in \mathcal{N}} \sum_{\alpha_i \in \mathcal{C}_i} x_{i\alpha_i} \leq \text{dist}_1(x, \mathcal{S}) \leq 2 \sum_{i \in \mathcal{N}} \sum_{\alpha_i \in \mathcal{C}_i} x_{i\alpha_i}, \tag{D.46}$$

which concludes the proof when combined with Eq. (D.42). ∎

According to Lemma D.1, we can derive two consistency results for the family $E_{i\alpha_i}(x)$:

1. $x \to \mathcal{S}$ if and only if $E_{i\alpha_i}(x) \to \infty$ for every action $\alpha_i \in \mathcal{C}_i$ and every player $i \in \mathcal{N}$;

2. For every neighborhood $\mathcal{U}_{\mathcal{S}}$ of $\mathcal{S}$, there exist constants $M' \geq M > 0$ such that $\mathcal{U}_{M'} \subseteq \mathcal{U}_{\mathcal{S}} \subseteq \mathcal{U}_M$, where

$$\mathcal{U}_M := \{x \in \mathcal{X} : E_{i\alpha_i}(x) > M \text{ for every } \alpha_i \in \mathcal{C}_i, i \in \mathcal{N}\}, \quad M > 0. \tag{D.47}$$

*Remark* 3. Our proof scheme for showing stability of closed under better replies faces is similar to the one proposed by Boone & Mertikopoulos [12] in a discrete setting, but uses a different family of energy functions in order to fix a flaw in their initialization domain's construction. To better understand this flaw, recall that they define the set of *outward deviations from $\mathcal{S}$* as $\mathcal{Z} = \{e_{i\beta_i} - e_{i\alpha_i} : \alpha_i \in \mathcal{B}_i, \beta_i \in \mathcal{C}_i, i \in \mathcal{N}\}$, and they argue that the associated family of energy functions $E_z(y) = \langle y, z \rangle$ forms a good Lyapunov function for $\mathcal{S}$. In particular, they show that $x := Q(y)$ converges to $\mathcal{S}$ whenever $\max_{z \in \mathcal{Z}} E_z(y) \to -\infty$, which is similar property that we have obtained in Lemma D.1. However, in their framework, the reverse implication is *not true* in general, i.e., there could be a sequence $(x^n)$ converging to $\mathcal{S}$ but such that $\max_{z \in \mathcal{Z}} E_z(y^n)$ remains lower-bounded. For instance, if there exists a player $i \in \mathcal{N}$ for which $|\mathcal{B}_i| \geq 2$ (this is equivalent to ask that $\mathcal{S}$ is not restricted to a vertex of $\mathcal{X}$) and if we fix some pure strategies $\alpha^* \in \mathcal{B}, \alpha_i \in \mathcal{B}_i \setminus \{\alpha_i^*\}, \beta_i \in \mathcal{C}_i$, then any sequence $(x^n)$ converging to $e_{\alpha^*} \in \mathcal{S}$ with $x_{i\alpha_i}^n = x_{i\beta_i}^n$ satisfies

$$\inf_{n \geq 1} E_z(y^n) = \inf_{n \geq 1} y_{i\beta_i}^n - y_{i\alpha_i}^n = \inf_{n \geq 1} \theta_i'(x_{i\beta_i}^n) - \theta_i'(x_{i\alpha_i}^n) = 0 > -\infty \tag{D.48}$$

for the deviation $z = e_{i\beta_i} - e_{i\alpha_i} \in \mathcal{Z}$. Due to this subtlety, the initialization domain $\mathcal{U}_0 = \{x \in \mathcal{X} : \max_{z \in \mathcal{Z}} E_z(y) < -2M, x = Q(y)\}$ (for $M > 0$ big enough) proposed by Boone & Mertikopoulos [12] does not describe a bona fide neighborhood of $\mathcal{S}$: in fact, from the previous example, the vertex $e_{\alpha^*} \in \mathcal{S}$ is not even included in $\mathcal{U}_0$. Since the initialization domain is required to be a neighborhood of the whole face in order to verify the stochastic stability criterion (cf. Definition 1), it means that the energy functions used in [12] are not precise enough to prove such a result.

To gain insight into how these energy functions arise in local stability proofs, let us assume $\mathcal{S}$ to be closed under better replies. By definition, we then know that $u_i(\alpha_i; x_{-i}) - u_i(x) < 0$ for every $\alpha_i \in \mathcal{C}_i, i \in \mathcal{N}$ and all $x \in \mathcal{S}$. Since the payoff functions $u_i$ are continuous on the compact space $\mathcal{X}$, it implies that there exists a neighborhood $\mathcal{U}$ of $\mathcal{S}$ and a finite constant $m > 0$ such that

$$u_i(\alpha_i; x_{-i}) - u_i(x) \leq -m \quad \text{for every } \alpha_i \in \mathcal{C}_i, i \in \mathcal{N}, x \in \mathcal{U}. \tag{D.49}$$

Now, let $X(t)$ be a solution orbit of (S-FTRL). Using Itô's lemma in tandem with the dual representation of the Bregman divergence (cf. Appendix A and the proof of Lemma F.1), we then get

$$dE_{i\alpha_i}(X(t)) = [u_i(X) - u_i(\alpha_i; X_{-i})]dt + \frac{1}{2} \text{tr}(\text{Jac } Q_i(Y_i)\Sigma_i(X))dt + \langle X_i - e_{i\alpha_i}, dM_i(t) \rangle \tag{D.50}$$

Specifically, if we temporarily ignore the martingale term $\langle X_i - e_{i\alpha_i}, dM_i(t) \rangle$ (i.e., if we consider the noiseless setting), then all energy functions are expected to locally increase along trajectories of (S-FTRL) evolving in $\mathcal{U}$ as $dE_{i\alpha_i}(X(t)) \geq mdt$. As a result, player's strategies will remain in $\mathcal{U}$ for any time and converge to $\mathcal{S}$ thanks to Lemma D.1.

However, once the noise term is reintroduced, this local increase of the energy no longer holds: the noise, being modelled as a continuous martingale with zero mean, can counterbalance the positive drift and, with positive probability, cause the energy to decrease locally. For instance, under Assumption 1, we know that trajectories of (S-FTRL) cannot even stay in $\mathcal{U}$ with probability 1; they always visit any strategy in ri $\mathcal{X}$ with positive probability (cf. Eq. (B.4) and the related discussion about uniform ellipticity in Appendix B).

With that being said, although trajectories could exit $\mathcal{U}$ with positive probability, we can still show that the total energy converges to $\infty$ along those trajectories that do stay into $\mathcal{U}$:

**Claim 3.** *If $E = \{X(t) \in \mathcal{U}$ for every $t \geq 0\}$ has positive probability, then $E_{i\alpha_i}(X(t)) \to \infty$ almost surely on $E$ for every $\alpha_i \in \mathcal{C}_i, i \in \mathcal{N}$.*

*Proof of Claim 3.* On the event $E$, integrating Eq. (D.50) yields

$$E_{i\alpha_i}(X(t)) = E_{i\alpha_i}(X(0)) + \int_0^t dE_{i\alpha_i}(X(s))ds \geq E_{i\alpha_i}(x) + mt + \xi_{i\alpha_i}(t) \tag{D.51}$$

where $\xi_{i\alpha_i}(t)$ is the square-integrable martingale given by $d\xi_{i\alpha_i}(t) = \langle X_i(t) - e_{i\alpha_i}, dM_i(t) \rangle$. By definition of $M_i(t)$, the quadratic variation of $\xi_{i\alpha_i}(t)$ can be upper-bounded as

$$d[\xi_{i\alpha_i}](t) = (X_i - e_{i\alpha_i})^T d[M_i](t)(X_i - e_{i\alpha_i}) = (X_i - e_{i\alpha_i})^T \Sigma_i(t)(X_i - e_{i\alpha_i})dt \leq \|X_i - e_{i\alpha_i}\|_2^2 \sigma_{\max}^2 dt \leq 2\sigma_{\max}^2 dt, \tag{D.52}$$

where $\sigma_{\max} < \infty$ by continuity of noise coefficients and compactness of $\mathcal{X}$. The strong law of large numbers for martingales (cf. Lemma B.5) can then be used to obtain $E_{i\alpha_i}(X(t)) \to \infty$ almost surely on the event $E$, which finishes the proof. ∎

With this claim established, we begin to see the connection to stochastically asymptotic stability: if we can demonstrate that trajectories remain within $\mathcal{U}$ with arbitrarily high probability, depending on how close their initial conditions are to $\mathcal{S}$, then we would directly obtain convergence toward $\mathcal{S}$ with the same (arbitrarily high) probability.

To show that, we use Lemma D.1 to describe the neighborhood $\mathcal{U}$ through the energy functions $E_{i\alpha_i}$. Indeed, we know from the aforementioned lemma that there exists a constant $M > 0$ (which can be taken as large as needed) such that $x \in \mathcal{U}$ whenever $x \in \mathcal{U}_M$, where we recall that $\mathcal{U}_M$ denotes the non-empty subset of $\mathcal{X}$ given by

$$\mathcal{U}_M = \{x \in \mathcal{X} : E_{i\alpha_i}(x) > M \text{ for every } \alpha_i \in \mathcal{C}_i, i \in \mathcal{N}\}. \tag{D.53}$$

The main idea is to show that if the initial condition is sufficiently close to $\mathcal{S}$, say with $x \in \mathcal{U}_{2M}$ (which describes a neighborhood of $S$ thanks to Lemma D.1), then we can choose $M$ big enough so that every energy functions remains above $M$ at all times, with arbitrarily high probability. In other words, we want to ensure that the noise perturbations are not strong enough to counterbalance the linearly increasing drift. More precisely, we aim to prove the following intermediate result:

**Claim 4.** *For every $\varepsilon > 0$, there exists $M := M(\varepsilon) > 0$ big enough such that*

$$\mathbb{P}_x(X(t) \in \mathcal{U}_M \text{ for every } t \geq 0) \geq 1 - \varepsilon \tag{D.54}$$

*whenever $x \in \mathcal{U}_{2M}$.*

*Proof of Claim 4.* For every $M$ big enough, let $\tau_M$ denotes the first time at which the minimal energy is lower than $M$, viz.

$$\tau_M = \inf\{t \geq 0 : \min_{\alpha_i \in \mathcal{C}_i, i \in \mathcal{N}} E_{i\alpha_i}(X(t)) \leq M\}. \tag{D.55}$$

Proving the claim is therefore equivalent to showing that $\mathbb{P}(\tau_M = \infty) \geq 1 - \varepsilon$ for a well-chosen constant $M$. To prove this, we will reduce the problem of estimating $\tau_M$ to examining specific hitting times of $\xi_{i\alpha_i}(t)$. Since $\xi_{i\alpha_i}(t)$ is a continuous square-integrable martingale, these hitting times will be significantly easier to estimate using standard results in stochastic analysis.

First, let us define the auxiliary hitting time

$$\tau = \inf\{t \geq 0 : \xi_{i\alpha_i}(t) = -M - mt \text{ for some } \alpha_i \in \mathcal{C}_i, i \in \mathcal{N}\}. \tag{D.56}$$

We claim that $\tau \leq \tau_M$ almost surely. Indeed, let us assume on the other hand that there exists a non-negligible event on which $\tau > \tau_M$. Then $\tau_M$ is necessarily finite on this event, and $X(t) \in \mathcal{U}$ for every $t < \tau_M$, so Eq. (D.50) leads to

$$E_{i\alpha_i}(X(\tau_M)) > 2M + m\tau_M + \xi_{i\alpha_i}(\tau_M) > 2M + m\tau_M - M - m\tau_M = M \tag{D.57}$$

for every $\alpha_i \in \mathcal{C}_i$ and $i \in \mathcal{N}$, where the second inequality is a consequence of $\tau_M < \tau$. This contradicts the definition of $\tau_M$, which prove that $\tau \leq \tau_M$ with probability 1 as required.

It follows from the previous argument that $\{\tau = \infty\} \subseteq \{\tau_M = \infty\}$, which readily implies the bound

$$\mathbb{P}(\tau_M < \infty) \leq \mathbb{P}(\tau < \infty) \leq \sum_{i \in \mathcal{N}} \sum_{\alpha_i \in \mathcal{C}_i} \mathbb{P}(\tau_{i\alpha_i} < \infty), \tag{D.58}$$

where $\tau_{i\alpha_i}$ is the hitting time given by

$$\tau_{i\alpha_i} = \inf\{t \geq 0 : \xi_{i\alpha_i}(t) = -M - mt\}. \tag{D.59}$$

To estimate the probability that $\tau_{i\alpha_i}$ is finite, remember that the quadratic variation of $\xi_{i\alpha_i}(t)$ is upper-bounded by $2\sigma_{\max}^2 t$ (a.s.). Moreover, the time-change theorem for continuous martingales [29, p. 174] states that there exists a standard Brownian motion $\widetilde{W}(t)$ (defined on a possibly enlarged probability space) such that $\xi_{i\alpha_i}(t) = \widetilde{W}([\xi_{i\alpha_i}](t))$.

Accordingly, if we let $\tilde{\tau}$ denotes the stopping time

$$\tilde{\tau} = \inf\left\{t \geq 0 : \widetilde{W}(t) = -M - \frac{mt}{2\sigma_{max}^2}\right\}, \tag{D.60}$$

then we readily get $\mathbb{P}(\tau_{i\alpha_i} < \infty) \leq \mathbb{P}(\tilde{\tau} < \infty)$ for every $\alpha_i \in \mathcal{C}_i, i \in \mathcal{N}$. This last hitting time is easily computed as it only involves a standard Brownian motion, for which classical results of the literature (see for instance [29, Subsection 3.5.C]) yield $\mathbb{P}(\tilde{\tau} < \infty) = e^{-\lambda M}$ with $\lambda = m/\sigma_{max}^2$.

Putting all these estimations back into Eq. (D.58) then allows us to obtain

$$\mathbb{P}(\tau_M < \infty) \leq |\mathcal{C}| e^{-\lambda M}. \tag{D.61}$$

where $|\mathcal{C}| := \sum_{i \in \mathcal{N}} |\mathcal{C}_i|$. In particular, if we take $M$ big enough so that $M > \frac{\sigma_{max}^2}{m} \log(|\mathcal{C}|/\varepsilon)$, then we finally get $\mathbb{P}(\tau_M = \infty) \geq 1 - \varepsilon$, which finishes the proof. ∎

With Claims 3 and 4 established, we are now ready to prove Theorem 2.

*Proof of Theorem 2 (club $\implies$ stable).* Let us fix a threshold $\varepsilon > 0$ and a neighborhood $\mathcal{U}_0$ of $\mathcal{S}$. By previously discussed arguments, if $\mathcal{S}$ is closed under better replies, we can find a neighborhood $\mathcal{U}$ and a constant $m > 0$ small enough such that

$$u_i(\alpha_i; x_{-i}) - u_i(x) \leq -m \quad \text{for every } \alpha_i \in \mathcal{C}_i, i \in \mathcal{N}, x \in \mathcal{U}. \tag{D.62}$$

Furthermore, we can choose $\mathcal{U} \subseteq \mathcal{U}_0$ without loss of generality. Claim 4 therefore yields a constant $M > 0$ big enough so that

$$\mathbb{P}_x(X(t) \in \mathcal{U}_0 \text{ for every } t \geq 0) \geq \mathbb{P}_x(X(t) \in \mathcal{U} \text{ for every } t \geq 0) \geq \mathbb{P}_x(X(t) \in \mathcal{U}_M \text{ for every } t \geq 0) \geq 1 - \varepsilon \tag{D.63}$$

whenever $x \in \mathcal{U}_{2M}$ (which describes a neighborhood of $\mathcal{S}$). Moreover, $X(t) \to \mathcal{S}$ almost surely on the event $\{X(t) \in \mathcal{U}$ for every $t \geq 0\}$ by Claim 3 and Lemma D.1, so $\mathcal{S}$ is stochastically asymptotically stable as required. ∎

*Proof of Theorem 2 (convergence rate).* Coming back to Eq. (D.51), we obtain that, on the event $E = \{X(t) \in \mathcal{U}$ for every $t \geq 0\}$ which has probability greater than $1 - \varepsilon$,

$$E_{i\alpha_i}(X(t)) \geq mt + \xi_{i\alpha_i}(t) \tag{D.64}$$

for every pure actions $\alpha_i \in \mathcal{C}_i$ and all players $i \in \mathcal{N}$, where $\xi_{i\alpha_i}(t)$ is a square-integrable martingale with $[\xi_{i\alpha_i}](t) \leq 2\sigma_{\max}^2 t$. In particular, if $[\xi_{i\alpha_i}](t) \nearrow \infty$, then the law of iterated logarithm (Lemma B.4) yields

$$\limsup_{t \to \infty} \frac{|\xi_{i\alpha_i}(t)|}{\sigma_{max}\sqrt{t \log \log t}} = \limsup_{t \to \infty} \frac{|\xi_{i\alpha_i}(t)|}{\sqrt{2[\xi_{i\alpha_i}](t) \log \log [\xi_{i\alpha_i}](t)}} \sqrt{\frac{2[\xi_{i\alpha_i}](t) \log \log [\xi_{i\alpha_i}](t)}{\sigma_{max}^2 t \log \log t}} \leq 2\sqrt{2} \quad (a.s.) \tag{D.65}$$

On the other hand, if $[\xi_{i\alpha_i}](t) \nearrow [\xi_{i\alpha_i}](\infty) < \infty$, then $\xi_{i\alpha_i}(t)$ converges to a finite random variable and so $|\xi_{i\alpha_i}(t)| = o(\sigma_{max}\sqrt{t \log \log t})$. Consequently, as $\mathcal{C}_i$ and $\mathcal{N}$ are finite, it implies that $\max_{\alpha_i \in \mathcal{C}_i, i \in \mathcal{N}} |\xi_{i\alpha_i}(t)| = \mathcal{O}\left(\sigma_{max}\sqrt{t \log \log t}\right)$ (a.s.) in any cases.

Now, in virtue of Lemma D.1 and by the fact that $\Phi_i$ is increasing ($\theta_i$ is convex), we therefore get

$$d_1(X(t), \mathcal{S}) \leq 2 \sum_{i \in \mathcal{N}} \sum_{\alpha_i \in \mathcal{C}_i} \Phi_i \left[ c_1 - mt + \max_{\alpha_i \in \mathcal{C}_i, i \in \mathcal{N}} |\xi_{i\alpha_i}(t)| \right] \lesssim \Phi \left[ c_1 - mt + \mathcal{O}\left( \sigma_{max} \sqrt{t \log \log t} \right) \right] \tag{D.66}$$

on the event $E$, which readily finish the proof. ∎

*Proof of Theorem 2 (stable $\implies$ club).* Assume that the face $\mathcal{S}$ is stochastically asymptotically stable but not closed under better replies. Then, there exists a player $i \in \mathcal{N}$ and pure strategies $\alpha \in \mathcal{B}$, $\alpha_i' \in \mathcal{C}_i$ such that $u_i(\alpha_i'; \alpha_{-i}) \geq u_i(\alpha)$.

Now, let us consider the restriction of (S-FTRL) to the face spanned by $\alpha$ and $(\alpha_i'; \alpha_{-i})$, by taking as initial condition $X(0) = (x_i; \alpha_{-i})$ where $\mathrm{supp}\, x_i = \{\alpha_i, \alpha_i'\}$. As $h$ is steep, trajectories of (S-FTRL) are invariant in this subface, meaning that $X(t) = (X_i(t); \alpha_{-i})$ with $\mathrm{supp}\, X_i(t) = \{\alpha_i, \alpha_i'\}$ for every time $t \geq 0$.

Consequently, the score difference between strategies $\alpha_i$ and $\alpha_i'$ is given at any time by

$$dY_{i\alpha_i} - dY_{i\alpha_i'} = \left[ u_i(\alpha) - u_i(\alpha_i'; \alpha_{-i}) \right] dt + d\xi(t) \leq d\xi(t), \tag{D.67}$$

where $\xi(t)$ is the square-integrable martingale given by $\xi(t) = \langle e_{i\alpha_i} - e_{i\alpha_i'}, M(t) \rangle$. The quadratic variation of $\xi(t)$ is lower-bounded as $[\xi](t) \geq 2\sigma_{min}^2 t$ (a.s.), which implies that $[\xi](t) \nearrow \infty$ when $t \to \infty$. We can therefore invoke the law of iterated logarithm for martingales (cf. Lemma B.4) to obtain $\liminf_{t \to \infty} \xi(t) = -\infty$ (a.s.). Coming back to Eq. (D.67), we then get

$$\liminf_{t \to \infty} Y_{i\alpha_i}(t) - Y_{i\alpha_i'}(t) \leq Y_{i\alpha_i}(0) - Y_{i\alpha_i'}(0) + \liminf_{t \to \infty} \xi(t) = -\infty \quad (a.s.). \tag{D.68}$$

This implies that, with probability one and for any initial condition, there exists a subsequence $(t_n) \nearrow \infty$ such that $Y_{i\alpha_i}(t_n) - Y_{i\alpha_i'}(t_n) \to -\infty$. Moreover, the Karush-Kuhn-Tucker conditions applied to the convex problem posed in the definition of $Q_i$ yield $y_{i\alpha_i} = \theta_i'(x_{i\alpha_i}) + \mu_i$ for some Lagrange multiplier $\mu_i \in \mathbb{R}$ whenever $x_i = Q_i(y_i)$. Consequently, we also get

$$\theta_i'(X_{i\alpha_i}(t_n)) = \theta_i'(X_{i\alpha_i'}(t_n)) + Y_{i\alpha_i}(t_n) - Y_{i\alpha_i'}(t_n) \leq \theta_i'(1) + Y_{i\alpha_i}(t_n) - Y_{i\alpha_i'}(t_n) \to -\infty \quad (a.s.), \tag{D.69}$$

meaning that $X_{i\alpha_i}(t_n) \to 0$ with probability one thanks to the convexity of $\theta_i$. This contradicts the fact that $\mathcal{S}$ is stochastically asymptotically stable (as $e_\alpha \in \mathcal{S}$, we should have $X(t) \to e_\alpha$ with positive probability for trajectories starting close enough to $e_\alpha$), thus $\mathcal{S}$ is necessarily closed under better replies. ∎

**D.3. Proof of Theorem 3.** In this section, we provide the proof of Theorem 3, whose statement is recalled below for convenience:

**Theorem 3.** *Suppose Assumptions 1–3 hold. Then:*

1. *$x^* \in \mathcal{X}$ is stochastically asymptotically stable under (S-FTRL) if and only if it is a strict equilibrium.*

2. *If $\mathbb{P}_x(\lim_{t \to \infty} X(t) = x^*) > 0$, then $x^*$ is a pure Nash equilibrium; in words, the only possible limits of (S-FTRL) are pure Nash equilibria.*

First, note that any stochastically asymptotically stable strategy must be pure, as shown by Corollary 3 (and similarly, any strategy toward which trajectories of (S-FTRL) converge with positive probability is pure). Therefore, it is sufficient to prove the following claims in order to establish Theorem 3:

**Claim 5.** *A pure strategy is stochastically asymptotically stable if and only if it is a strict Nash equilibrium.*

**Claim 6.** *Trajectories of (S-FTRL) converge only to Nash equilibria.*

*Proof of Claim 5.* By Theorem 2 and the fact that strict Nash equilibria are the only pure strategies whose support is closed under better replies, Claim 5 is automatically proven true. ∎

Claim 6, on the other hand, does not follow from any of our previous results. However, this property of (S-FTRL) has already been established by Bravo & Mertikopoulos [13] under the same kind of assumptions as ours. For the sake of completeness, we also provide the proof of this result here.

*Proof of Claim 6.* For the sole purpose of proving a contradiction, assume that there exists an event $E$ with positive probability and a strategy $x^*$ that is not a Nash equilibrium such that $X(t) \to x^*$ on $E$. As $x^*$ in not Nash, there must exists a player $i \in \mathcal{N}$ and actions $\alpha_i \in \text{supp}(x_i^*)$, $\beta_i \in \mathcal{A}_i$ verifying $v_{i\alpha_i}(x^*) < v_{i\beta_i}(x^*)$. By continuity of $v$, we can therefore find a small enough neighborhood $\mathcal{U}$ of $x^*$ and a positive constant $m > 0$ such that $v_{i\beta_i}(x) - v_{i\alpha_i}(x) \geq m$ for every $x \in \mathcal{U}$. As $X(t)$ converges to $x^*$ on $E$, it follows that there exists a finite (random) time $t_0$ such that $X(t) \in \mathcal{U}$ for every $t \geq t_0$ when conditioned on the event $E$. In particular, for $t$ big enough, we get

$$Y_{i\alpha_i}(t) - Y_{i\beta_i}(t) = y_{i\alpha_i} - y_{i\beta_i} + \int_0^{t_0} (v_{i\alpha_i} - v_{i\beta_i}) \, ds + \int_{t_0}^t (v_{i\alpha_i} - v_{i\beta_i}) \, ds + \xi(t) \leq C - mt + \xi(t), \tag{D.70}$$

where $C = y_{i\alpha_i} - y_{i\beta_i} + \int_0^{t_0} (v_{i\alpha_i} - v_{i\beta_i}) \, ds + mt_0$ is a random constant finite on $E$, and $\xi(t)$ is the square-integrable martingale given by

$$\xi(t) = \sum_{k=1}^m \int_0^t \left[ \sigma_{i\alpha_i k}(X(s)) - \sigma_{i\beta_i k}(X(s)) \right] dW_k(s). \tag{D.71}$$

As the quadratic variation of $\xi$ is upper-bounded by $C't$ for some positive deterministic constant $C'$, the strong law of large numbers (cf. Lemma B.5) implies that $Y_{i\alpha_i}(t) - Y_{i\beta_i}(t) \to -\infty$ on $E$, which directly leads to $X_{i\alpha_i}(t) \to 0$ by the same argument used in the proof of Theorem 2. This contradicts the fact that we should have $X_{i\alpha_i}(t) \to x_{i\alpha_i}^* > 0$ on $E$, hence proving that $x^*$ must be a Nash equilibrium. ∎

# E    Harmonic games and closedness under better replies

In this appendix, we present some general facts about *harmonic games*, following the framework of Legacci et al. [34], and establish the following original result: *no subface of a harmonic game can be closed under better replies.*

Originally introduced by Candogan et al. [15] and later generalized by Abdou et al. [1], harmonic games games model strategic scenarios where players have *conflicting, anti-aligned interests*. These games encompass two-player zero-sum games with a fully mixed equilibrium, a widely studied framework for modeling conflicting interactions [48]. By contrast, harmonic games are, in a precise sense, complementary to the class of *potential games* of Monderer & Shapley [52], which model strategic situations where players' interests *align* to maximize outcomes by collectively optimizing a shared potential function.

A defining feature of harmonic games is the existence of a special strategy, known as *strategic center*. Intuitively, the center has the property that the game's payoff vector is always *parallel to an ellipsoid* centered at this point, imparting a circular character to the game's strategic structure. This should be compared with the payoff vector of potential games [59], which is always *perpendicular the level sets* of the underlying potential function. In the deterministic setting, these strategic structures reflect in the dynamics as follows: harmonic games exhibit a *constant of motion* with bounded level sets, closely related to the *Fenchel coupling* introduced in Appendix A; conversely, potential games are characterized by a *Lyapunov function* in the form of their potential.

An important additional property of harmonic games is the following: excluding trivial situations where all unilateral payoff deviations are identically zero, harmonic games cannot have *strict* Nash equilibria. In Theorem E.1, we shall prove that this is an instance of a more general result: given a harmonic game $\Gamma = \Gamma(\mathcal{N}, \mathcal{A}, u)$, no proper subset $\mathcal{B} \subset \mathcal{A}$ of pure action profiles can be *closed under better replies* (club).

Since their introduction, harmonic games have generated a substantial body of literature; for a brief survey, we refer the reader to Legacci et al. [35].

**E.1. Harmonic games.**    Roughly speaking, a finite game $\Gamma = \Gamma(\mathcal{N}, \mathcal{A}, u)$ is *harmonic* if, whenever a player considers deviating *towards* a specific action profile $\alpha \in \mathcal{A}$, there are other players inclined to deviate *away* from that profile. More formally, a game is harmonic if there exist strictly positive *weights* $\mu_{i\alpha_i} > 0$ that each player $i \in \mathcal{N}$ assigns to each of their pure actions $\alpha_i \in \mathcal{A}_i$, such that for any action profile $\alpha \in \mathcal{A}$, the $\mu$-weighted sum of unilateral utility deviations to $\alpha$ is zero:

$$\sum_{i \in \mathcal{N}} \sum_{\beta_i \in \mathcal{A}_i} \mu_{i\beta_i} \left[ u_i(\alpha_i; \alpha_{-i}) - u_i(\beta_i; \alpha_{-i}) \right] = 0 \quad \text{for all } \alpha \in \mathcal{A}. \tag{E.1}$$

An immediate consequence of this definition is that, excluding trivial games where all unilateral payoff deviations vanish identically, harmonic games cannot have *strict* Nash equilibria: If $\alpha$ were a strict NE, all terms within the square brackets in Eq. (E.1) would by definition be strictly positive, leading to a contradiction.

This stands in stark contrast to the behavior of potential games, which always admit at least one pure (and generically strict) Nash equilibrium. This important difference stems from a rather deep geometrical fact: the *orthogonal* nature of potential and harmonic games, discussed in detail by Abdou et al. [1], Candogan et al. [15], which we briefly describe here. For fixed sets of players $\mathcal{N}$ and actions $\mathcal{A}$ with cardinalities $N$ and $n$ respectively, the payoff function $u$ of a game $\Gamma = \Gamma(\mathcal{N}, \mathcal{A}, u)$ can be represented as an element of a $nN$-dimensional vector space. Potential games and harmonic games comprise linear subspaces of this vector space; moreover, endowed with a suitable inner product, this vector space admits an orthogonal direct sum decomposition into the subspaces of potential and harmonic games (modulo *strategic equivalence*, an equivalence relation that identifies games with the same strategic structure). In other words, up to strategic equivalence, the payoff functions of any finite game $\Gamma = \Gamma(\mathcal{N}, \mathcal{A}, u)$ can be uniquely decomposed as $u = u_\mathrm{p} + u_\mathrm{h}$, where $(\mathcal{N}, \mathcal{A}, u_\mathrm{p})$ is a potential game and $(\mathcal{N}, \mathcal{A}, u_\mathrm{h})$ is a harmonic one.

Moving on, the defining property (E.1) for a finite game $\Gamma$ to be harmonic can be reformulated in terms its mixed extension's payoff vector $v$:

**Proposition E.1** ([34]). *A finite game* $\Gamma = \Gamma(\mathcal{N}, \mathcal{A}, u)$ *is harmonic if and only if its mixed extension* $\Delta(\Gamma) = (\mathcal{N}, \mathcal{X}, u)$ *fulfills the following: there exist (i) a tuple* $m \in \mathbb{R}^N_{>0}$, *and (ii) a fully mixed strategy* $q \in \mathrm{ri}\, \mathcal{X}$, *such that*

$$\sum_{i \in \mathcal{N}} m_i \langle v_i(x), x_i - q_i \rangle = 0 \quad \textit{for all } x \in \mathcal{X}. \tag{E.2}$$

*Whenever the above holds true, m and q are called respectively* mass *and* strategic center *of the underlying harmonic game.*

*Proof.* By standard arguments, Eq. (E.1) holds true if and only if its multilinear extension holds true, that is if and only if $\sum_{i \in \mathcal{N}} \sum_{\beta_i \in \mathcal{A}_i} \mu_{i\beta_i} [u_i(x_i; x_{-i}) - u_i(\beta_i; x_{-i})] = 0$ for all $x \in \mathcal{X}$. By Eqs. (2) and (3), this is equivalent to

$$\sum_{i \in \mathcal{N}} \left[ \langle v_i(x), x_i \rangle \sum_{\beta_i} \mu_{i\beta_i} - \langle \mu_i, v_i(x) \rangle \right] = \sum_{i \in \mathcal{N}} \sum_{\beta_i} \mu_{i\beta_i} \left[ \langle v_i(x), x_i - \frac{\mu_i}{\sum_{\beta_i} \mu_{i\beta_i}} \rangle \right] = 0 \quad \text{for all } x \in \mathcal{X}, \tag{E.3}$$

where we denote $\mu_i := (\mu_{i\alpha_i})_{\alpha_i \in \mathcal{A}_i}$, and all sums involving players' pure actions are taken over $\beta_i \in \mathcal{A}_i$. Now, assume that Eq. (E.3) holds true; then Eq. (E.2) holds true by setting $m_i = \sum_{\beta_i} \mu_{i\beta_i}$ and $q_{i\alpha_i} = \mu_{i\alpha_i} / \sum_{\beta_i} \mu_{i\beta_i}$, for all $i \in \mathcal{N}$ and $\alpha_i \in \mathcal{A}_i$. Conversely, if Eq. (E.2) holds true, then Eq. (E.3) holds also true by setting $\mu_{i\alpha_i} = m_i q_{i\alpha_i}$ for all $i \in \mathcal{N}$ and $\alpha_i \in \mathcal{A}_i$. ∎

An immediate corollary is that harmonic games encompass two-player zero-sum games with a fully mixed equilibrium:

**Corollary E.1.** *Every two-player zero-sum game with a fully mixed Nash equilibrium* $x^*$ *is harmonic, with weights* $\mu = x^*$.

*Proof.* An interior equilibrium of two-player zero-sum games is *null variationally stable* [47–49], that is $\sum_{i \in \mathcal{N}} \langle v_i(x), x_i - x_i^* \rangle = 0$ for all $x \in \mathcal{X}$, which implies that Eq. (E.2) is fulfilled with $m_i = 1$ and $q_i = x_i^*$. ∎

Proposition E.1 implies that the payoff vector $v$ of a harmonic game is parallel to the parametric family of hyperellipsoids given by the level sets of the function $x \mapsto \sum_{i \in \mathcal{N}} m_i (x_i - q_i)^2 \in \mathbb{R}$. Each hyperellipsoid in this family is centered at $q$, with the ratios among the semi-axes determined by the game's mass $m$, and their absolute lengths varying based on the level value. As shown by Legacci et al. [34, Th. 2], a consequence of this "circular" strategic structure is that the (deterministic) dynamics (FTRL) are "almost-periodic" in harmonic games – more precisely, they exhibit *Poincaré recurrence*. The authors establish this result by demonstrating that the FTRL dynamics in harmonic games admit a *constant of motion*; for completeness, we include the proof of this fact here.

**Proposition E.2.** *Assume* (FTRL) *is run in a harmonic game with mass m and strategic center q. Then the function*

$$F_{(m,q)}(y) := \sum_{i \in \mathcal{N}} m_i \left[ h_i(q_i) + h_i^*(y_i) - \langle q_i, y_i \rangle \right] \tag{E.4}$$

*remains constant along score trajectories* $y(t)$.

*Remark.* Eq. (E.4) expresses the sum of Fenchel couplings $F(q_i, y_i)$, as defined in Eq. (A.6), for each player $i \in \mathcal{N}$. Each term is computed relative to the strategic center $q_i$, and weighted by the corresponding mass $m_i$.

*Proof.* By chain rule,

$$\frac{d}{dt} F_{(m,q)}(y(t)) = \sum_{i \in \mathcal{N}} m_i \left[ \langle \nabla h_i^*(y_i), \dot{y}_i \rangle - \langle q_i, \dot{y}_i \rangle \right] = \sum_{i \in \mathcal{N}} m_i \langle x_i(t) - q_i, v_i(x(t)) \rangle = 0 \qquad \text{(E.5)}$$

where we used Eq. (A.7), the definition of (FTRL), and the characterization (E.2) of harmonic games. ∎

In the following sections, we will show that the absence of strict Nash equilibria in harmonic games is a special case of a broader principle, namely the absence of sets that are *closed under better replies*. To this end, we first review some basic properties of the better-reply correspondence over pure strategies, later extending our results to the case of mixed strategies.

**E.2. Pure better reply correspondence and club sets.** Given a finite game $\Gamma = \Gamma(\mathcal{N}, \mathcal{A}, u)$, Ritzberger & Weibull [56] introduced the *better reply correspondence* $\mathtt{btr}_i \colon \mathcal{A} \rightrightarrows \mathcal{A}_i$ as the set-valued map returning all (weakly) profitable deviations for player $i \in \mathcal{N}$ from a given action profile $\alpha \in \mathcal{A}$:

$$\mathtt{btr}_i(\alpha) = \{ \beta_i \in \mathcal{A}_i : u_i(\beta_i; \alpha_{-i}) \geq u_i(\alpha_i; \alpha_{-i}) \} . \qquad \text{(E.6)}$$

Extending this to all players, we will write $\mathtt{btr} \coloneqq \prod_{i \in \mathcal{N}} \mathtt{btr}_i \colon \mathcal{A} \rightrightarrows \mathcal{A}$ for the product correspondence; we then say that a subset $\mathcal{B} \subseteq \mathcal{A}$ of action profiles is *closed under better replies* (club) if $\mathtt{btr}(\alpha) \subseteq \mathcal{B}$ for all $\alpha \in \mathcal{B}$. Clearly, the whole $\mathcal{A}$ is always closed under better replies; the question of whether a set of action profiles is club or not is thus non-trivial only in the case of *proper* subsets $\mathcal{B} \subset \mathcal{A}$.

Next, we provide a simple characterization of club sets in terms of unilateral utility deviations: heuristically, a subset $\mathcal{B}$ of action profiles is club if and only if each unilateral deviation towards $\mathcal{B}$ is strictly profitable. To make this precise, we set some notation.

**Definition E.1.** Let $\Gamma = \Gamma(\mathcal{N}, \mathcal{A}, u)$ be a finite game. Two action profiles $\alpha, \beta \in \mathcal{A}$ form a *unilateral deviation* if they differ in the action of precisely one player; whenever this is the case, we write $\alpha \leftrightarrow \beta$, and simply say that $\beta$ is a deviation from $\alpha$. The deviations from any $\alpha \in \mathcal{A}$ constitute the set $\mathcal{Z}_\alpha \coloneqq \{ \beta \in \mathcal{A} : \beta \leftrightarrow \alpha \}$. Whenever $\alpha \leftrightarrow \beta$, we write $Du(\alpha, \beta) \coloneqq u_i(\alpha) - u_i(\beta_i; \alpha_{-i})$ for the payoff difference of the (necessarily existing and unique) player $i \in \mathcal{N}$ deviating from $\beta$ to $\alpha$; furthermore, if the game is harmonic with weights $\mu$, we write $\mu(\alpha, \beta) \coloneqq \mu_{i\beta_i}$, where again $i$ is the (existing and unique) deviating player.[5] Note that $Du(\cdot, \cdot)$ is anti-symmetric in its two arguments, while $\mu(\cdot, \cdot)$ does not generically possess any symmetry. Consider now a subset $\mathcal{B} \subseteq \mathcal{A}$: For any fixed $\alpha \in \mathcal{A}$, we denote by $\mathcal{Z}_{\alpha, \mathcal{B}} \coloneqq \mathcal{Z}_\alpha \cap \mathcal{B}$ the set of deviations from $\alpha$ that belong to $\mathcal{B}$, and by $\mathcal{Z}_{\alpha, \mathcal{B}}^* \coloneqq \mathcal{Z}_\alpha - \mathcal{Z}_{\alpha, \mathcal{B}}$ the set of deviations from $\alpha$ that do *not* belong to $\mathcal{B}$.

*Remark.* These notations allow to recast the defining property (E.1) of harmonic games as

$$\sum_{\beta \in \mathcal{Z}_\alpha} \mu(\alpha, \beta) Du(\alpha, \beta) = 0 \quad \text{for all } \alpha \in \mathcal{A} . \qquad \text{(E.7)}$$

**Lemma E.1** (Characterization of club sets). *Given a finite game $\Gamma = \Gamma(\mathcal{N}, \mathcal{A}, u)$, a proper subset $\mathcal{B} \subset \mathcal{A}$ of action profiles is closed under better replies if and only if the following condition holds true:*

$$Du(\alpha, \beta) > 0 \quad \text{for all } \alpha \in \mathcal{B} \text{ and all } \beta \in \mathcal{Z}_{\alpha, \mathcal{B}}^* . \qquad \text{(E.8)}$$

*Proof.* We proceed by contradiction. Let $\mathcal{B}$ be a club set, and assume the existence of a pair $\alpha \in \mathcal{B}$, $\beta \in \mathcal{Z}_{\alpha, \mathcal{B}}^*$ such that $Du(\alpha, \beta) \leq 0$. Then by Eq. (E.6), $\beta_i \in \mathtt{btr}_i(\alpha)$ for some unique $i \in \mathcal{N}$. Since $\beta_{-i} \equiv \alpha_{-i}$ and $\alpha_j \in \mathtt{btr}_j(\alpha)$ for all $j \in \mathcal{N}$, this implies that $\beta \in \mathtt{btr}(\alpha)$. This is a contradiction, since by assumption $\alpha \in \mathcal{B}$, $\beta \notin \mathcal{B}$, and $\mathcal{B}$ is closed under better replies.

Conversely, let Eq. (E.8) hold true, and assume $\mathcal{B}$ is not closed under better replies. Then there must exist a (non-strictly) profitable deviation leaving $\mathcal{B}$, that is pair $\alpha \in \mathcal{B}$, $\beta \in \mathcal{Z}_{\alpha, \mathcal{B}}^*$, such that $\beta \in \mathtt{btr}(\alpha)$. This in turn implies $Du(\alpha, \beta) \leq 0$, a contradiction. ∎

*Remark.* Note that in the above we consider *any* subset $\mathcal{B} \subset \mathcal{A}$; in particular, $\mathcal{B}$ is not necessarily spanning a subface of $\mathcal{X}$, that is, it is not necessarily factoring as $\mathcal{B} = \prod_{i \in \mathcal{N}} \mathcal{B}_i$ for a family $(\mathcal{B}_i \subset \mathcal{A}_i)_{i \in \mathcal{N}}$. The case in which $\mathcal{B}$ does span a subface of $\mathcal{X}$ is described in Corollary E.2.

---

[5]It is important to observe that $\mu(\alpha, \beta) = \mu_{i\beta_i}$ as defined in the text is indeed a function of both of its arguments: While the first argument does not explicitly appear on the right-hand side, player $i$ in the definition depends implicitly on both $\alpha$ and $\beta$. A more explicit, albeit verbose, definition would be $\mu(\alpha, \beta) = \mu_{\mathrm{play}(\alpha, \beta) \beta_{\mathrm{play}(\alpha, \beta)}}$, where $\mathrm{play}(\alpha, \beta) \in \mathcal{N}$ is the player deviating between $\alpha$ and $\beta$, for any pair $\alpha \leftrightarrow \beta$.

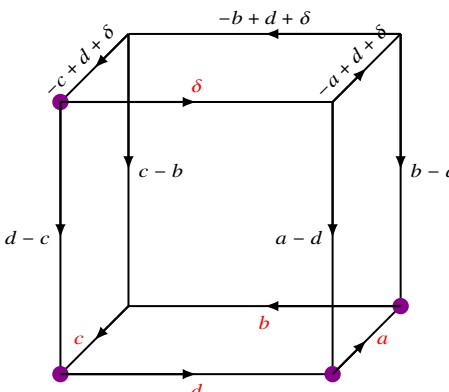
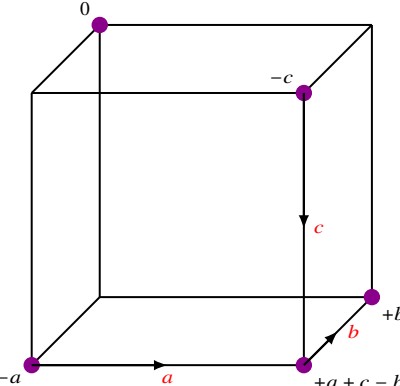

**Figure 2: Left** – Response graph of a $2 \times 2 \times 2$ uniform harmonic game. *Red labels on edges:* freely chosen payoff deviations $a, b, c, d, \delta \in \mathbb{R}$. *Black labels on edges:* payoff deviations constrained as to make the game harmonic according to Eq. (E.1). *Violet vertices:* generic subset $\mathcal{B}$ of pure action profiles. As detailed in Example E.1, there is no way to choose the parameters $a, b, c, d, \delta \in \mathbb{R}$ as to make the set $\mathcal{B}$ closed under better replies. **Right** – Response graph of a generic $2 \times 2 \times 2$ game. *Violet vertices:* generic subset $\mathcal{B}$ of pure action profiles. *Red labels on edges:* unilateral payoff deviations between elements of $\mathcal{B}$. *Black labels on vertices $\alpha \in \mathcal{B}$:* sum of unilateral payoff deviations towards $\alpha$ within $\mathcal{B}$, that is $\sum_{\beta \in \mathcal{Z}_{\alpha, \mathcal{B}}} Du(\alpha, \beta)$. As detailed in Lemma E.2, summing these values for all $\alpha \in \mathcal{B}$ yields zero by the anti-symmetry of the $Du(\cdot, \cdot)$ operator. Note that $\mathcal{Z}_{\alpha, \mathcal{B}} = \varnothing$ for $\alpha = \texttt{TLP}$, which hence does not contribute to the sum.

**E.3. No pure club sets in harmonic games.** With the notions of the previous sections at hand, we move on to show that harmonic games do not admit proper subsets of action profiles that are closed under better replies. Rather than directly stating and proving the general result, we take a more pedagogical approach to develop intuition by proceeding as follows: (*i*) We begin with Example E.1, which considers the special case of a *uniform* harmonic game, where $\mu_{i\alpha_i} \equiv 1$ for all $i \in \mathcal{N}$ and all $\alpha_i \in \mathcal{A}_i$, involving three players each having two available actions; (*ii*) Next, we provide the proof for the case of uniform harmonic games with arbitrary number of player and actions, as presented in Proposition E.3; (*iii*) Finally, we extend the result to general harmonic games, as detailed in Theorem E.1. Readers familiar with the topic may choose to proceed directly to this final proof.

**Example E.1** ($2 \times 2 \times 2$ uniform harmonic game). A harmonic game is called *uniform* if all players assign a weight of 1 to each of their actions. As shown by Candogan et al. [15, Prop. 4.1], the number of degrees of freedom available in choosing unilateral payoff deviations in a uniform harmonic game is given by the formula $(N - 1) \prod_{i \in \mathcal{N}} n_i - \sum_{i \in \mathcal{N}} \prod_{j \neq i} n_j + 1$, where $N$ is the number of players and $n_i$ is the number of actions available to player $i$. For a $2 \times 2 \times 2$ game, this yields 5 degrees of freedom; we denote them by $a, b, c, d, \delta \in \mathbb{R}$, and represent them in Fig. 2 (left) as payoff deviations on the game's *response graph* [15], an oriented graph with a node for each of the 8 action profiles and an edge for each of the 12 unilateral deviations of the game.[6]

The values of the remaining 7 payoff deviations required for the game to be harmonic are readily computed by Eq. (E.1), and are also represented in Fig. 2 (left). For example, label the action profiles of the game by the positions they occupy in the response graph.[7] Deviating *towards* the $\texttt{TLA}$ profile entails a payoff difference of $c - d$ for the first player, of $-\delta$ for the second player, and of $-c + d + \delta$ for the third player: Summing these differences yields zero, as required for the game to be harmonic.

Having the general form of a $2 \times 2 \times 2$ uniform harmonic game in place, we claim that one cannot find values for the free parameters $a, b, c, d, \delta \in \mathbb{R}$ such that some proper subset of action profiles be closed under better replies. Consider, for example, the set $\mathcal{B} = \{\texttt{TLA}, \texttt{BLA}, \texttt{BRA}, \texttt{BRP}\} \subset \mathcal{A}$, whose elements are highlighted in violet in Fig. 2 (left). By Lemma E.1, if $\mathcal{B}$ were closed under better replies, then each unilateral payoff deviation from a profile not in $\mathcal{B}$ to any profile in $\mathcal{B}$ would be strictly positive. This would yield the following system of inequalities for the parameters $a, b, c, d, \delta \in \mathbb{R}$:

$$\texttt{TLA} : \begin{cases} -c + d + \delta > 0 \\ -\delta > 0 \end{cases} \quad , \quad \texttt{BLA} : \{ c > 0 \quad , \quad \texttt{BRA} : \{ a - d > 0 \quad , \quad \texttt{BRP} : \begin{cases} b - a > 0 \\ -b > 0 \end{cases} . \tag{E.9}$$

---

[6]The label on each oriented edge represents the payoff difference for the player performing the deviation indicated by the arrow.

[7]$\texttt{T}$ and $\texttt{B}$ for *top* and *bottom*; $\texttt{L}$ and $\texttt{R}$ for *left* and *right*; $\texttt{A}$ and $\texttt{P}$ for *anterior* and *posterior*.

This system is easily verified to have no solution, as summing all of the left-hand sides yields zero. Consequently, it is impossible to select values for the parameters $a, b, c, d, \delta \in \mathbb{R}$ such that the set $\mathcal{B}$ closed under better replies. It is straightforward to verify that this does not depend on the particular choice of $\mathcal{B}$, and holds true for any subset of action profiles of the game. §

The reasoning employed in the previous example can be generalized to the case of uniform harmonic games with arbitrary number of players and actions, as follows.

**Proposition E.3.** *In a uniform harmonic game, no proper subset of action profiles can be closed under better replies.*

*Proof.* Let $\Gamma = \Gamma(\mathcal{N}, \mathcal{A}, u)$ be a uniform harmonic game, and assume by contradiction the existence of some $\mathcal{B} \subset \mathcal{A}$ that is club. For any $\alpha \in \mathcal{B}$, specialize Eq. (E.7) characterizing harmonic games to the uniform case ($\mu_{i\alpha_i} = 1$ for all $i \in \mathcal{N}, \alpha_i \in \mathcal{A}_i$), and split the domain of summation as the disjoint union $\mathcal{Z}_\alpha = \mathcal{Z}_{\alpha,\mathcal{B}} \sqcup \mathcal{Z}^*_{\alpha,\mathcal{B}}$, to obtain

$$\sum_{\beta \in \mathcal{Z}_{\alpha,\mathcal{B}}} Du(\alpha, \beta) + \sum_{\beta \in \mathcal{Z}^*_{\alpha,\mathcal{B}}} Du(\alpha, \beta) = 0 \quad \text{for all } \alpha \in \mathcal{B} \,. \tag{E.10}$$

By Lemma E.1, each term of the second summation is strictly positive, and so is the whole second summation; hence, the first summation must be strictly negative. In other words, if a uniform harmonic game admitted a club set $\mathcal{B}$, it should be true that

$$\sum_{\beta \in \mathcal{Z}_{\alpha,\mathcal{B}}} Du(\alpha, \beta) < 0 \quad \text{for all } \alpha \in \mathcal{B} \,. \tag{E.11}$$

However, this contradicts Lemma E.2 below, concluding our proof. ∎

**Lemma E.2** (Antisymmetric contraction). *In any finite game $\Gamma = \Gamma(\mathcal{N}, \mathcal{A}, u)$ and for any $\mathcal{B} \subseteq \mathcal{A}$,*

$$\sum_{\alpha \in \mathcal{B}} \sum_{\beta \in \mathcal{Z}_{\alpha,\mathcal{B}}} Du(\alpha, \beta) \equiv 0 \,. \tag{E.12}$$

*Proof.* The idea is that for each term appearing in the sum, it's opposite also appears, by anti-symmetry of $Du(\cdot, \cdot)$. Take $\alpha \in \mathcal{B}$. If $\mathcal{Z}_{\alpha,\mathcal{B}} = \varnothing$, there is nothing to show. Assume then that $\mathcal{Z}_{\alpha,\mathcal{B}} \neq \varnothing$. Each $\beta \in \mathcal{Z}_{\alpha,\mathcal{B}}$ contributes with a term $Du(\alpha, \beta)$ to the sum. However, $\beta \in \mathcal{Z}_{\alpha,\mathcal{B}}$ implies that $\alpha \in \mathcal{Z}_{\beta,\mathcal{B}}$, since $\alpha \leftrightarrow \beta$ is symmetric and both $\alpha$ and $\beta$ belong to $\mathcal{B}$. Hence each $\beta \in \mathcal{Z}_{\alpha,\mathcal{B}}$ contributes to the sum also with a term $Du(\beta, \alpha) = -Du(\alpha, \beta)$, canceling out the previous contribution. ∎

**Example E.2** (Antisymmetric contraction). Let Fig. 2 (right) be the response graph of a generic $2 \times 2 \times 2$ game. Consider the set of profiles $\alpha \in \mathcal{B} = \{\texttt{BLA}, \texttt{BRA}, \texttt{BRP}, \texttt{TRA}, \texttt{TLP}\}$, with payoff deviations among them given by $a, b, c \in \mathbb{R}$; it is not necessary to consider any of the deviations to $\texttt{TLP}$, since none of them belongs to $\mathcal{B}$. We have that $\sum_{\alpha \in \mathcal{B}} \sum_{\beta \in \mathcal{Z}_{\alpha,\mathcal{B}}} Du(\alpha, \beta) = -a + (a + c - b) + b - c \equiv 0$, in agreement with Lemma E.2.

Finally, we generalize this result to the case of harmonic games with non-unitary weights.

**Theorem E.1.** *Assume $\Gamma = \Gamma(\mathcal{N}, \mathcal{A}, u)$ is a harmonic game. Then no proper subset $\mathcal{B} \subset \mathcal{A}$ can be closed under better replies.*

*Proof.* The proof is formally almost identical to that of Proposition E.3, but hinges on a more general version of the key Lemma E.2, taking into account the "symmetry breaking" due to the presence of weights. Let $\Gamma = \Gamma(\mathcal{N}, \mathcal{A}, u)$ be a harmonic game with weights $\mu$, and assume by contradiction the existence of some $\mathcal{B} \subset \mathcal{A}$ that is club. For any $\alpha \in \mathcal{B}$, invoke Eq. (E.7) characterizing harmonic games, and split the domain of summation as the disjoint union $\mathcal{Z}_\alpha = \mathcal{Z}_{\alpha,\mathcal{B}} \sqcup \mathcal{Z}^*_{\alpha,\mathcal{B}}$, to obtain

$$\sum_{\beta \in \mathcal{Z}_{\alpha,\mathcal{B}}} \mu(\alpha, \beta) Du(\alpha, \beta) + \sum_{\beta \in \mathcal{Z}^*_{\alpha,\mathcal{B}}} \mu(\alpha, \beta) Du(\alpha, \beta) = 0 \quad \text{for all } \alpha \in \mathcal{B} \,. \tag{E.13}$$

By Lemma E.1, each term of the second summation is strictly positive, and so is the whole second summation; hence, the first summation must be strictly negative. In other words, if a harmonic game with weights $\mu$ admitted a club set $\mathcal{B}$, it should be true that

$$\sum_{\beta \in \mathcal{Z}_{\alpha,\mathcal{B}}} \mu(\alpha, \beta) Du(\alpha, \beta) < 0 \quad \text{for all } \alpha \in \mathcal{B} \,. \tag{E.14}$$

However, this leads to a contradiction: indeed, for any $\mathcal{B} \subseteq \mathcal{A}$, we claim that

$$\sum_{\alpha \in \mathcal{B}} \left[ \prod_{j \in \mathcal{N}} \mu_{j\alpha_j} \right] \sum_{\beta \in \mathcal{Z}_{\alpha,\mathcal{B}}} \mu(\alpha, \beta) Du(\alpha, \beta) \equiv 0 \,. \tag{E.15}$$

If we show that Eq. (E.15) holds true, our proof is complete; note that this equation is a generalization of Lemma E.2 in the presence of non-uniform weights. As such, it is proved analogously: recall that for any profile $\alpha \in \mathcal{B}$ and any deviation $\beta \in \mathcal{Z}_{\alpha,\mathcal{B}}$, the map $(\alpha, \beta) \mapsto Du(\alpha, \beta)$ is anti-symmetric, and observe that (on the same domain) the map $(\alpha, \beta) \mapsto \left[ \prod_{j \in \mathcal{N}} \mu_{j\alpha_j} \right] \mu(\alpha, \beta)$ is symmetric. To verify the latter statement, denote by $i \in \mathcal{N}$ the (existing and unique) player deviating between $\alpha$ and $\beta$, that is, $\beta = (\beta_i, \alpha_{-i})$. Then,

$$\left[ \prod_{j \in \mathcal{N}} \mu_{j\alpha_j} \right] \mu(\alpha, \beta) = \left[ \prod_{j \in \mathcal{N} - \{i\}} \mu_{j\alpha_j} \right] \mu_{i\alpha_i} \mu_{i\beta_i} = \left[ \prod_{j \in \mathcal{N}} \mu_{j\beta_j} \right] \mu(\beta, \alpha) \,. \tag{E.16}$$

For conciseness we denote this symmetric map by $S(\alpha, \beta) \equiv \left[ \prod_{j \in \mathcal{N}} \mu_{j\alpha_j} \right] \mu(\alpha, \beta)$ for all $\alpha \in \mathcal{B}, \beta \in \mathcal{Z}_{\alpha,\mathcal{B}}$. Eq. (E.15) then reduces to $\sum_{\alpha \in \mathcal{B}} \sum_{\beta \in \mathcal{Z}_{\alpha,\mathcal{B}}} S(\alpha, \beta) Du(\alpha, \beta) = 0$, and follows as the contraction of a symmetric map with an anti-symmetric one: for each term appearing in the sum, it's opposite also appears. Take $\alpha \in \mathcal{B}$. If $\mathcal{Z}_{\alpha,\mathcal{B}} = \varnothing$, there is nothing to show. Assume then that $\mathcal{Z}_{\alpha,\mathcal{B}} \neq \varnothing$. Each $\beta \in \mathcal{Z}_{\alpha,\mathcal{B}}$ contributes with a term $S(\alpha, \beta) Du(\alpha, \beta)$ to the sum. However, $\beta \in \mathcal{Z}_{\alpha,\mathcal{B}}$ implies that $\alpha \in \mathcal{Z}_{\beta,\mathcal{B}}$, since $\alpha \leftrightarrow \beta$ is symmetric and both $\alpha$ and $\beta$ belong to $\mathcal{B}$. Hence each $\beta \in \mathcal{Z}_{\alpha,\mathcal{B}}$ contributes to the sum also with a term $S(\beta, \alpha) Du(\beta, \alpha) = -S(\alpha, \beta) Du(\alpha, \beta)$, canceling out the previous contribution. ∎

**Example E.3** (Antisymmetric contraction revisited)**.** Consider again the deviations of Fig. 2 (right), this time assigning a positive weight $\mu_{i\alpha_i}$ to each player's action. Let $\mathcal{B} = \{\texttt{BLA}, \texttt{BRA}, \texttt{BRP}, \texttt{TRA}, \texttt{TLP}\}$. The condition given by Eq. (E.14) for $\mathcal{B}$ to be club assuming that the game is harmonic yields the following system of inequalities (one for each $\alpha \in \mathcal{B}$ such that $\mathcal{Z}_{\alpha,\mathcal{B}} \neq \varnothing$):

$$\begin{aligned} &\texttt{BLA} : \mu_{\text{R}}(-a) < 0 &\quad &\texttt{BRA} : \mu_{\text{L}} a + \mu_{\text{T}} c + \mu_{\text{P}}(-b) < 0 \\ &\texttt{BRP} : \mu_{\text{A}} b < 0 &\quad &\texttt{TRA} : \mu_{\text{B}}(-c) < 0 \end{aligned} \tag{E.17}$$

This system of inequalities admits no solution in the domain $\mu_{i\alpha_i} > 0, a, b, c \in R$. Indeed,

$$\begin{aligned} \sum_{\alpha \in \mathcal{B}} \left[ \prod_{i \in \mathcal{N}} \mu_{i\alpha_i} \right] \sum_{\beta \in \mathcal{Z}_{\alpha,\mathcal{B}}} \mu(\alpha, \beta) Du(\alpha, \beta) = \ & \mu_{\text{B}} \mu_{\text{L}} \mu_{\text{A}} [-\mu_{\text{R}} a] \\ &+ \mu_{\text{B}} \mu_{\text{R}} \mu_{\text{P}} [\mu_{\text{A}} b] \\ &+ \mu_{\text{T}} \mu_{\text{R}} \mu_{\text{A}} [-\mu_{\text{B}} c] \\ &+ \mu_{\text{B}} \mu_{\text{R}} \mu_{\text{A}} [\mu_{\text{L}} a + \mu_{\text{T}} c - \mu_{\text{P}} b] \equiv 0 \,, \end{aligned} \tag{E.18}$$

in agreement with Eq. (E.15), making it impossible for the system (E.17) to admit a solution.

**E.4. No club subfaces in harmonic games.** We conclude this appendix by extending Theorem E.1 to the case of mixed strategies. To set the notation we recall the definition of the following standard objects:

**Definition E.2.** Given the mixed extension $\Delta(\Gamma)$ of a finite game $\Gamma = \Gamma(\mathcal{N}, \mathcal{A}, u)$, the *support* of any $x_i \in \mathcal{X}_i$ is $\text{supp}(x_i) := \{\alpha_i \in \mathcal{A}_i : x_{i\alpha_i} > 0\}$. Given a family $(\mathcal{B}_i \subseteq \mathcal{A}_i)_{i \in \mathcal{N}}$, the *face spanned by* $\mathcal{B} := \prod_{i \in \mathcal{N}} \mathcal{B}_i$ is

$$\mathcal{S}_{\mathcal{B}} := \prod_{i \in \mathcal{N}} \{x_i \in \mathcal{X}_i : \text{supp}(x_i) \subseteq \mathcal{B}_i\} \equiv \prod_{i \in \mathcal{N}} \mathcal{S}_{\mathcal{B}_i} \,. \tag{E.19}$$

A *subface of* $\mathcal{X}$ (sometimes also referred to as a *proper face*) is a proper subset of $\mathcal{X}$ that can be written in the form (E.19) for some $(\mathcal{B}_i \subset \mathcal{A}_i)_{i \in \mathcal{N}}$. Moving on, the better reply correspondence (E.6) extends to mixed strategies as the correspondence $\texttt{btr} : \mathcal{X} \rightrightarrows \mathcal{X}$ define by

$$\texttt{btr}(x) = \prod_{i \in \mathcal{N}} \{x_i' \in \mathcal{X}_i : u(x_i', x_{-i}) \geq u(x)\} \,, \tag{E.20}$$

and we say that a subface $\mathcal{S}$ of $\mathcal{X}$ is *closed under better replies* (club) if $\texttt{btr}(x) \subseteq \mathcal{S}$ for all $x \in \mathcal{S}$.

**Corollary E.2.** *Assume* $\Delta(\Gamma)$ *is the mixed extension of a finite harmonic game* $\Gamma = \Gamma(\mathcal{N}, \mathcal{A}, u)$. *Then no subface of* $\mathcal{X}$ *can be closed under better replies.*

*Proof.* Let $\mathcal{S}_{\mathcal{B}} \subset \mathcal{X}$ be a subface of $\mathcal{X}$ with spanning set $\mathcal{B} = \prod_{i \in \mathcal{N}} \mathcal{B}_i \subset \mathcal{A}$, with $\mathcal{B}_i \subset \mathcal{A}_i$ for all $i \in \mathcal{N}$. Our claim follows as a consequence of Theorem E.1, and the fact that

$$\mathcal{S}_{\mathcal{B}} \text{ is club} \implies \mathcal{B} \text{ is club}. \tag{E.21}$$

Indeed, assume that $\mathcal{S}_{\mathcal{B}}$ is club. Then for any $x \in \mathcal{S}_{\mathcal{B}}$ and any $x' \in \mathcal{X}$, if $x' \in \texttt{btr}(x)$, that is, if $u_i(x'_i, x_{-i}) \geq u_i(x)$ for all $i \in \mathcal{N}$, then $x' \in \mathcal{S}_{\mathcal{B}}$. For any $\alpha \in \mathcal{B}$, this holds true in particular at the mixed profile $\alpha \in \mathcal{S}_{\mathcal{B}}$ such that $\text{supp}(\alpha_i) = \{\alpha_i\}$, i.e., at the mixed representation of the pure profile $\alpha \in \mathcal{B}$, in which each player $i$ plays the strategy $\alpha_i$ with probability 1. Hence, for any $\alpha \in \mathcal{B}$ and any $x' \in \mathcal{X}$, if $u_i(x'_i, \alpha_{-i}) \geq u_i(\alpha)$ for all $i \in \mathcal{N}$, then $x' \in \mathcal{S}_{\mathcal{B}}$. With the same reasoning, this holds true in particular at the mixed representation $\beta \in \mathcal{X}$ of any $\beta \in \mathcal{A}$: for any $\alpha \in \mathcal{B}$ and any $\beta \in \mathcal{X}$, if $u_i(\beta_i, \alpha_{-i}) \geq u_i(\alpha)$ for all $i \in \mathcal{N}$, then $\beta \in \mathcal{S}_{\mathcal{B}}$, meaning that $\text{supp}(\beta_i) \subseteq \mathcal{B}_i$, i.e., $\beta_i \in \mathcal{B}_i$. ∎

As a consequence, we can now prove Theorem 6, which we restate here for ease of reference:

**Theorem 6.** *Suppose Assumption 3 holds. If the game is harmonic, there is no proper face $\mathcal{S}$ of $\mathcal{X}$ that is stochastically asymptotically stable under* (S-FTRL).

*Proof.* By Theorem 2, stochastically asymptotically stable subfaces of $\mathcal{X}$ are necessarily closed under better replies. Corollary E.2 shows that harmonic game do not admit any such faces, which completes our proof. ∎

# F  Omitted proofs from Section 5

In this appendix, we prove the various results stated in Section 5 regarding the fragility of deterministic recurrence in a noisy environment. As mentioned in the main text, the primary object of interest of this section will be the energy function

$$H(x) = \sum_{i \in \mathcal{N}} m_i D_i(q_i, x_i) \tag{F.1}$$

where $m$ and $q$ denote respectively the mass and the strategy center of a fixed harmonic game $\Gamma$, and $D_i$ denotes the standard Bregman divergence generated by the regularizer function $h_i$.

In fact, we will not directly work with the energy function $H$ through its "primal" definition Eq. (F.1), but instead use its "dual" representation

$$F(y) = \sum_{i \in \mathcal{N}} m_i F_i(q_i, y_i) \tag{F.2}$$

where $F_i$ denotes the Fenchel coupling defined in Appendix A. In particular, Proposition A.2 implies that $H(x) = F(y)$ whenever $x = Q(y)$, which allows this change of function. This choice of representation greatly simplifies the computations, as it avoids the need to use an explicit expression of trajectories in $\mathcal{X}$. In the deterministic setting for instance, it provides a quick proof that the energy $H$ (resp. $F$) is a constant of motion for (FTRL) in harmonic games (cf. Appendix E).

**F.1. Proof of Theorems 4 and 5.**  The main idea underlying the proofs of Section 5 is to show that the infinitesimal generator $\mathcal{L}$ of (S-FTRL) applied to $F$ is positive on $\mathcal{Y}$. To be more precise, recall from Appendix B that

$$\mathbb{E}_y[F(Y(t))] = F(y) + \mathbb{E}_y\left[\int_0^t \mathcal{L}F(Y(s)) \, ds\right] \tag{F.3}$$

for every $t \geq 0$, so $t \mapsto \mathbb{E}_y[F(Y(t))]$ is increasing whenever $\mathcal{L}F$ is anywhere positive. The interesting aspect is that this result also holds when $t$ is replaced by any (almost surely bounded) hitting time $\tau$ (see Lemma B.1), enabling us to obtain even finer results beyond just the average increase of the energy function

For this reason, we begin with a preliminary lemma allowing us to accurately estimate this quantity in harmonic games:

**Lemma F.1.** *For every harmonic game $\Gamma$ and every $y \in \mathcal{Y}$,*

$$\frac{\sigma_{\min}^2}{2} \sum_{i \in \mathcal{N}} m_i \, \text{tr}(\text{Jac}\, Q_i(y_i)) \leq \mathcal{L}F(y) \leq \frac{\sigma_{\max}^2}{2} \sum_{i \in \mathcal{N}} m_i \, \text{tr}(\text{Jac}\, Q_i(y_i)). \tag{F.4}$$

*Proof.* Let us fix the player's index $i \in \mathcal{N}$. By Itô's formula, we have

$$dF_i(q_i, Y_i) = \langle \operatorname{grad} F_i(q, Y_i), dY_i \rangle + \frac{1}{2} \sum_{\alpha\beta} \frac{\partial^2 F_i}{\partial y_\alpha \partial y_\beta}(q_i, Y_i) d[Y_{i\alpha}, Y_{i\beta}] \tag{F.5}$$

$$= \langle Q_i(Y_i) - q_i, v_i(X) \rangle dt + \frac{1}{2} \sum_{\alpha\beta} \frac{\partial Q_{i\alpha}}{\partial y_\beta}(Y_i)(\Sigma_i(X))_{\alpha\beta} dt + d\xi_i \tag{F.6}$$

$$= \langle X_i - q_i, v_i(X) \rangle dt + \frac{1}{2} \operatorname{tr}(\operatorname{Jac} Q_i(Y_i) \Sigma_i(X)) dt + d\xi_i \tag{F.7}$$

where $\Sigma_i = \sigma_i \sigma_i^T$ and $\xi_i$ denotes a square-integrable martingale starting from 0. Consequently, we get

$$\mathcal{L}F(y) = \sum_{i \in \mathcal{N}} m_i \mathcal{L}F_i(q_i, y_i) = \frac{1}{2} \sum_{i \in \mathcal{N}} m_i \operatorname{tr}(\operatorname{Jac} Q_i(y_i) \Sigma_i(x)) \tag{F.8}$$

for every $y \in \mathcal{Y}$ and $x = Q(y)$.

Now, notice that the eigenvalues of each $\Sigma_i$ are all included in the interval $[\sigma_{\min}^2, \sigma_{\max}^2]$ by Assumption 1 on the diffusion matrix. Furthermore, $\operatorname{Jac} Q_i(y) = \operatorname{Hess} h_i^*(y)$ is symmetric and positive semi-definite for any $y \in \mathcal{Y}$ (this is a consequence of $h^*$ being convex and of Proposition A.1). By classical results of linear algebra, we therefore obtain

$$\sigma_{\min}^2 \operatorname{tr}(\operatorname{Jac} Q_i(y_i)) \leq \operatorname{tr}(\operatorname{Jac} Q_i(y_i) \Sigma_i(x)) \leq \sigma_{\max}^2 \operatorname{tr}(\operatorname{Jac} Q_i(y_i)) \tag{F.9}$$

for every player $i \in \mathcal{N}$, which finishes the proof when combined with Eq. (F.8). ∎

With this lemma in hand, we are now ready to prove every results stated in Section 5.

*Proof of Theorem 4 ($H \to \infty$ in average).* From Lemma F.1 and the convexity of $h$, it is evident that $\mathcal{L}F(y) \geq 0$ for every $y \in \mathcal{Y}$. According to Eq. (F.3), it then implies that $t \mapsto \mathbb{E}[F(Y(t))]$ is a non-decreasing function. In particular, it admits a (possible infinite) limit. Assume for a moment that this limit is finite. Then, Eq. (F.3) and Fubini's theorem would imply that $\int_0^t \mathcal{L}F(Y(s)) ds$ should be finite almost-surely. But as $\mathcal{L}F(y)$ is positive for every $y \in \mathcal{A}$, this is possible only if $\mathcal{L}F(Y(t)) \to 0$ almost-surely. Due to the lower bound of Lemma F.1, it would mean that $\operatorname{tr}(\operatorname{Jac} Q_i(Y_i(t))) \to 0$ for every player $i \in \mathcal{N}$, but this can only occurs if $X(t) = Q(Y(t))$ converges to the boundary bd $\mathcal{X}$ (indeed, this quantity is strictly positive on the relative interior of $\mathcal{X}$ due to Lemma A.1). However, this leads to a contradiction because $F(Y(t)) = H(X(t))$ explodes to infinity whenever $X(t)$ converges to the boundary. Consequently, the limit of $\mathbb{E}[F(Y(t))]$ is necessarily infinite as required. ∎

*Proof of Theorem 5.* 1. Let $\mathcal{K}$ be compact subset of $\mathcal{X}$ disjoint from bd$(\mathcal{X})$. Due to Dynkin's lemma (Lemma B.1) and Lemma F.1 we have, for every fixed $t \geq 0$,

$$M(\mathcal{K}) \geq \mathbb{E}_x[F(Y(\tau_{\mathcal{K}} \wedge t))] \geq F(y) + \frac{\sigma_{\min}^2}{2} \mathbb{E}_y \left[ \int_0^{\tau_{\mathcal{K}} \wedge t} \sum_{i \in \mathcal{N}} m_i \operatorname{tr}(\operatorname{Jac} Q_i(Y_i(s)) ds \right] \tag{F.10}$$

$$\geq F(y) + \frac{\sigma_{\min}^2}{2} m(\mathcal{K}) \mathbb{E}_x[\tau_{\mathcal{K}} \wedge t] \tag{F.11}$$

where

$$M(\mathcal{K}) := \max\{H(x) : x \in \mathcal{K}\} < \infty, \tag{F.12}$$

$$m(\mathcal{K}) := \min\left\{ \sum_{i \in \mathcal{N}} m_i \operatorname{tr}(\operatorname{Jac} Q_i(y_i)) : x = Q(y), x \in \mathcal{K} \right\} > 0. \tag{F.13}$$

The finiteness of $M(\mathcal{K})$ follows from properties of the Bregman divergence, while the positiveness of $m(\mathcal{K})$ comes from Lemma A.1. Rearranging the inequality and taking the monotone limit as $t \to \infty$, we therefore obtain

$$\mathbb{E}_x[\tau_{\mathcal{K}}] \leq 2 \frac{M(\mathcal{K}) - H(x)}{m(\mathcal{K}) \sigma_{\min}^2} < \infty; \tag{F.14}$$

hence the mean escape time is finite as required.

2. From Lemma F.1 and the fact that $h_i^*$ is both convex and $L$-smooth (cf. Appendix A), we obtain

$$0 \leq \mathcal{L}F(y) \leq \frac{L\sigma_{\min}^2}{2} \sum_{i \in \mathcal{N}} m_i =: K. \tag{F.15}$$

In particular, it implies that $F(Y(t))$ (resp. $H(X(t))$) is a submartingale (this is a consequence of the martingale characterization of infinitesimal generators, cf. Eq. (B.2)). Let us consider the compact subset $\mathcal{K} = \{x \in \mathcal{X} : H(x) \geq M\}$ and assume that $\mathbb{E}_x[\tau_{\mathcal{K}}] < \infty$ for some initial condition $x \in \mathrm{ri}\,\mathcal{X}$ such that $H(x) \geq M + 1$. Then, Dynkin's lemma and Eq. (F.15) yield

$$\mathbb{E}_x[H(X(\tau_{\mathcal{K}} \wedge t))] \leq H(x) + K\,\mathbb{E}_x[\tau_{\mathcal{K}} \wedge t] \leq H(x) + K\,\mathbb{E}_x[\tau_{\mathcal{K}}] < \infty. \tag{F.16}$$

Consequently, the stopped process $(H(X(\tau_{\mathcal{K}} \wedge t)))_{t \geq 0}$ is a uniformly integrable submartingale. Doob's martingale convergence theorem [29, Theorem 3.15 p.17] therefore implies that

$$H(x) \leq \mathbb{E}_x[H(X(\tau_{\mathcal{K}} \wedge t))] \to \mathbb{E}_x[H(X(\tau_{\mathcal{K}}))] \leq M, \tag{F.17}$$

which contradicts the fact that $H(x) \geq M + 1$. We therefore deduce that $\mathbb{E}_x[\tau_{\mathcal{K}}] = \infty$ for some $x \in \mathcal{K}$, so trajectories of (S-FTRL) cannot be positive recurrent in the sense of Definition 3. In particular, the mean escape time is also infinite for any compact subset containing $\mathrm{bd}(\mathcal{X})$ due to the transience/recurrence dichotomy of Theorem 7.

∎

*Proof of Theorem 4 (Hitting time estimate).* The proof follows directly from Eq. (F.14) with the particular choice $\mathcal{K} = \{x \in \mathcal{X} : H(x) \leq M\}$, which leads to $M(\mathcal{K}) = M$ and $m(\mathcal{K}) = \varepsilon(M)$. ∎

*Proof of Corollary 4.* Let us consider the entropic kernel $\theta_i(z) = z \log z$, and let us drop the player's index $i$ for a moment. A standard computation shows that, in this case, the Bregman divergence $D(q, x)$ is equal to the *Kullback-Leibler divergence* between $q$ and $x$, given by

$$\mathrm{KL}(q, x) = \sum_\alpha q_\alpha \log\left(\frac{q_\alpha}{x_\alpha}\right) = h(q) - \sum_\alpha q_\alpha \log x_\alpha. \tag{F.18}$$

On the other hand, computing the Jacobian matrix of the logit map $Q$ leads to

$$\mathrm{tr}(\mathrm{Jac}\,Q(y)) = \sum_\alpha x_\alpha(1 - x_\alpha). \tag{F.19}$$

Applying the inverse exponential function on Eq. (F.18) then yields

$$e^{-D(q,x)} = e^{-h(q)} \prod_\alpha x_\alpha^{q_\alpha} = e^{-h(q)}\left(\prod_\alpha x_\alpha\right)^{q^*} \prod_\alpha x_\alpha^{q_\alpha - q^*} \leq e^{-h(q)}\left(\prod_\alpha x_\alpha\right)^{q^*}, \tag{F.20}$$

where $q^* = \min_\alpha q_\alpha > 0$. Furthermore, we can upper bound $\prod_\gamma x_\gamma$ as

$$\prod_\gamma x_\gamma = \frac{1}{n(n-1)} \sum_{\alpha \neq \beta} \prod_\gamma x_\gamma \leq \frac{1}{n(n-1)} \sum_{\alpha \neq \beta} x_\alpha x_\beta = \frac{1}{n(n-1)} \sum_\alpha x_\alpha(1 - x_\alpha), \tag{F.21}$$

where the the penultimate inequality follows by taking $x_\gamma \leq 1$ for every $\gamma \notin \{\alpha, \beta\}$, and the last equality uses the simplex constraint $\sum_\alpha x_\alpha = 1$. Putting this bound back into Eq. (F.20) therefore leads to

$$e^{-\frac{1}{q^*}D(q,y)} \leq e^{-\frac{1}{q^*}h(q)}[n(n-1)]^{-1} \sum_\alpha x_\alpha(1 - x_\alpha) = c(q)\,\mathrm{tr}(\mathrm{Jac}\,Q(y)). \tag{F.22}$$

with $c > 0$ a constant depending only on $q$ and $n$.

Adding back the players' indices, we define $q^* = \min_i q_i^*$ and $c(q) = \max_i c_i(q_i)$, which allows us to write

$$\sum_{i \in \mathcal{N}} m_i\,\mathrm{tr}(\mathrm{Jac}\,Q_i(y_i)) \geq \frac{1}{c(q)} \sum_{i \in \mathcal{N}} m_i e^{-\frac{1}{q^*}D_i(q_i, x_i)} \geq \frac{m}{c(q)} e^{-\frac{1}{mq^*}\sum_i m_i D_i(q_i, x_i)} = c_1 e^{-c_2 H(q,x)}, \tag{F.23}$$

where $m = \sum_i m_i$, $c_1$ and $c_2$ are positive constants, and the penultimate inequality holds thanks to the convexity of $e^{-x}$. Accordingly, if $H(q, x) \leq M$, then

$$\sum_{i \in \mathcal{N}} m_i\,\mathrm{tr}(\mathrm{Jac}\,Q_i(y_i)) \geq c_1 e^{-c_2 M}, \tag{F.24}$$

and, therefore, the same lower bound also holds true for $\varepsilon(M)$. Substituting this bound inside the hitting time estimate of Theorem 4 then yields the desired result, accordingly that

$$\mathbb{E}_x[\tau_M] \lesssim \frac{M}{\sigma_{\min}^2} e^{cM} \tag{F.25}$$

for some positive constant $c > 0$. ∎

**F.2. Comparison with related works: the pure noise setup.** To conclude this appendix, we discuss the main differences between the behavior of (S-FTRL) and those of (SRD-PI) and (SRD-AS) in harmonic games. In particular, we aim to clarify why the Itô correction term in (S-FTRL) is the one that preserves most of the rationally admissible properties of the deterministic case.

For this purpose, we focus on the *pure noise* regime, i.e., the case where $v \equiv 0$ across all $\mathcal{X}$ (which, by definition, is also a trivial harmonic game). In this setting, the system's behavior is entirely determined by the noise and the corresponding Itô correction term, making it a suitable approximation for analyzing the impact of noise on (S-FTRL).

To simplify the analysis, we consider uncorrelated noise where each $\sigma_{i\alpha_i}$ is independent of the players' strategies. Under these assumptions, the dynamics of (S-FTRL) reduce to the following form:

$$Y_{i\alpha_i}(t) = \sigma_{i\alpha_i} W_{i\alpha_i}(t), \quad X_i(t) = Q_i(Y_i(t)). \tag{FTRL-N}$$

Since $Q^{-1}$ is generally neither continuous nor single-valued, the behavior of $X(t)$ cannot be inferred directly from that of $Y(t)$. Instead, we must map $Y(t)$ to the *space of payoff differences* $\mathcal{Z}$(cf. Appendix B), where (FTRL-N) then takes the form:

$$Z_{i\alpha_i}(t) = \sigma_{i\alpha_i} W_{i\alpha_i}(t) - \sigma_{i\hat{\alpha}_i} W_{i\hat{\alpha}_i}(t); \quad X_i(t) = \hat{Q}_i(Z_i(t)) \tag{F.26}$$

for every $\alpha_i \neq \hat{\alpha}_i$, where $\hat{\alpha}_i \in \mathcal{A}_i$ is a fixed benchmark action and $\hat{Q}$ denotes the payoff-adjusted mirror map (whose inverse is continuous and single-valued by Lemma B.3). In particular, the long-run behaviors of $X(t)$, i.e., the classification of those as either transient or recurrent, are completely the same as that of a (correlated) Brownian motion.

This identification explains why the deterministic stability properties of (FTRL) largely persist under any level of noise: in terms of players' strategies, the induced noise essentially behaves like a standard Brownian motion and is therefore negligible compared to most drifts, as justified by the strong law of large numbers (Lemma B.5).

Additionally, this also explains why positive recurrence is impossible in harmonic games: the absence of significant drift results in trajectories of (S-FTRL) resembling ones of a (correlated) Brownian motion. Since the effective dimension of such a process is at least 2 (and finite games require at least 2 players), it generally cannot exhibit positive recurrence.

To illustrate these points, consider two specific examples of pure-noise settings:

**Example F.4** ($2 \times 2$ zero game). Let $\Gamma$ be a $2 \times 2$ zero game, i.e., a game with two players each having two strategies, and take all noise coefficients equal to 1 for simplicity. In this case, we get

$$Z_1(t) = W_{1\alpha_1}(t) - W_{1\hat{\alpha}_2}(t) = \frac{1}{\sqrt{2}} B_1(t) \tag{F.27}$$

$$Z_2(t) = W_{2\alpha_2}(t) - W_{2\hat{\alpha}_2}(t) = \frac{1}{\sqrt{2}} B_2(t) \tag{F.28}$$

where $B(t) = (B_1(t), B_2(t))$ is a standard 2-dimensional Brownian motion. Accordingly, $Z(t)$ is proportional to a 2-dimensional Brownian motion, which is known to be null recurrent from classical results (see e.g., Examples 3.11 and 3.12 of [32]). Accordingly, $X(t)$ must also be null-recurrent by Lemma B.3.

**Example F.5** ($2 \times 2 \times 2$ zero game). Let $\Gamma$ be a $2 \times 2 \times 2$ zero game, with then three players each having two strategies, and all noise coefficient equal to 1. Similarly to the previous example, we therefore obtain that $Z(t) = \frac{1}{\sqrt{3}} B(t)$ for $B(t)$ a 3-dimensional Brownian motion. Such a process is transient, so $X(t)$ must also be transient.

The previous examples highlight an important observation about (S-FTRL) in harmonic games: although trajectories converge *on average* toward the boundary by Theorem 4, this is not enough to distinguish between convergence with probability one (transience) and infinite oscillations in the strategy space (null recurrence).

With that in mind, let us now examine the different stochastic differential equations proposed to study random perturbations to the replicator dynamics (RD) (and more generally in (FTRL)), namely the *stochastic replicator dynamics with aggregate shocks* (SRD-AS) [14, 21, 24, 28] and the *stochastic replicator dynamics of pairwise imitation* (SRD-PI) [10, 17, 20, 46].

Still in the pure-noise setting and with a slight abuse of notation, both of these dynamics can be recast within the framework of (S-FTRL) (at least for the entropic regularizer from Example 1) as:

$$Y_{i\alpha_i}^{AS}(t) = -\frac{\sigma_{i\alpha_i}^2}{2}t + \sigma_{i\alpha_i}W_{i\alpha_i}(t); \quad X_i^{AS}(t) = Q_i(Y_i^{AS}(t)) \tag{FTRL-AS-N}$$

and

$$dY_{i\alpha_i}^{PI}(t) = -\frac{1}{2}\Big(1 - 2X_{i\alpha_i}^{PI}(t)\Big)\sigma_{i\alpha_i}^2 dt + \sigma_{i\alpha_i}dW_{i\alpha_i}(t); \quad X_i^{PI}(t) = Q_i(Y_i^{PI}(t)) \tag{FTRL-PI-N}$$

respectively. Mapping both (FTRL-AS-N) and (FTRL-PI-N) into the payoff differences space $\mathcal{Z}$ then yield

$$Z_{i\alpha_i}^{AS}(t) = -\frac{1}{2}(\sigma_{i\alpha_i}^2 - \sigma_{i\hat{\alpha}_i}^2)t + \sigma_{i\alpha_i}W_{i\alpha_i}(t) - \sigma_{i\hat{\alpha}_i}W_{i\hat{\alpha}_i}(t) \tag{AS-Z}$$

and

$$dZ_{i\alpha_i}^{PI}(t) = -2\Big(X_{i\hat{\alpha}_i}^{PI}(t) - X_{i\alpha_i}^{PI}(t)\Big)dt + \sigma_{i\alpha_i}dW_{i\alpha_i}(t) - \sigma_{i\hat{\alpha}_i}dW_{i\hat{\alpha}_i}(t). \tag{PI-Z}$$

To highlight their differences, we discuss each of these dynamics in separate examples.

**Example F.6** (Stochastic replicator dynamics with aggregate shocks). Notice that the drift in (AS-Z) is deterministic and only depend on the relative difference of noise magnitude. By the strong law of large numbers (Lemma B.5), we therefore get the following classification of its behaviors:

1. If $\sigma_{i\alpha_i}^2 > \sigma_{i\hat{\alpha}_i}^2$ for some $\alpha_i \neq \hat{\alpha}_i$, then $Z_{i\alpha_i}^{AS}(t) \to -\infty$ (a.s.), and so $X_{i\alpha_i}^{AS}(t) \to 0$ (a.s.): trajectories are *transient*.

2. If $\sigma_{i\alpha_i}^2 < \sigma_{i\hat{\alpha}_i}^2$ for some $\alpha_i \neq \hat{\alpha}_i$, then $Z_{i\alpha_i}^{AS}(t) \to \infty$ (a.s.), and so $X_{i\hat{\alpha}_i}^{AS}(t) \to 0$ (a.s.): trajectories are *transient*.

3. If $\sigma_{i\alpha_i} \equiv \sigma_i$ for every $\alpha_i \in \mathcal{A}$, then $Z_{i\alpha_i}^{AS}(t) = \sigma_i(W_{i\alpha_i}(t) - W_{i\hat{\alpha}_i}(t))$: same behavior as (FTRL-N).

This classification can also be recovered from results proved in [24, 28] concerning the extinction of dominated strategies and stability of equilibria under (SRD-AS).

**Example F.7** (Stochastic replicator dynamics of pairwise imitation). Note that if $X^{PI}(t)$ remains close enough to a pure strategy, say $\hat{\alpha}_i$, then the drift of (PI-Z) is strictly decreasing with order $-t$ for every $\alpha_i \neq \hat{\alpha}_i$. In other words, trajectories have a strong tendency to drift toward pure strategies. Moreover, a similar argument to the one developed in the proof of Theorem 2 then shows that each pure strategy is stochastically asymptotically stable under (FTRL-PI-N). As a result, trajectories are always transient and attracted strongly toward pure strategies. This aligns with the findings of Engel & Piliouras [17] in two-player zero-sum games, where pure strategies are attractive in terms of their invariant measures, and trajectories converge to the boundary regardless of the noise level.

From the previous examples, we conclude that both (SRD-AS) and (SRD-PI) exhibit a strong bias towards the strategy boundary, even when the payoffs do not favor any particular outcome. In contrast, (S-FTRL) does not show such tendencies and behaves as we would expect in a pure noise setting, namely, similarly to white noise (Brownian motion).

