# OpenReview forum: "The impact of uncertainty on regularized learning in games"
_ICML.cc/2025/Conference — ICML 2025 poster_

### Official Review · Reviewer_fPnc · 2025-03-06

**Overall Recommendation:** 3

**Summary:**

The paper examines the behavior of the stochastic FTRL dynamics in games, in which the usual FTRL algorithm is randomly perturbed. It provides a general characterization: every player reaches an almost pure strategy in finite time. This stands in contrast to the deterministic setting. A consequence of the previous result is that the only possible limits of stochastic FTRL are pure Nash equilibria. The final main result shows that a span of pure strategies is attracting if and only if it is closed under better replies, which mirrors some earlier results in the deterministic setting.

**Claims And Evidence:**

All claims made in the paper are sound and supported by clear evidence.

**Essential References Not Discussed:**

All relevant references have been discussed; I did not identify any notable omission. Overall, the paper adequately places its contributions in the context of the related work.

**Experimental Designs Or Analyses:**

Not applicable; the paper makes a theoretical contribution.

**Methods And Evaluation Criteria:**

Not applicable; the paper makes a theoretical contribution.

**Other Comments Or Suggestions:**

- A minor typo: after (8) there should be a comma.

- In section 5, it is claimed that harmonic games is a much broader class than zero-sum games; but, if I am not missing something, this is not really true.

**Other Strengths And Weaknesses:**

Overall, the paper makes concrete contributions to a fundamental problem in multi-agent systems. It provides a comprehensive characterization of the behavior of stochastic FTRL in the continuous-time setting. Perhaps the most surprising result is Theorem 1, as it represents a departure from the deterministic setting. The main message here is very clear and interesting: stochasticity favors pure strategies. Theorems 2 and 3 mirror some existing results from the deterministic setting, but are still interesting and new. From a technical standpoint, the paper is highly non-trivial. The writing and organization of the paper are also exemplar, and it was a joy to read the paper.

**Questions For Authors:**

No further questions.

**Relation To Broader Scientific Literature:**

The paper is part of a broader effort to characterize the behavior of natural learning algorithms in multi-player games. While most prior work has focused on the deterministic setting, this paper makes progress on the stochastic setting.

**Theoretical Claims:**

All claims appear to be sound; I did not find any notable issues.

---

> ### Author Rebuttal · Authors · 2025-04-01
>
> Dear reviewer,
>
> Thank you for your time, input, and overall appreciation, both in terms of results and presentation! We reply to your remarks and questions below:
>
> > Theorems 2 and 3 mirror some existing results from the deterministic setting, but are still interesting and new.
>
> Indeed, Theorems 2 and 3 extend the result by [15] on the asymptotic convergence of FTRL to strict Nash equilibria. However, as you rightly point out, generalizing this result to the stochastic setting is far from straightforward. The analysis requires substantially different techniques and introduces non-trivial challenges that do not arise in the deterministic case. In particular, while carrying on those results to our stochastic FTRL setting works as intended, this is not the case at all when considering different (yet rational) stochastic models. For instance, they are not true for the stochastic replicator dynamics with aggregate shocks (SRD-AS) nor for the stochastic replicator dynamics of pairwise imitation (SRD-PI); both of which that can be seen as stochastic versions of the (continuous-time) multiplicative weights algorithm.
>
> > (..) after (8) there should be a comma
>
> Will fix, thanks for spotting it!
>
> > In section 5, it is claimed that harmonic games is a much broader class than zero-sum games; but, if I am not missing something, this is not really true.
>
> To be clear, in the context of Poincaré recurrence (Section 5), any reference to zero-sum games should be understood as shorthand for “two-player zero-sum games with a fully mixed equilibrium”, or $\text{2PZS}\*$ for brevity. The claim that harmonic games form a significantly broader class than $\text{2PZS}\*$ simply referred to the fact that, while every $\text{2PZS}\*$ game is harmonic (cf. Corollary E.1), harmonic games allow for any number of players, and the players' payoffs need not sum to zero. We will make this point more explicit, thanks for flagging it.
>
> ---
> We thank you again for your time and positive evaluation! Please let us know if there are any lingering questions in the above.
>
> Kind regards,
>
> The authors

---

### Official Review · Reviewer_yDto · 2025-03-11

**Overall Recommendation:** 3

**Summary:**

This work investigates the impacts of the noises on the observed payoffs when applying continuous-time FTRL for game solving. The main conclusions are that every player will reach an almost pure strategy in finite time and the limit of the dynamics of the continuous-time FTRL will converge to the pure Nash equilibria.

**Claims And Evidence:**

Yes.

**Essential References Not Discussed:**

No.

**Experimental Designs Or Analyses:**

Not applicable.

**Methods And Evaluation Criteria:**

Yes.

**Other Comments Or Suggestions:**

NA.

**Other Strengths And Weaknesses:**

**Strengths**
1. **Theoretical Results**: The theoretical results in this work seem to be new and might be of great interest to the community.
2. **Writing**: This paper is generally well-written.

**Weaknesses**: If any, I would be curious about the proof idea and basic techniques that lead to the main results in this paper, which do seem to be discussed in the main body of the paper. I think the results in this work are fundamental to the literature so I'm a bit surprised that these results weren't discovered in previous work. Therefore, I think it will benefit the readers a lot if the proof sketches and techniques of the main results are given.

Overall, I think the results in this paper are valuable, but I am not fully familiar with the techniques of continuous FTRL, so I would recommend a weak acceptance of this work while maintaining a low confidence level.

**Questions For Authors:**

1. If I understand correctly, the results in this work hold in the self-play setting, where each player adopts the same FTRL algorithm. Can the noise $M_i(t)$ on the LHS of Line 187 account for the case where some players are potential adversaries?
2. I am a bit confused about some statements about the main results. Does the conclusion (c) on the RHS of Line 431 imply the conclusion (a) and (b) on the RHS of Line 429-431? That is, can we say “the strategy of every player reaches an almost pure Nash equilibrium in finite time” and “every player’s limit set contains a pure Nash equilibrium strategy”?

**Relation To Broader Scientific Literature:**

To my knowledge, the findings are new and technically valuable.

**Theoretical Claims:**

I did not check the correctness of the proof details.

---

> ### Author Rebuttal · Authors · 2025-04-01
>
> Dear reviewer,
>
> Thank you for your time, input, and positive evaluation! We reply to your remarks and questions below:
>
> > I would be curious about proof idea and basic techniques that lead to the main results (…) I'm a bit surprised that these results weren't discovered in previous work. Therefore, I think it will benefit the readers a lot if the proof sketches and techniques of the main results are given.
>
> We also very much aligned with the reviewer's opinion that more extensive proof sketches would be of benefit ot the reader, but we were unfortunately limited due to space constraints. That said, we will be happy to use the first revision opportunity (and the extra page provided) to add an overview of the technical apparatus and trajectory required to prove our results; note that all the details are already provided in the appendix.
>
> > If I understand correctly, the results in this work hold in the self-play setting, where each player adopts the same FTRL algorithm. Can the noise $M_i(t)$ on the LHS of Line 187 account for the case where some players are potential adversaries?
>
> The "adversarial" setting that you describe would be more aptly captured by making the evolution $X_i(t)$ of certain players arbitrary. The noise $M(t)$ has a lot of statistical features, so it cannot simply be replaced by an arbitrary stream of payoff-like observations.
>
> > Does the conclusion (c) on the RHS of Line 431 imply the conclusion (a) and (b) on the RHS of Line 429-431? That is, can we say “the strategy of every player reaches an almost pure Nash equilibrium in finite time” and “every player’s limit set contains a pure Nash equilibrium strategy”?
>
> To clarify, the formal version of conclusions (a), (b), (c) is as follows:
> - (a) For every neighborhood $U_{i}$ of pure strategies for player $i$, there exists a finite (random) time $t$ such that $X_{i}(t) \in U_{i}$ (**Theorem 1**).
> - (b) The (random) limit set $\lbrace x_{i} \in \mathcal{X_{i}} : X_i(t_{n}) \to x_{i} \text{ for some sequence } t_{n} \nearrow \infty  \rbrace$ contains (at least) a pure strategy (**Corollary 1**).
> - (c) If there exists a strategy $x^{\*}$ such that $X(t) \to x^{\*}$ with positive probability, then it is necessarily a pure Nash equilibrium (**second part of Theorem 3**).
>
> Accordingly, (c) only make sense for trajectories that _do converge_ (in a pointwise sense), whereas conclusions (a) and (b) hold for every trajectories even those that do not converge to a point (for instance, those that wander indefinitely in the state space or those that converge only toward some subface of the boundary). As such, (c) does not imply any of the other two conclusions, and it is thus not possible to conclude that "the strategy of every player reaches an almost pure Nash equilibrium in finite time" nor that "every player’s limit set contains a pure Nash equilibrium strategy" (in fact, there may not even be any pure Nash equilibrium in some game, so those conclusions cannot hold in full generality).
>
> ---
> We thank you again for your time and positive evaluation! Please let us know if there are any lingering questions in the above.
>
> Kind regards,
>
> The authors

---

> > ### Comment · Reviewer_yDto · 2025-04-03
> >
> > I would like to thank the authors for the detailed responses. I have no further questions and would like to keep my rating for this work unchanged.

---

### Official Review · Reviewer_57JY · 2025-03-14

**Overall Recommendation:** 3

**Summary:**

This paper investigates the impact of uncertainty on the dynamics of the Follow-The-Regularized-Leader (FTRL) algorithm. The author first shows that under uncertainty, the FTRL algorithm approaches a pure strategy. Then, the author demonstrates that pure strategies are the only possible limit points of the stochastic FTRL. Additionally, the paper proves that the recurrent behavior observed in FTRL dynamics under deterministic settings disappears when uncertainty is introduced.

**Claims And Evidence:**

The claims are generally supported by mathematical proofs.

**Essential References Not Discussed:**

The paper appropriately cites relevant prior works on learning in games under the noisy feedback setting.

**Experimental Designs Or Analyses:**

The empirical results on (twisted) matching pennies, entry deterrence, and harmonic games are provided.

**Methods And Evaluation Criteria:**

Since FTRL is a well-studied algorithm for learning in games, it is meaningful to investigate its behavior under uncertainty.

**Other Comments Or Suggestions:**

Since I’m not familiar with harmonic games, it’s unclear to me how the harmonic center $q$ is determined in a two-player zero-sum game setting.

Furthermore, in the discrete-time setting, it might be possible to control the noise level by adjusting a learning rate sequence. By doing so, is there any possibility of preventing the FTRL dynamics from approaching a pure strategy?

**Other Strengths And Weaknesses:**

Please see “Other Comments Or Suggestions”.

**Questions For Authors:**

Please see “Other Comments Or Suggestions”.

**Relation To Broader Scientific Literature:**

The theoretical results seem novel compared to the existing literature. However, I am curious about the difference between Corollary 3, Theorem 3, and the results presented by [19]. Do Corollary 3 and Theorem 3 generalize their results to broader settings?

Furthermore, as far as I understand, it is well known that FTRL satisfies the no-regret property even in the stochastic setting. I am wondering if this fact contradicts the derived theorems and corollaries in the paper.

**Theoretical Claims:**

The proofs employ standard techniques for stochastic differential equations. While I did not verify all of the proofs in the paper, the presented theorems and corollaries appear rigorous and align with my intuition.

---

> ### Author Rebuttal · Authors · 2025-04-01
>
> Dear reviewer,
>
> Thank you for your time, input, and positive evaluation! We reply to your remarks and questions below:
>
> >I am curious about the difference between Corollary 3, Theorem 3, and the results presented by [19]. Do Corollary 3 and Theorem 3 generalize their results to broader settings?
>
> Regarding Theorem 3, the general setting of [19] and our own are qualitatively similar - regularized learning with stochastic feedback - but quantitatively quite different - discrete vs. continuous-time, a decreasing step-size in [19], different assumptions on the noise, etc. Thus, even though Part I of Theorem 3 of our paper and the corresponding result of [19] involve similar notions - stochastic asymptotic stability - they are otherwise completely distinct as results.
>
> In addition, the extension of the discrete-time analysis of [19] to the continuous-time setting of our paper is far from trivial. One major difficulty that arises has to do with the way noise is injected into the system, and which plays a crucial role in the long-run behavior of the dynamics . For instance, the stated result breaks down completely when considering the stochastic replicator dynamics with aggregate shock (SRD-AS); the stochastic replicator dynamics of pairwise imitation (SRD-PI), for which stable points of trajectories usually need to be strict Nash equilibria in a noise-adjusted payoff field, cf. [21, 37].
>
> In a similar vein, Corollary 3 and the second part of Theorem 2 are totally new results compared to [19]: Instead of characterizing only (stochastically asymptotically) stable strategies as in [19], those results also identify which states can appear as (possibly non-stable) limits of the stochastic dynamics. In words, **pure Nash equilibria are the only strategies appearing as pointwise limits of (S-FTRL)**, and among these equilibria only those that are strict can be (and are) stochastically stable.
>
> > (...) as far as I understand, it is well known that FTRL satisfies the no-regret property even in the stochastic setting. I am wondering if this fact contradicts the derived theorems and corollaries in the paper.
>
> Again, this is a matter of the general setting of each paper: the no-regret property of FTRL is, indeed, classical in discrete time; in continuous time however, the situation is much more complicated, and the only no-regret result that we're aware of is [10], which involves a vanishing learning rate or noise that becomes vanishingly small over time. Both of these postulates are orthogonal to our paper, where we assume positive noise covariance (Assumption 1), which, in turn, rules out the "vanishing noise" framework..
>
> > (...) it’s unclear to me how the harmonic center is determined in a two-player zero-sum game setting.
>
> A generic two-player zero-sum game is not harmonic; however, if it admits a fully mixed Nash equilibrium, then it is harmonic (cf. Corollary E.1). In that case, the harmonic center coincides with the Nash equilibrium and can be calculated in the same way.
>
> > In the discrete-time setting, it might be possible to control the noise level by adjusting a learning rate sequence. By doing so, is there any possibility of preventing the FTRL dynamics from approaching a pure strategy?
>
>
> Great question, thanks for raising it! We conjecture that adding a slowly decreasing learning rate as per [10] (decreasing to zero slowly enough over time) would mitigate the escape to the boundary, so we would expect the result that "any player's choices approach pure strategies" to be false in this setting. However, because of the extra terms introduced in the analysis by Itô's formula, the calculations are far from trivial and require a drastically different approach.
>
> Considering a learning rate also opens a vast array of other interesting non-trivial questions, such as what happens to the "non-recurrence" results of Section 5 or what can we deduce on the discrete-time FTRL algorithm. As those questions typically require different and more precise tools than the ones we have used here (the stochastic dynamics becoming time-inhomogeneous due to the presence of a time-dependent learning rate), we have chosen to not explore those directions in this paper, and to defer such considerations to future work.
>
> ---
> We thank you again for your time and positive evaluation! Please let us know if any of the above points is not sufficiently clear.
>
> Kind regards,
>
> The authors

---

### Official Review · Reviewer_BYeZ · 2025-03-14

**Overall Recommendation:** 4

**Summary:**

The authors consider the continuous time limit of the FTRL dynamics under noisy observations of game payoff matrixes and prove that, unlike in the noiseless case, the dynamics converge to pure Nash, along with several other results.

## Update After Rebuttal
The authors answered my questions quite well -- I might suggest adding in parts of said answer to the discussion section of the paper, as space permits.

**Claims And Evidence:**

Yes.

**Essential References Not Discussed:**

N/A

**Experimental Designs Or Analyses:**

N/A

**Methods And Evaluation Criteria:**

N/A

**Other Comments Or Suggestions:**

N/A

**Other Strengths And Weaknesses:**

(+) I thought this paper was exceptionally well written -- even as someone without a strong background in the area, I could understand the high-level theoretical take-aways and implications thereof. Kudos to the authors on this -- I often struggle to do this myself on papers this theoretical :).

**Questions For Authors:**

These questions are not particularly important but I'd be curious for responses if the authors have time:

1) I'm mostly familiar with the discretized, discrete-time version of FTRL. Could you comment on how your results would transfer to this setting? Like, with noisy payoffs, should I expect discrete-time FTRL to avoid circling equilibria and only converging on average?

2) When I think of the discrete-time FTRL dynamics, people will often use ideas like optimism to avoid the player strategies circling the Nash equilibrium (i.e., only converging on average rather than for the last iterate). My understanding is that the continuous-time analog of these results requires taking some "high-resolution" limits (e.g. https://arxiv.org/abs/2112.13826). I was wondering if there is anything interesting one could say about that flavor of approach in comparison to the ideas explored here.

**Relation To Broader Scientific Literature:**

Nope.

**Theoretical Claims:**

Nope.

---

> ### Author Rebuttal · Authors · 2025-04-01
>
> Dear reviewer,
>
> Thank you for your time, positive evaluation and encouraging words! We reply to your remarks and questions below:
>
> > I'm mostly familiar with the discretized, discrete-time version of FTRL. Could you comment on how your results would transfer to this setting? Like, with noisy payoffs, should I expect discrete-time FTRL to avoid circling equilibria and only converging on average?
>
> This is a very interesting question, thanks for raising it!
>
> In a nutshell, it depends.
>
> To begin, there are several factors that come into play - whether FTRL is run with a constant or vanishing step-size / learning rate, the origin of the noise in the algorithm (e.g., pure payoff vector information versus bandit, payoff-based feedback) and, of course, the specific result under study.
>
> - For games with "circling FTRL dynamics" (e.g., zero-sum games with a fully mixed equilibrium, or harmonic games) our intuition is as follows:
> If the algorithm is run with a vanishing step-size, it will most likely converge to some random (Bregman) distance from the equilibrium / center of the game, but it won't necessarily return close to where it started.
> - If the algorithm is run with a constant step-size, the picture is less clear: some first results in [1] for the exponential weights algorithm with pure payoff vector information suggest that the algorithm converges to the boundary in two-player zero-sum games. We conjecture that an analogue of Theorems 1 and 5 also holds in this setting, but this is likely a paper in itself.
>
> [1] Bailey et al. "Stochastic Multiplicative Weights Updates in Zero-Sum Games"
>
>
> > I was wondering if there is anything interesting one could say about that flavor of approach [optimism and high-resolution limit] in comparison to the ideas explored here.
>
> Agreed: in discrete time with full information (that is, perfect observations of the players' mixed payoff vectors), optimism can mitigate non-convergence in certain games where the continuous-time dynamics are Poincaré recurrent (such as two-player zero-sum games or harmonic games). However, this gain only manifests itself in the deterministic case; in the stochastic case, optimistic methods (and other extrapolation-based methods, like extra-gradient and its variants) fail to converge altogether and, as you say, only time-averages remain convergent. Because of this, it does not seem that the properties of FTRL would be particularly different from optimistic FTRL in the presence of noise, because the noise in the process would overshadow the finer, smooth structure of optimistic FTRL. For similar reasons, a higher-resolution approximation would not help either, because the noise would cancel any gain obtaine from using a finer discretization scheme.
>
> ---
>
> Thank you again for your time and encouraging words - please do not hesitate to reach out if you have any further questions!
>
> Kind regards,
>
> The authors

---

### Decision · Program_Chairs · 2025-05-01

**Decision:**

Accept (poster)

**Comment:**

All reviewers appreciate the contributions of the paper.